# Trajectory Consistency for One-Step Generation on Euler Mean Flows

Zhiqi Li [1]   Yuchen Sun [1]   Duowen Chen [1]   Jinjin He [1]   Bo Zhu [1]

## Abstract

We propose *Euler Mean Flows (EMF)*, a flow-based generative framework for one-step and few-step generation that enforces long-range trajectory consistency with minimal training cost. The key idea of EMF is to replace the trajectory consistency constraint, which is difficult to supervise and optimize over long time scales, with a principled linear surrogate that enables direct data supervision for long-horizon flow-map compositions. We derive this approximation from the semigroup formulation of flow-based models and show that, under mild regularity assumptions, it faithfully approximates the original consistency objective while being substantially easier to optimize. This formulation leads to a unified, JVP-free training framework that supports both $u$-prediction and $x_1$-prediction variants, avoiding explicit Jacobian computations and significantly reducing memory and computational overhead. Experiments on image synthesis, particle-based geometry generation, and functional generation demonstrate improved optimization stability and sample quality under fixed sampling budgets, together with approximately $50\%$ reductions in training time and memory consumption compared to existing one-step methods for image generation.

## 1. Introduction

Recent advances in generative modeling, particularly diffusion models and flow matching methods, have achieved remarkable success in image generation (Lipman et al., 2023; Song et al., 2021a), video synthesis (Ho et al., 2022b;a), and 3D geometry modeling (Luo & Hu, 2021; Vahdat et al., 2022; Zhang et al., 2025a). From a continuous-time perspective, these methods can be unified by the continuity equation, which learns a time-dependent velocity field of probability flow to transform simple noise distributions into complex data distributions (Lipman et al., 2024). Under this formulation, the generation process corresponds to a continuous trajectory evolving from noise space to data space, and model training aims to characterize the dynamics of this flow-map trajectory at different time points.

While such trajectory-based models provide strong expressive power, sampling from the learned dynamics typically requires a large number of time steps, resulting in substantial inference cost. To improve efficiency, a growing body of recent work focuses on one- and few-step generation, aiming to approximate long sampling trajectories with only a small number of steps (Song et al., 2023; Frans et al., 2025; Guo et al., 2025; Geng et al., 2025a), thereby reducing inference time while maintaining competitive generation quality. A central challenge in one-step and few-step generation lies in learning trajectory consistency (Frans et al., 2025; Guo et al., 2025), meaning that predictions at different points along the trajectory should agree with each other.

Mathematically, trajectory consistency can be characterized by the semigroup property: for all $t \le s \le r$, the flow maps satisfy $\phi_{t \to r} = \phi_{s \to r} \circ \phi_{t \to s}$. Here, $\phi_{t \to r} : \mathcal{X} \to \mathcal{X}$ maps a state from time $t$ to time $r$ along the underlying dynamics, so that $\phi_{t \to r}(x_t) = x_r$ for any trajectory $(x_t)_{t \in [0,1]}$. This property promotes coherent flow maps across time scales (Pazy, 2012). However, trajectory consistency alone is insufficient as data supervision for generative modeling: standard flow-based models (Lipman et al., 2023; Liu et al., 2023) do not provide an explicit data-derived reference map $\phi_{t \to r}$, nor its conditional counterpart, for long-range transport. Thus, consistency constraints may enforce a valid compositional structure, but they do not necessarily inject target-distribution information or guarantee that the learned long map matches the data distribution. Poorly formulated consistency objectives may further disrupt the flow-map structure, causing unstable training or degraded generation quality (Boffi et al., 2025).

Existing approaches for addressing this issue can be categorized into two classes. The first category methods progressively extend short-range transitions to longer intervals by composing locally learned dynamics (Frans et al., 2025; Guo et al., 2025). Although conceptually simple, such methods suffer from error accumulation for trajectories, as

---

[1]Georgia Institute of Technology, Atlanta, GA, USA. Correspondence to: Zhiqi Li <zli3167@gatech.edu>.

*Proceedings of the $43^{rd}$ International Conference on Machine Learning*, Seoul, South Korea. PMLR 306, 2026. Copyright 2026 by the author(s).

long-range behavior is inferred indirectly from short-range estimates without explicit global supervision. The second class, represented by MeanFlow and related methods (Geng et al., 2025a; Zhang et al., 2025b), derives training objectives directly from continuity equations. By introducing consistency constraints at the level of flow maps, these methods provide principled supervision for long-range dynamics. However, they rely on explicit gradient computation with several practical limitations: (1) Explicit gradient computation incurs substantial memory and computational overhead that limits efficient network architectures and training procedures (e.g., FlashAttention (Dao et al., 2022)). (2) Incorporating explicit gradients into the loss may lead to numerical instability, especially under mixed-precision training, as observed in our image and SDF generation experiments. (3) Gradient-based objectives are poorly compatible with sparse computation primitives, limiting their applicability to domains such as point cloud modeling.

In this work, we propose a new approach for trajectory-consistent one-step generation by revisiting the semigroup structure of flow maps. Our key idea is to apply a local linearization to the trajectory consistency equation and enable direct supervision from the data distribution for long-range flow maps. This linear approximation transforms the original long-range consistency constraint into a learnable surrogate objective without calculating derivatives. We proved that, under reasonable conditions (Assumption 1 and Theorem 4.3), this surrogate loss faithfully approximates the original consistency objective and enables accurate learning of the instantaneous velocity along long-range flow maps. Based on this analysis, we further develop a gradient-free training framework that significantly reduces memory and computational cost and leads to more stable optimization. Motivated by the manifold assumption advocated in (Li & He, 2025), we formulate a unified framework for one-step and few-step generation that supports both $u$-prediction and $x_1$-prediction, with the latter emphasizing direct supervision of the terminal state of the flow, which is particularly useful for high-dimensional domains. Our linearized formulation is inspired by Euler time integration (Hairer et al., 1993) in numerical analysis; accordingly, we refer to our approach as *Euler Mean Flows* (EMF).

Our main contributions are summarized as follows:

- We propose *Euler Mean Flows (EMF)*, a trajectory-consistent framework for one-step and few-step generation based on a linearized semigroup formulation.

- We introduce a surrogate loss obtained by local linearization of the semigroup consistency objective, with theoretical guarantees under mild assumptions.

- We develop a unified, JVP-free training scheme that avoids explicit derivative computations and supports both $u$-prediction and $x_1$-prediction variants, leading to more stable optimization, lower training overhead, and broader applicability across diverse domains.

## 2. Related Work

**Diffusion and Flow Matching.** Diffusion models (Ho et al., 2020; Song & Ermon, 2019; Song et al., 2021b) have achieved remarkable success in data generation by progressively denoising random initial samples to produce high-quality data. This generative process is commonly formulated as the solution of stochastic differential equations (SDEs). In contrast, Flow Matching methods (Liu et al., 2023; Lipman et al., 2023; Albergo & Vanden-Eijnden, 2023) learn the velocity fields that define continuous flow trajectories between probability distributions.

**Few-step Diffusion/Flow Models.** Consistency models (Song et al., 2023; Song & Dhariwal, 2023; Geng et al., 2025c; Lu & Song, 2025) were proposed as independently trainable one-step generators, alongside distillation-based approaches (Salimans & Ho, 2022; Meng et al., 2023; Geng et al., 2023). Recent works extend self-consistency principles to related generative frameworks (Yang et al., 2024; Frans et al., 2025; Zhou et al., 2025). Mean Flow (Geng et al., 2025a) learns time-averaged velocities by differentiating the Mean Flow identity, while $\alpha$-Flow (Zhang et al., 2025b) improves training by disentangling conflicting components in the Mean Flow objective and combining it with Shortcut-style consistency (Frans et al., 2025). SplitMeanFlow (Guo et al., 2025) removes Jacobian–vector products (JVPs) via interval-splitting consistency. Flow Map Matching (Boffi et al., 2025) provides a unified view of Mean Flow, SplitMeanFlow, ShortCut, and related flow-map objectives. While both SplitMeanFlow and EMF are JVP-free, SplitMeanFlow is distillation-based, whereas EMF supports independent training with direct supervision for long-range flow maps.

## 3. Background

Let $\mathcal{D} = \{x^i \in \mathcal{X}\}_{i=1}^n$ be a dataset drawn from an unknown data distribution $p_{\text{data}}$ on space $\mathcal{X}$. Flow Matching aims to learn $p_1 = p_{\text{data}}$ by learning a continuous-time velocity field $u(x,t)$, $t \in [0,1]$, that transports a base distribution $p_0$, typically Gaussian distribution $\mathcal{N}(0, \sigma^2)$, to $p_1$ along a continuous path of distributions $(p_t)_{t \in [0,1]}$. The evolution of the distribution path is governed by the continuity equation

$$\frac{\partial}{\partial t} p_t(x) + \nabla \cdot \big(p_t(x) u_t(x)\big) = 0 \qquad (1)$$

Given learned $u_t(x)$, sampling $x_0 \sim p_0$, samples from $p_1$ can be obtained by integrating the ODE

$$\frac{dx_t}{dt} = u(x_t, t), \quad x_0 \sim p_0 \qquad (2)$$

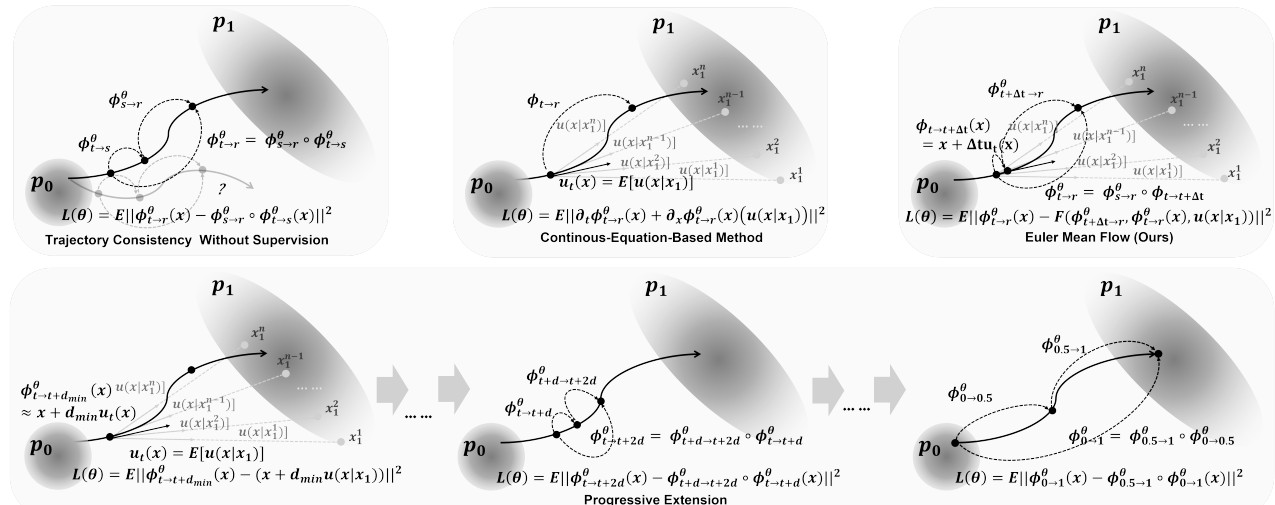

*Figure 1.* Illustration of trajectory consistency and Euler Mean Flow (EMF). Top-left: Trajectory consistency alone only enforces self-consistency and does not identify the data-transporting solution; translucent flow maps denote alternative self-consistent trajectories that fail to transport $p_0$ to $p_1$. Top-middle: Continuous-equation-based methods provide a more direct long-range objective from the continuous consistency equation, but require JVP-based derivative computation. Bottom: Progressive extension methods learn local dynamics from data and recursively extend them to longer horizons, making long-range supervision indirect. Top-right: EMF uses a local Euler linearization to obtain a direct data-supervised target for long flow maps without JVPs, where $F$ denotes the resulting linear function.

The associated flow $\phi_t : \mathcal{X} \to \mathcal{X}$ is defined by $\phi_t(x_0) = x_t$ for any $x_0, x_t$ satisfying the ODE and it satisfies

$$\frac{\partial}{\partial t}\phi_t = u_t \circ \phi_t, \quad \phi_0 = \mathrm{Id}_{\mathcal{X}}. \tag{3}$$

We further define the flow map $\phi_{t \to r} = \phi_r \circ \phi_t^{-1}$. The path $p_t$ can be written as a pushforward $p_t = (\phi_t)_\sharp p_0$.

Flow Matching seeks to learn the velocity field $u_t(x)$. Given a parameterized model $u_t^\theta(x)$, samples are generated by numerically integrating Equation 2 from $t = 0$ to $t = 1$. A natural training objective for training is $\mathcal{L}^{FM}(\theta) = \mathbb{E}_{t,x \sim p_t(x)}\|u_t^\theta(x) - u_t(x)\|^2$, which directly matches the model velocity to the reference velocity field. However, this objective cannot be optimized in practice, since both $u_t(x)$ and the marginal distribution $p_t(x)$ are not directly observable from the dataset. To incorporate supervision from data, Flow Matching introduces conditional velocities $u_t(x|x_1) = \frac{x_1 - x}{1 - t}$ and conditional flows $\phi_t(x|x_1) = t(x_1 - x) + x$ for arbitrary $x_1 \in \mathcal{X}$. These conditional quantities induce a conditional distribution $p_t(x|x_1) = (\phi(\cdot|x_1))_\sharp p_0$, and the marginal velocity field and distribution can then be recovered by marginalization $u_t(x) = \mathbb{E}_{x_1 \sim p_t(x_1|x)}[u_t(x|x_1)]$ and $p_t(x) = \mathbb{E}_{x_1 \sim p_t(x_1|x)}[p(x|x_1)]$ respectively. Based on these constructions, Flow Matching defines the conditional surrogate objective $\mathcal{L}_c^{FM}(\theta) = \mathbb{E}_{t,x_1 \sim p_{data}, x \sim p_t(x|x_1)}\|u_t^\theta(x) - u_t(x|x_1)\|^2$, which admits supervision from data samples. It has been shown that $\nabla_\theta \mathcal{L}_c^{FM}(\theta) = \nabla_\theta \mathcal{L}^{FM}(\theta)$ and therefore $\mathcal{L}_c^{FM}$ serves as a valid surrogate for optimizing $\mathcal{L}^{FM}$.

Flow Matching learns the instantaneous velocity field $u_t(x)$. As a result, sample generation requires iterative numerical

integration, making it inherently a multi-step process. In contrast, one-step and few-step generative models aim to directly learn the flow maps $\phi_{t \to r}$, enabling efficient generation with a small number of transitions.

## 4. One-Step Generation on Euler Mean Flows

A valid generative flow map should satisfy (1) generative transport to the data distribution, i.e., $[\phi_{0 \to 1}]_\sharp p_0 = p_1$, and (2) trajectory consistency. Satisfying trajectory consistency alone is easy; the central challenge is to enforce trajectory consistency while matching the data distribution, which hinders the learning of accurate long-range dynamics. In this section, we study how to introduce effective data-driven supervision for long-range trajectory consistency and present Euler Mean Flow with its theoretical justification and the $x_1$-prediction variant.

### 4.1. Challenge of Trajectory Consistency

Consider a trajectory $(x_t)_{t \in [0,1]}$, with $x_t = \phi_t(x_0)$, that satisfies Equation 2, where $\phi_t$ denotes the flow defined in Equation 3. For any $t \le s \le r$ with $t, s, r \in [0, 1]$, the following **trajectory consistency** holds:

$$\phi_{t \to r}(x_t) = \phi_{s \to r}(x_s), x_s = \phi_{t \to s}(x_t) \tag{4}$$

as illustrated in Figure 1. Taking the limit $s \to t$, this formulation admits a continuous formulation,

$$\partial_t \phi_{t \to r}(x) + \partial_x \phi_{t \to r}(x)(\partial_s \phi_{t \to s}(x))|_{s=t} = 0 \tag{5}$$

Leveraging the trajectory consistency formulation, we can derive discrete trajectory consistency loss $\mathcal{L}^C(\theta)$ to train

a long-range model $\phi_{t\to r}^{\theta}(x_t)$ that represents transitions across arbitrary temporal horizons $(t, r)$.

$$\mathcal{L}^C(\theta) = \mathbb{E}_{t,s,r,x_t=(1-t)x_0+tx_1,x_1\sim p_{data},x_0\sim p_0}$$
$$\frac{1}{w(t,r)}\|\phi_{t\to r}^{\theta}(x_t) - \phi_{s\to r}^{\theta}(\phi_{t\to s}^{\theta}(x))\|_2^2 \quad (6)$$

where $\frac{1}{w(t,r)}$ denotes a tunable weight. For efficiency, parts of the formulation can be implemented with a stop-gradient operator (sg) without altering the underlying semantics.

However, trajectory consistency alone is insufficient to uniquely determine the flow map. In particular, the consistency constraint $\mathcal{L}^C(\theta)$ admits infinitely many solutions and does not, by itself, introduce supervision from the data distribution. This ambiguity stems from two fundamental issues: (1) Like velocity fields in Flow Matching, flow maps $\phi_{t\to r}(x)$ do not admit an analytic reference derived from the data distribution $p_{data}$ and dataset $\mathcal{D}$; (2) Flow maps do not possess a conditional counterpart $\phi_{t\to r}(x|x_1)$ analogous to conditional velocities $u_t(x|x_1)$ which could calculate from dataset, as formalized below. These issues make it difficult to inject data supervision directly into $\mathcal{L}^C(\theta)$.

**Theorem 4.1** (Non-existence of conditional flow maps). *There exist no conditional flow maps $\phi_{t\to r}(x|x_{t_1})$ that simultaneously (i) is consistent with the conditional velocity $u(x|x_1)$ under Equation 3, and (ii) satisfies the consistency relation $\phi_{t\to r}(x) = \mathbb{E}_{x_1\sim p_t(x_1|x)}[\phi_{t\to r}(x|x_1)]$ with marginal flow maps. As a result, a self-consistent conditional cumulative field does not exist. (See subsection B.1 for a proof.)*

Existing methods address this indeterminacy mainly through two strategies. **Progressive Extension** methods, such as Split-Mean Flow (SplitMF) (Guo et al., 2025), PSD (Boffi et al., 2025), and ShortCut (Frans et al., 2025), first learn local dynamics and then extend them to longer horizons using the trajectory-consistency constraint in Equation 6. For example, ShortCut parameterizes the flow map as $\phi_{t\to r}^{\theta}(x) = x + (r-t)u_{t\to r}^{\theta}(x)$, first supervises the instantaneous velocity by minimizing $\mathbb{E}\|u_{t\to t}^{\theta}(x) - u_t(x|x_1)\|^2$, $u_t(x|x_1) = \frac{x_1-x}{1-t}$, $x_1 \in \mathcal{D}$, and then extends shorter flow maps to longer ones by enforcing $\mathbb{E}\|u_{t\to t+2d}^{\theta}(x_t) - \frac{1}{2}[u_{t\to t+d}^{\theta}(x_t) + u_{t+d\to t+2d}^{\theta}(x_{t+d})]\|^2$ with $d \in [\frac{1}{128}, ..., \frac{1}{2}]$. Although effective in practice, these methods do not directly supervise long-range maps, such as $u_{t\to t+2d}^{\theta}$, with data samples from $\mathcal{D}$; instead, their long-range supervision is inherited indirectly from local dynamics, which can weaken long-horizon constraints and lead to error accumulation. In contrast, **Continuous-Equation-Based** methods, exemplified by MeanFlow, derive long-range objectives from the continuous trajectory-consistency equation in Equation 5. Specifically, MeanFlow optimizes $\mathbb{E}\|u_{t\to r}^{\theta}(x) - sg[u_t(x|x_1) + (r-t)(u_t(x|x_1)\partial_x u_{t\to r}^{\theta}(x) + \partial_t u_{t\to r}^{\theta}(x))]\|^2$, where $u_t(x|x_1) = \frac{x_1-x}{1-t}$ and $x_1 \in \mathcal{D}$. This formulation pro-

vides direct supervision for long-range flow maps, but it requires explicit derivative computation through Jacobian–vector products (JVPs), which introduces substantial computational overhead, can destabilize optimization, and narrows its practical applicability

This leads to a natural question: **Can we directly supervise long flow maps from data without relying on JVP-based continuous consistency?**

### 4.2. Euler Mean Flow

To answer this question, we propose the Euler Mean Flows (EMF) framework. Our key idea is to start from the semi-group objective in Equation 6 and reformulate this objective via a local linear approximation, which enables direct supervision from data. We also provide a rigorous theoretical justification for the validity of this approximation in Theorem 4.3 under reasonable Assumption 1 on the flow maps.

**Theorem 4.2** (Local Linear Approximation). *Let $f : X \to Y$ be a smooth mapping between finite-dimensional spaces, and let $\mathbf{x}_0 \in X$. When $\mathbf{x}$ is sufficiently close to $\mathbf{x}_0$, $f$ can be approximated by a linear function of the perturbation:*

$$f(\mathbf{x}) \approx f(\mathbf{x}_0) + Df(\mathbf{x}_0)(\mathbf{x} - \mathbf{x}_0) + o(\|x - \mathbf{x}_0\|). \quad (7)$$

*which means in the small-perturbation limit, nonlinear effects enter only at higher order, and the local behavior of $f$ is governed by its linearization.*

To reformulate the trajectory consistency objective, we follow MeanFlow (Geng et al., 2025a) and define the mean velocity field $u_{t\to r}(x) = \frac{\phi_{t\to r}(x)-x}{r-t}$. Under this definition, the trajectory consistency relation can be rewritten as:

$$(r-t)u_{t\to r}(x_t) = (s-t)u_{t\to s}(x_t) + (r-s)u_{s\to r}(x_s) \quad (8)$$

Dividing both sides by $s-t$, we obtain

$$u_{t\to s}(x_t) = (r-s)\frac{u_{t\to r}(x_t) - u_{s\to r}(x_s)}{(s-t)} + u_{t\to r}(x_t) \quad (9)$$

Unlike Shortcut and SplitMF, in our EMF we choose $s$ and $t$ to be close by setting $s = t + \Delta t$ with a small fixed step size $\Delta t$. We then apply Theorem 4.2 to obtain a local approximation of the flow maps with respect to $s$: $\phi_{t\to s}(x) \approx \phi_{t\to t}(x) + \frac{\partial \phi_{t\to s}(x)}{\partial s}|_{s=t}(s-t)$. Substituting this into the relation $\phi_{t\to s}(x) = (s-t)u_{t\to s}(x)+x$ between flows and average velocity yields $u_{t\to s}(x) \approx u_{t\to t}(x)$ when $s$ is sufficiently close to $t$. Based on this approximation, we obtain the following approximation with Equation 9:

$$u_{t\to t}(x_t) \approx (r-t-\Delta t)\frac{u_{t\to r}(x_t) - u_{t+\Delta t\to r}(x_{t+\Delta t})}{\Delta t} + u_{t\to r}(x_t)$$

$$u_{t\to r}(x_t) \approx u_{t\to t}(x_t) + (r-t-\Delta t)\frac{u_{t+\Delta t\to r}(x_{t+\Delta t}) - u_{t\to r}(x_t)}{\Delta t} \quad (10)$$

where $x_{t+\Delta t}$ is calculated as $x_{t+\Delta t} = \Delta t u_{t\to t+\Delta t}(x_t) + x_t \approx \Delta t u_{t\to t}(x_t) + x_t$. In the above derivation, the high-

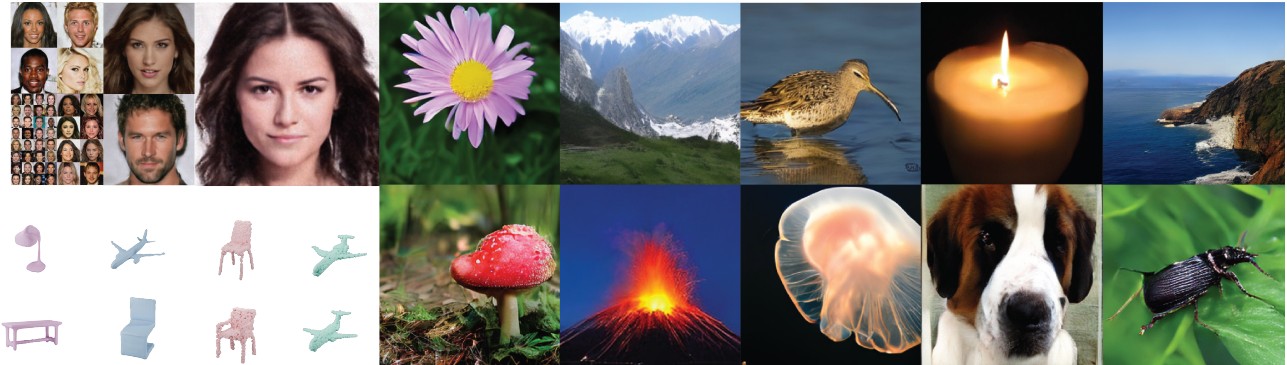

*Figure 2.* We present 1-step generation results of our EMF method for functional image generation (top left), SDF generation conditioned on 64 surface points (bottom left), unconditional point cloud generation on ShapeNet (Chang et al., 2015) (bottom right), and ImageNet (Deng et al., 2009) class-conditional generation (right).

lighted velocity field $u_{t \to t}$ is obtained using the local linear approximation in Theorem 4.2. Similar to MeanFlow, we replace $u_{t \to t}$ on the right-hand side of Equation 10 with the conditional instantaneous velocity to obtain supervision from the dataset, which leads to the following loss

$$\mathcal{L}^E(\theta) = \mathbb{E}_{t,r,x_1 \sim p_1, x \sim p_t(x|x_1), x' = sg(\Delta t u_{t \to t}^{\theta}(x)) + x}$$

$$[\|u_{t \to r}^{\theta}(x) - (u_t(x|x_1) + (r - t - \Delta t)_+ sg(\frac{u_{t+\Delta t \to r}^{\theta}(x') - u_{t \to r}^{\theta}(x)}{\Delta t}))\|^2] \quad (11)$$

Following MeanFlow, we sample a fraction of training pairs with $r = t$. With the positive clamp $(r - t - \Delta t)_+$, the proposed loss Equation 11 reduces to the Flow Matching objective $\|u_{t \to t}^{\theta}(x) - u_t(x|x_1)\|^2$ when $r = t$. This encourages $u_{t \to t}^{\theta}(x)$ to accurately learn the instantaneous velocity $u_t(x)$, which plays a crucial role in both the theoretical correctness and the stability of practical training. Under the mild assumptions introduced in Assumption 1, which are empirically verified in subsection 6.1, and using Lemma 1, we theoretically justify the validity of the loss Equation 11 through the following theorem.

**Theorem 4.3** (Surrogate Loss Validity)**.** *With $M_g =< +\infty$, $M_x < +\infty$, and $M_t < +\infty$ hold in Assumption 1, our Euler Mean Flow loss $\mathcal{L}^E(\theta)$ and the trajectory consistency loss $\mathcal{L}^C(\theta)$ satisfy*

$$D(\nabla_\theta L^E(\theta), \nabla_\theta L^C(\theta)) \leq M_g \sqrt{\mathbb{E}_{t,r,x \sim p_t(x)}[\|u_{t \to t}^{\theta}(x) - u_t(x)\|^2]}$$

$$+ (M_g M_t + M_x M_t)\Delta t + O(\Delta t^2) \quad (12)$$

*see subsection B.3 for proof.*

Theorem 4.3 shows that, provided condition $\mathbb{E}_{t,x \sim p_t(x)}[\|u_{t \to t}^{\theta}(x) - u_t(x)\|^2] \to 0$ holds during training, $\mathcal{L}^E(\theta)$ serves as a valid surrogate for $\mathcal{L}^C(\theta)$ up to $O(\Delta t)$. This condition can be promoted by local linear approximation and by mixing a fixed proportion of samples with $r = t$ in time sampling, as discussed below.

**Rationale for the Local Linear Approximation.** In Equation 10, we apply the local linear approximation in two places. First, we approximate $u_{t \to s}(x)$ in the summation

by $u_{t \to t}(x)$, enabling conditioning as $u(x|x_1)$ and introducing direct data supervision for long-range trajectory consistency. This choice reduces the objective to standard Flow Matching when $r = t$, allowing $u_{t \to t}^{\theta}(x)$ to be optimized toward $u_t(x)$ and providing the boundary condition required by Theorem 4.3. Second, in the update $x_{t+\Delta t} = x_t + \Delta t \, u_{t \to t+\Delta t}(x_t)$, we approximate $u_{t \to t+\Delta t}$ by $u_{t \to t}$. This approximation is motivated by efficiency, as $u_{t \to t}(x)$ is substantially easier to estimate under memory constraints, while using $u_{t \to t+\Delta t}$ offers no noticeable quality improvement (see Table 13).

**Comparison with Previous Methods.** To provide an intuitive comparison highlighting the key differences among related methods, we summarize them in Table 1.

*Table 1.* Comparison of flow map–based one-step methods by training strategy, JVP usage, and prediction type.

| Method | Scratch | Distill | JVP-free | $u$-pred | $x_1$-pred |
|---|---|---|---|---|---|
| MeanFlow (Geng et al., 2025a;b) | ✓ | ✓ | ✗ | ✓ | ✗ |
| AlphaFlow (Zhang et al., 2025b) | ✓ | ✓ | ✗ | ✓ | ✗ |
| ShortCut (Frans et al., 2025) | ✓ | ✓ | ✗ | ✓ | ✗ |
| SplitMF (Guo et al., 2025) | ✗ | ✓ | ✓ | ✓ | ✗ |
| Ours | ✓ | ✓ | ✓ | ✓ | ✓ |

### 4.3. $x_1$-prediction Euler Mean Flows

Whether minimizing $\mathcal{L}^E(\theta)$ in Equation 11 correctly enforces trajectory consistency depends on condition $\mathbb{E}_{t,x \sim p_t(x)}[\|u_{t \to t}^{\theta}(x) - u_t(x)\|^2] \to 0$, namely that $u_{t \to t}^{\theta}(x)$ accurately approximates the reference instantaneous velocity $u_t(x)$. However, in several high-dimensional applications, including pixel-space image generation in subsection 6.2.2 and SDF generation in subsubsection 6.2.3, $u$-prediction may fail to reliably learn $u_t(x)$, as discussed from a data-manifold perspective in (Li & He, 2025). Consequently, the velocity-based loss $\mathcal{L}^E(\theta)$, which relies on accurate local velocity estimation, may become ineffective.

To overcome this limitation, inspired by (Li & He, 2025), we adopt an $x_1$-prediction formulation and introduce the $x_1$-prediction Euler mean flow. Specifically, we define the

$x_1$-prediction mean field

$$\tilde{x}_{t\to r}(x) = (1-t)\frac{\phi_{t\to r}(x)-x}{r-t} + x \qquad (13)$$

where $\tilde{x}_{t\to r}(x)$ satisfies $\tilde{x}_{t\to r}(x) = (1-t)\,u_{t\to r}(x) + x$, which mirrors the instantaneous $x_1$-prediction flow-matching field $\tilde{x}_t(x) = (1-t)\,u_t(x) + x$. Under this formulation, the trajectory consistency relation can be rewritten as

$$\tilde{x}_{t\to r}(x_t) = \tilde{x}_{t\to s}(x_t) + (r-s)\frac{(1-t)}{(1-r)}\frac{\tilde{x}_{s\to r}(x_s)-\tilde{x}_{t\to r}(x_t)}{s-t} \qquad (14)$$

Following the $u$-prediction case, we set $s = t + \Delta t$ and use a local approximation of the flow map, giving $\tilde{x}_{t\to s} \approx \tilde{x}_{t\to t}$ for small $\Delta t$. This leads to the approximation in Equation 13

$$\tilde{x}_{t\to r}(x_t) \approx \tilde{x}_{t\to t}(x_t) + (r-t-\Delta t)\frac{(1-t)}{(1-r)}\frac{\tilde{x}_{t+\Delta t\to r}(x_{t+\Delta t})-\tilde{x}_{t\to r}(x_t)}{\Delta t} \qquad (15)$$

where $x_{t+\Delta t}$ is calculated as $x_{t+\Delta t} = \frac{\Delta t}{1-t}(\tilde{x}_{t\to s}(x_t) - x_t) + x_t \approx \frac{\Delta t}{1-t}(\tilde{x}_{t\to t}(x_t) - x_t) + x_t$. The highlighted field $\tilde{x}_{t\to t}$ is obtained using the local linear approximation for $\tilde{x}_{t\to r}$. Similar to $u-$prediction version, we replace $\tilde{x}_{t\to t}$ on the right-hand side of Equation 10 with the conditional instantaneous $\tilde{x}$ field, namely $\tilde{x}_t(x|x_1) = x_1$, to obtain supervision from the dataset, which leads to the following loss

$$\mathcal{L}^{E'}(\theta) = \mathbb{E}_{t,r,x_1\sim p_1, x\sim p_t(x|x_1), x'=sg(\Delta t\frac{\tilde{x}^\theta_{t\to t}(x)-x}{1-t})+x}[\|\tilde{x}^\theta_{t\to r}(x)-$$
$$(\tilde{x}_t(x|x_1) + (r-t-\Delta t)_+\frac{1-t}{1-r}sg(\frac{\tilde{x}^\theta_{t+\Delta t\to r}(x')-\tilde{x}^\theta_{t\to r}(x)}{\Delta t}))\|^2] \qquad (16)$$

As in the $u$-prediction setting, we sample a fraction of training pairs with $r = t$, such that Equation 16 reduces to the $x_1$-prediction flow-matching objective $\|\tilde{x}^\theta_{t\to t}(x) - \tilde{x}_t(x|x_1)\|^2$ when $r = t$. Under the Assumption 2 on $\tilde{x}^\theta$ (empirically validated in subsection 6.1), we further establish a surrogate-loss validity result analogous to that of the $u$-prediction EMF.

**Theorem 4.4** (Surrogate Loss Validity for $x_1$-Prediction). *With $M'_g < +\infty$, $M'_x < +\infty$, and $M'_t < +\infty$ hold in Assumption 2 and Lemma 2, our Euler Mean Flow loss $\mathcal{L}^{E'}(\theta)$ and the trajectory consistency loss $\mathcal{L}^{C'}(\theta)$ satisfy*

$$MSE(\nabla_\theta L^{E'}(\theta), \nabla_\theta L^{C'}(\theta)) \le M'_g\sqrt{\mathbb{E}_{t,r,x\sim p_t(x)}[\|\tilde{x}^\theta_{t\to t}(x)-\tilde{x}_t(x)\|^2]}$$
$$+ (M'_g M'_t + M'_x M'_t)\Delta t + O(\Delta t^2) \qquad (17)$$

*See subsection B.6 for proof.*

**Optimization of Time Weights.** When $r = t$, $\mathcal{L}^{E'}$ in Equation 16 reduces to the $x_1$-prediction flow-matching objective $\|\tilde{x}^\theta_{t\to t}(x)-\tilde{x}_t(x|x_1)\|^2$. As shown in Theorem 4.4, enforcing trajectory consistency further depends on how well $\tilde{x}^\theta_{t\to t}(x)$ approximates $\tilde{x}_t(x|x_1)$. However, (Li & He, 2025) demonstrate that loss $\|\tilde{x}^\theta_{t\to t}(x) - \tilde{x}_t(x|x_1)\|^2$ yields suboptimal fitting, and to mitigate this issue, (Li & He, 2025) introduces a time weight $\frac{1}{(1-t)^2}$, leading to the weighted loss

---

**Algorithm 1** Euler Mean Flow: Training
*Highlighted steps are used for conditional generation. $C$ represents the class label, and $C_0$ the corresponding unconditional label. $w$ and $k$ are parameter for CFG*

**Require:** Dataset $\mathcal{D}$, parameters $\theta$, learning rate $\eta$, noise sampler $\mathcal{N}$, time sampler $\mathcal{T}$
1: **repeat**
2:     Sample $x_1, C \sim \mathcal{D}, x_0 \sim \mathcal{N}, t, r \sim \mathcal{T}$
3:     $x_t \leftarrow (1-t)x_0 + tx_1$
4:     **if** $u$-prediction **then**
5:         $U^u \leftarrow u^\theta_{t\to t}(x_t, C_0), U^c \leftarrow u^\theta_{t\to t}(x_t, C)$
6:         $u_t(x|x_1) \leftarrow (1-w-k)U^u + kU^c + w(x_1 - x_0)$
7:         $x_{t+\Delta t} \leftarrow \Delta t U^c_t + x_t$
8:         $\mathcal{L} \leftarrow \|u^\theta_{t\to r}(x_t, C) - sg(u_t(x|x_1) + (r-t-$
        $\Delta t)_+\frac{u^\theta_{t+\Delta t\to r}(x_{t+\Delta t},C)-u^\theta_{t\to r}(x_t,C)}{\Delta t})\|^2$
    **else if** $x_1$-prediction **then**
9:         $X^u \leftarrow \tilde{x}^\theta_{t\to t}(x_t, C_0), X^c \leftarrow \tilde{x}^\theta_{t\to t}(x_t, C)$
10:     $\tilde{x}_t(x|x_1) \leftarrow (1-w-k)X^u + kX^c + wx_1$
11:     $x_{t+\Delta t} \leftarrow \Delta t\frac{X^c-x_t}{1-x_t} + x_t$
12:     $\mathcal{L} \leftarrow \|\tilde{x}^\theta_{t\to r}(x_t, C) - sg(\tilde{x}_t(x|x_1) + (r-t-$
        $\Delta t)_+\frac{1-t}{1-r}\frac{\tilde{x}^\theta_{t+\Delta t\to r}(x_{t+\Delta t})-\tilde{x}^\theta_{t\to r}(x_t)}{\Delta t})\|^2$
13:     **end if**
14:     $\theta \leftarrow \theta - \eta\nabla_\theta\mathcal{L}$
15: **until** convergence

---

$\frac{1}{(1-t)^2}\|\tilde{x}^\theta_{t\to t}(x) - \tilde{x}_t(x|x_1)\|^2$ (referred to as the $x$-pred & $u$-loss). Following this strategy, we adopt the same strategy and incorporate the time weight $\|\tilde{x}^\theta_{t\to t}(x) - \tilde{x}_t(x|x_1)\|^2$ into $\mathcal{L}^{E'}$ in Equation 16 to improve the learning of $\tilde{x}^\theta_{t\to t}(x)$. For numerical stability, we clamp the denominators $1 - t$ and $1 - r$ to a minimum value of 0.02.

## 4.4. Algorithm

Building on the above discussion, we derive the training and sampling procedures of Euler Mean Flows for both conditional and unconditional generation, as summarized in Algorithms 1 and 2. For conditional generation, following (Geng et al., 2025a), we adopt classifier-free guidance (CFG) during training, with an effective guidance scale given by $w' = \frac{w}{1-k}$, where $w$ and $k$ denote the CFG coefficients. Additional details on CFG, adaptive loss weighting, and time sampling strategies are provided in subsection C.1.

## 5. JVP-Free Training

### 5.1. Training Speed and Memory Efficiency

Here, we further analyze the memory and computational cost of our training algorithm in Algorithm 1. The comparison is reported in Table 2, where we compare against MeanFlow (Geng et al., 2025a), AlphaFlow (Zhang et al., 2025b), SplitMeanFlow (Guo et al., 2025), ShortCut (Frans et al., 2025), ESD, LSD, and the PSD(U/M) variants of Flow

Map Matching (Boffi et al., 2025).

For conditional generation, our training requires three stop-gradient forward passes $u_{t \to t}^{\theta}(x, C)$, $u_{t \to r}^{\theta}(x, C_0)$ and $u_{t+\Delta t \to r}^{\theta}(x_{t+\Delta t}, C)$ and one optimized forward pass $u_{t \to r}^{\theta}(x, C)$, while MeanFlow (Geng et al., 2025a) requires two stop-gradient forward passes $u_{t \to t}^{\theta}(x, C)$, $u_{t \to r}^{\theta}(x, C_0)$, one JVP computation, and one optimized forward pass $u_{t \to r}^{\theta}(x, C)$. Although the latter two are jointly computed via `torch.jvp` in PyTorch, the JVP operation introduces non-negligible overhead. Compared to MeanFlow, our method replaces one JVP computation with a stop-gradient forward pass, resulting in lower memory and runtime costs. Moreover, by avoiding JVPs, our method is compatible with FlashAttention, whereas MeanFlow does not support FlashAttention due to its reliance on JVP.

For unconditional generation, our method requires two stop-gradient forward passes $u_{t \to t}^{\theta}(x)$ and $u_{t+\Delta t \to r}^{\theta}(x_{t+\Delta t})$ and one optimized forward pass $u_{t \to r}^{\theta}(x)$, whereas MeanFlow only requires one JVP and one optimized forward pass $u_{t \to r}^{\theta}(x)$. Although our approach remains more efficient, the efficiency gap becomes smaller. To further reduce the cost,

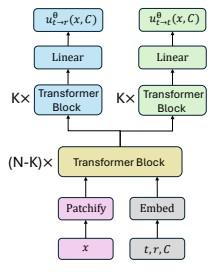

*Figure 3.* Auxiliary Branch for $u_{t \to t}^{\theta}(x)$

we adopt the strategy of (Geng et al., 2025b) by introducing a lightweight auxiliary branch to predict $u_{t \to t}^{\theta}(x)$, while the main branch predicts $u_{t \to r}^{\theta}(x)$. The auxiliary and main branches share forward computations, and an additional loss $\mathcal{L}^F$ is used to improve the approximation of $u_{t \to t}^{\theta}$. The final loss is given as $\mu_1 \mathcal{L}^{EMF} + \mu_2 \mathcal{L}^F$, with hyperparameters $\mu_1$ and $\mu_2$, where we set $\mu_1 = 1$ and $\mu_2 = 1$ in practice. With this design, training only requires one stop-gradient forward pass and one optimized forward pass, leading to substantially reduced memory and computational cost.

### 5.2. Discussion of $\Delta t$

Unlike ShortCut (Frans et al., 2025), SplitMeanFlow (Guo et al., 2025), and AlphaFlow (Zhang et al., 2025b), which require a schedule over interval lengths, EMF uses a fixed $\Delta t = 10^{-3}$ for the local linearization on the short interval $[t, s]$ in all experiments. We study the sensitivity to this choice in Table 4. EMF remains stable for a broad range of $\Delta t$, from $6 \times 10^{-4}$ to $10^{-2}$. Very small values of $\Delta t$ lead to noisy finite differences due to finite-precision effects, whereas overly large values make the local approximation inaccurate. This ablation shows that EMF is not sensitive to the precise choice of $\Delta t$ within a reasonable range.

*Table 2.* Comparison of memory and computational cost between our method and MeanFlow for unconditional (CelebA-HQ) and conditional (ImageNet-1000) generation using the DiT-B/2 model. "Peak" denotes the maximum GPU memory usage during training, "Fixed" refers to the constant memory overhead, and aux-EMF indicates our method with a 4-block auxiliary head. All experiments are conducted on a single H200 GPU with batch sizes of 64 for CelebA-HQ and 128 for ImageNet-1000, using EMA and AdamW optimization with mixed-precision (FP16) training in PyTorch.

| Method | Peak | Fixed | Speed / Iter | FID |
|---|---|---|---|---|
| **CelebA-HQ** | | | | |
| MeanFlow / ESD | 32.1GB | 2.3GB | 151.4ms | 12.4 |
| EMF (Ours) | 23.3GB | 2.3GB | 91.2ms | **10.9** |
| aux-EMF (Ours) | **17.6GB** | 2.8GB | **84.2ms** | 11.7 |
| AlphaFlow | 32.6GB | 2.3GB | 127.7ms | 11.6 |
| SplitMeanFlow / PSD(U) | 23.8GB | 2.3GB | 94.6ms | 15.7 |
| ShortCut / PSD(M) | 23.8GB | 2.3GB | 94.6ms | 20.5 |
| LSD | 52.2GB | 2.3GB | 426.7ms | 12.3 |
| **ImageNet** | | | | |
| MeanFlow | 101.9GB | 2.4GB | 400.9ms | 11.1 |
| EMF (Ours) | **57.9GB** | 2.4GB | **198.8ms** | **7.2** |

*Table 4.* Ablation on the short-step size $\Delta t$ on CelebA-HQ. We use a fixed $\Delta t$ in all experiments. Lower FID is better.

| $\Delta t$ | 1e−4 | 3e−4 | 6e−4 | 1e−3 | 2e−3 | 3e−3 | 1e−2 | 3e−2 | 1e−1 |
|---|---|---|---|---|---|---|---|---|---|
| FID | 25.8 | 12.7 | 10.9 | 10.9 | 10.7 | 10.8 | 10.7 | 12.2 | 25.2 |

### 5.3. Optimization Stability

The original MeanFlow framework often exhibits anomalous loss escalation during training. As shown in Figure 4, the training loss of Mean-Flow tends to increase abnormally as optimization progresses, even when

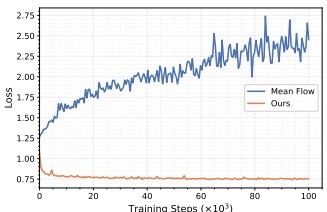

*Figure 4.* Training loss comparison between Euler Mean Flow and Mean Flow.

adaptive loss weighting is applied for stabilization, resulting in high variance and unstable dynamics. In contrast, our method achieves steadily decreasing loss with well-controlled variance, even without adaptive weighting. Moreover, we observe that MeanFlow is prone to training collapse in image generation tasks, including both latent-space (Figure 18) and pixel-space (Figure 21) settings, especially under mixed-precision training, whereas our approach remains robust. As a result of its improved stability, our method consistently outperforms MeanFlow on both image generation Table 3 and SDF generation Table 7 tasks.

### 5.4. Broader Applications

Many sparse computation libraries, such as PVCNN and TorchSparse, do not support JVP operations, limiting the applicability of MeanFlow in these domains. In contrast, EMF is fully JVP-free and achieves strong performance on functional and point cloud generation tasks, while enabling

*Table 3.* Comparison of training objectives under equivalent architecture (DiT-B) and compute. FID-50k scores (lower is better) are shown over 128-, 4-, and 1-step denoising.

| Method | CelebA-HQ-256 (unconditioned) | | | ImageNet-256 (class conditioned) | | |
| --- | --- | --- | --- | --- | --- | --- |
| | 128-Step | 4-Step | 1-Step | 128-Step | 4-Step | 1-Step |
| Diffusion (Song et al., 2021a) | 23.0 | 123.4 | 132.2 | 39.7 | 464.5 | 467.2 |
| FM (Lipman et al., 2023) | 7.3 | 63.3 | 280.5 | 17.3 | 108.2 | 324.8 |
| PD (Salimans & Ho, 2022) | 302.9 | 251.3 | 14.8 | 201.9 | 142.5 | 35.6 |
| CD (Song et al., 2023) | 59.5 | 39.6 | 38.2 | 132.8 | 98.01 | 136.5 |
| Reflow (Liu et al., 2023) | 16.1 | 18.4 | 23.2 | 16.9 | 32.8 | 44.8 |
| CM (Song et al., 2023) | 53.7 | 19.0 | 33.2 | 42.8 | 43.0 | 69.7 |
| ShortCut (Frans et al., 2025) | 6.9 | 13.8 | 20.5 | 15.5 | 28.3 | 40.3 |
| MF (Geng et al., 2025a) | 7.8 | 8.9 | 12.4 | 6.4 | 7.1 | 11.1 |
| EMF (Ours) | 7.6 | 8.4 | 10.8 | 5.6 | 6.9 | 7.2 |

efficient one-step and few-step generation in sparse settings (subsection 6.2.4, subsection 6.2.5).

# 6. Experiment

## 6.1. Validation and Ablation

Our theorems in Theorem 4.3 and Theorem 4.4 rely on Assumptions Assumption 1 and Assumption 2, respectively. To validate these assumptions, we train a DiT-B/2 model on CelebA-HQ dataset and monitor the values of $M_g$ ($M'_g$), $M_x$ ($M'_x$), and $M_t$ ($M'_t$) throughout training. The training protocol, model architecture, and hyperparameters follow subsection 6.2.1. To estimate the spectral norms in $M_g(M'_g)$ and $M_x(M'_x)$, for any matrix $M$, we randomly sample $n_1$ unit vectors $|v| = 1$ and approximate $\|M\|_2$ by $\max \|Mv\|$. For expectations of the form $\mathbb{E}_{t,r,x\sim p_t(x)}$, we instead evaluate $\mathbb{E}_{t,r,x\sim p_t(x|x_1),x_1\sim p\text{data}}$. We sample $n_2$ points from $t, r \sim \mathcal{T}$, draw $x_1 \sim p_\text{data}$ and $x \sim p_t(x|x_1)$, and estimate the expectations via Monte Carlo averaging. Results are reported in Figure 19 and Figure 20. Additional experimental details on memory and timing statistics are provided in Appendix D.

We further separate the roles of removing JVPs and introducing the local-linearization surrogate. Removing JVPs alone mainly improves training stability, as reflected by the loss curves in Figure 4, and efficiency, yielding approximately 50% gains in both memory usage and training speed in our setting. However, without the local-linearization surrogate, the long flow map loses direct data supervision and achieves the worst final performance. In contrast, the local-linearization surrogate restores direct data supervision in the JVP-free setting. Combining these two components gives EMF, which achieves both improved stability and stronger final performance, as shown in Table 5.

*Table 5.* Ablation of removing JVPs and introducing the local-linearization surrogate on CelebA-HQ.

| Method | EMF | w/o removing JVP | w/o Local Linearization |
| --- | --- | --- | --- |
| FID | **10.9** | 12.4 | 13.0 |

## 6.2. Applications

### 6.2.1. LATENT SPACE IMAGE GENERATION

We evaluate our method on latent-space image generation tasks using two datasets: ImageNet-1000 (Deng et al., 2009) and CelebA-HQ (Liu et al., 2015), both resized to a resolution of $256 \times 256$. Following the latent-space generation paradigm, we adopt a DiT-B/2 backbone (Peebles & Xie, 2023) together with a standard pre-trained VAE from Stable Diffusion (Rombach et al., 2022b), which maps a $256 \times 256 \times 3$ image into a compact latent representation of size $32 \times 32 \times 4$. For training efficiency, we employ mixed-precision training with FP16, in contrast to the FP32 training used in (Geng et al., 2025a). Our method consistently outperforms existing approaches on both ImageNet-1000 and CelebA-HQ (Table 3). Moreover, as reflected in the training dynamics compared with MeanFlow (Figure 4), our method exhibits improved optimization stability.

### 6.2.2. PIXEL SPACE IMAGE GENERATION

For pixel-space image generation, we adopt the JiT framework following (Li & He, 2025). JiT is a plain Vision Transformer that directly processes images as sequences of pixel patches, without relying on VAEs or other latent representations. To accommodate the high dimensionality of pixel-space generation, JiT employs relatively large patch sizes. We build our model upon JiT-B/16 and train it on the CelebA-HQ dataset at a resolution of $256 \times 256$. In the one-step generation setting, we observe behavior consistent with prior findings on JiT: the $u$-prediction variant of EMF produces images with significant noise and poor visual quality Figure 7. This further highlights the necessity of the $x_1$-prediction variant. A comprehensive comparison

*Table 6.* Comparison of pixel-space generative methods under equivalent architectures (DiT-B) and computational budgets on CelebA-HQ-256. FID-50k scores (lower is better) are reported for 2-step and 1-step denoising, while FID-10k scores are used for the 128-step setting. The "Variant" column follows the format prediction/loss; e.g., $x_1/u$ denotes $x_1$-prediction trained with a loss defined in the $u$-space.

| Method | Variant | 128-Step | 2-Step | 1-Step |
|---|---|---|---|---|
| JiT (Li & He, 2025) | $u/u$ | 339.7 | 384.4 | 407.0 |
| JiT (Li & He, 2025) | $x_1/x_1$ | 42.3 | 429.3 | 399.4 |
| JiT (Li & He, 2025) | $x_1/u$ | 27.9 | 441.6 | 440.1 |
| MeanFlow (Li & He, 2025) | $u/u$ | 321.3 | 323.9 | 327.0 |
| MeanFlow (Li & He, 2025) | $x_1/x_1$ | 42.2 | 41.5 | 56.8 |
| MeanFlow (Li & He, 2025) | $x_1/u$ | 24.0 | 26.6 | 39.2 |
| EMF (Ours) | $u/u$ | 329.4 | 323.3 | 324.6 |
| EMF (Ours) | $x_1/x_1$ | 35.8 | 34.8 | 36.3 |
| EMF (Ours) | $x_1/u$ | **21.4** | **26.4** | **30.6** |

is provided in Table 6. Moreover, the training dynamics in Figure 18 show that our method achieves substantially improved stability compared to MeanFlow.

### 6.2.3. SDF GENERATION

Next, we evaluate our method on SDF generation. We adopt the Functional Diffusion framework (Zhang & Wonka, 2024), in which the model is conditioned on a sparse set of observed surface points (64 points) and generates the complete SDF function from noise using an attention-based architecture. Experiments are conducted on the ShapeNet-CoreV2 dataset (Chang et al., 2015) and evaluated using Chamfer Distance, F-score, and Boundary Loss, which measure surface accuracy and boundary fidelity (see subsection C.5 for details). As shown in Table 3, our method outperforms MeanFlow and achieves performance comparable to multi-step generation. We also apply the same framework to a 2D MNIST-based SDF generation task (Figure 23), where handwritten digits are converted into SDFs. In this case, the $u$-prediction variant of EMF suffers from attention variance collapse during training, whereas only the $x_1$-prediction successfully generates high-quality shapes.

### 6.2.4. POINT CLOUD GENERATION

To demonstrate the applicability of our method to sparse and irregular domains, we apply EMF to point cloud generation. We adopt the Latent Point Diffusion Model (LION) architecture (Vahdat et al., 2022), which builds on a VAE that encodes each shape into a hierarchical latent representation comprising a global shape latent and a point-structured latent point cloud. We use pre-trained encoders and decoders based on Point-Voxel CNNs (PVCNNs) and fine-tune both the global and point cloud latents using EMF on the airplane and chair categories. Training and model details are provided in subsection C.6. For evaluation, we compare generated samples against reference sets using Coverage (COV) and 1-Nearest Neighbor Accuracy (1-NNA), com-

*Table 7.* Quantitative comparison of reconstruction quality. The model is trained on the ShapeNet dataset, where the conditional input consists of 64 points sampled from the target surface. The model is required to reconstruct the surface based on these 64 points. Step denotes the number of inference steps.

| Method | Step | Chamfer ↓ | F-Score ↑ | Boundary ↓ |
|---|---|---|---|---|
| 3DS2VS (Zhang et al., 2023) | 18 | 0.144 | 0.608 | 0.016 |
| FD (Zhang & Wonka, 2024) | 64 | 0.101 | 0.707 | 0.012 |
| MF (Li et al., 2025) | 1 | 0.060 | 0.584 | 0.011 |
| EMF (Ours) | 1 | 0.046 | 0.674 | 0.011 |

(↓ lower is better; ↑ higher is better.)

puted with either Chamfer Distance or Earth Mover's Distance, to assess sample diversity and distributional alignment. As shown in Figure 4, our method achieves competitive performance among one-step generation approaches.

### 6.2.5. FUNCTION-BASED IMAGE GENERATION

We further include a sparse functional image generation experiment to examine whether our method generalizes beyond dense grid-based generation. Using the Infty-Diff (Bond-Taylor & Willcocks, 2024) architecture, we train on randomly sampled pixel coordinates with 25% observed pixels on FFHQ (Karras et al., 2019) and CelebA-HQ at $256 \times 256$ resolution. Details of the functional representation, architecture, and multi-resolution evaluation are deferred to subsection C.4. As shown in Figure 10, our method remains competitive in one-step generation.

## 7. Conclusion

We proposed EMF as a trajectory-consistent framework for efficient one-step and few-step generation, enabling direct data supervision of long-range flow maps via a local linear approximation of the semigroup objective. EMF avoids explicit derivative computation through a unified, JVP-free training scheme with theoretical guarantees, and extending it to broader tasks, larger models, and more general theoretical settings is an important direction for future work.

EMF has several limitations. The local-linearization surrogate may be less accurate for large intervals or rapidly varying dynamics, and its performance can depend on the prediction parameterization and data domain. Our scaling experiments are limited by available compute, although additional 3D results suggest that the JVP-free formulation remains stable across tasks. Scaling EMF to larger models and broader trajectory settings remains future work.

### Acknowledgements

We sincerely thank the anonymous reviewers for their valuable feedback. Georgia Tech authors acknowledge NSF CAREER #2420319, IIS #2433307, OISE #2433313, IIS #2433322, ECCS #2318814, and CNS #2450401 for funding support, and the NVIDIA Academic Grant Program for hardware support.

## Impact Statement

This paper presents work whose goal is to advance the field of machine learning. There are many potential societal consequences of our work, none of which we feel must be specifically highlighted here.

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

## A. Design Philosophy of the Euler MeanFlow Loss

Here we provide an overview of the design rationale of Euler MeanFlow, explaining the principles behind the Euler MeanFlow losses. The loss design of Euler MeanFlow follows the same fundamental logic as Flow Matching: at their core, both aim to learn a direct target. For example, Flow Matching optimizes $\mathcal{L}^{FM}(\theta) = \mathbb{E}_{t,x\sim p_t}\|u_t^\theta(x) - u_t(x)\|_2^2$, where $u_t(x)$ is the reference velocity field. Since $u_t(x)$ is not directly accessible from data, Flow Matching introduces a conditional distribution $p_t(x|x_1)$ and a conditional velocity $u(x|x_1)$ on $x_1 \in \mathcal{D}$, yielding the conditional loss $\mathcal{L}_c^{FM}(\theta) = \mathbb{E}_{t,x\sim p_t(\cdot|x_1),x_1\sim p_1}\|u_t^\theta(x) - u_t(x|x_1)\|_2^2$, which is shown that $\nabla\mathcal{L}_c^{FM}(\theta) = \nabla\mathcal{L}^{FM}(\theta)$. Because $u(x|x_1) = \frac{x-x_1}{1-t}$, with $x$ sampled from the tractable conditional distribution $p(x|x_1)$, is directly computable from dataset $\mathcal{D}$, the conditional loss $\mathcal{L}_c^{FM}(\theta)$ can be used to train a model targeting the original objective $\mathcal{L}^{FM}(\theta)$.

Euler MeanFlow is built on the same principle. Its ideal learning objective is $\mathcal{L}^C(\theta) = \mathbb{E}_{t,s,r,x_t=(1-t)x_0+tx_1,x_1\sim p_{data},x_0\sim p_0}\frac{1}{w(t,r)}\|\phi_{t\to r}^\theta(x_t) - \phi_{s\to r}^\theta(\phi_{t\to s}^\theta(x))\|_2^2$ in Equation 6. Similar to Flow Matching, this direct objective does not explicitly leverage information from the training data. A straightforward solution is to impose supervision only at boundary conditions and propagate it outward, as in previous works (Guo et al., 2025; Frans et al., 2025). However, such boundary-based supervision remains sparse and indirect, which is insufficient to constrain long-range dynamics and often leads to unstable training and degraded performance. Therefore, our goal is to design a training objective that provides dense, data-driven supervision for $u_{t\to r}^\theta(x)$ while avoiding reliance on boundary constraints.

The central difficulty is that, unlike instantaneous velocity fields, long-range velocity fields do not admit a natural conditional form (Theorem 4.1), making it unclear how to incorporate dataset supervision. To overcome this challenge, we propose a two-step strategy.

1. First, we observe that $\mathcal{L}^C(\theta)$ involves three time segments $t \to s$, $s \to r$ and $t \to r$. We select one segment $t \to s$ to be sufficiently short and apply a local linear approximation (Theorem 4.2) on this interval. This transforms part of the long-range transport into an instantaneous velocity field, which admits a well-defined conditional counterpart $u_{t\to t}(x|x_1)$. As a result, we obtain an intermediate surrogate objective $\mathcal{L}^{\tilde{C}}(\theta)$ in Equation 20 that partially connects long-range dynamics with locally defined velocities.

2. Second, since $\mathcal{L}^{\tilde{C}}(\theta)$ now involves instantaneous velocity fields, we can follow the Flow Matching framework and replace them with conditional instantaneous velocity fields. This step injects explicit dataset supervision into the objective and yields the final loss $\mathcal{L}^{EMF}(\theta)$.

In Lemma 1 and Theorem 4.3, we theoretically justify this construction by showing that $\nabla_\theta\mathcal{L}^{\tilde{C}}(\theta) \approx \nabla_\theta\mathcal{L}^{EMF}(\theta)$ and $\nabla_\theta\mathcal{L}^{\tilde{C}}(\theta) \approx \nabla_\theta\mathcal{L}^C(\theta)$. These results indicate that optimizing $\mathcal{L}^{EMF}$ provides a faithful approximation to the ideal objective $\mathcal{L}^C$, while simultaneously incorporating explicit dataset supervision for learning long-range dynamics.

The $x_1$-prediction variant follows the same strategy. It is worth noting that, although two time variables $t$ and $r$ are involved, only the quantities at time $t$ are generated through sampling. Consequently, only variables at time $t$ can be naturally conditioned on observed data.

## B. Missing Proofs and Derivations

Here, we introduce the assumptions used in our theoretical analysis, which are empirically validated in subsection 6.1.

**Assumption 1** (Assumption of $u_{t\to r}^\theta$). We assume that $u_{t\to r}^\theta$ is differentiable with respect to its parameters and satisfies the following regularity conditions: (1) $M_g = \sqrt{\mathbb{E}_{t,r,x\sim p_t(x)}[\frac{1}{m}\|\nabla_\theta u_{t\to r}^\theta(x)\|_2^2]} < +\infty$,(2) $M_x = \sqrt{\mathbb{E}_{t,r,x\sim p_t(x)}[\frac{1}{m}\|\partial_x u_{t+\Delta t\to r}^\theta(x')\|_2^2]} < +\infty$, (3) $M_t = \sqrt{\mathbb{E}_{t,r,x\sim p_t(x)}[\|\partial_s u_{t\to s}^\theta|_{s=t}\|^2]} < +\infty$ where $x' = x + \Delta t u_{t\to t}^\theta(x)$, $m$ is the model size and and $\|\cdot\|_2$ denotes the matrix 2-norm.

**Assumption 2** (Assumption of $\tilde{x}_{t\to r}^\theta$). We assume that $\tilde{x}_{t\to r}^\theta$ is differentiable with respect to its parameters and satisfies the following regularity conditions: (1) $M_g' = \sqrt{\mathbb{E}_{t,r,x\sim p_t(x)}[\frac{1}{m}\|\nabla_\theta\tilde{x}_{t\to r}^\theta(x)\|_2^2]} < +\infty$, (2) $M_x' = \frac{1}{1-r}\sqrt{\mathbb{E}_{t,r,x\sim p_t(x)}[\frac{1}{m}\|\partial_x\tilde{x}_{t+\Delta t\to r}^\theta(x')\|_2^2]} < +\infty$, (3) $M_t' = \sqrt{\mathbb{E}_{t,r,x\sim p_t(x)}[\|\partial_s\tilde{x}_{t\to s}^\theta|_{s=t}\|^2]} < +\infty$, where $x' = x + \Delta t\frac{\tilde{x}_{t\to t}^\theta(x)-x}{1-t}$, $m$ is the model size and and $\|\cdot\|_2$ denotes the matrix 2-norm (spectral norm).

## B.1. Proof of Theorem 4.1

**Theorem 4.1** (Non-existence of conditional flow maps) There exists no conditional flow maps $\phi_{t\rightarrow r}(x|x_{t_1})$ that simultaneously (i) is consistent with the conditional velocity $u(x|x_1)$ under Equation 3, and (ii) satisfies the consistency relation $\phi_{t\rightarrow r}(x) = \mathbb{E}_{x_1\sim p_t(x_1|x)}[\phi_{t\rightarrow r}(x|x_1)]$ with marginal flow maps. As a result, a self-consistent conditional cumulative field does not exist.

*Proof.* First, we denote the mappings $\phi_{t\rightarrow r}(x)$ obtained from (1) and (2) as $\phi_{t\rightarrow r}^{(1)}(x)$ and $\phi_{t\rightarrow r}^{(2)}(x)$, respectively. Specifically, $\phi_{t\rightarrow r}^{(1)}(x) = \phi_r(\phi_t^{-1}(x))$, and $\phi_{t\rightarrow r}^{(2)}(x) = \mathbb{E}_{x_1\sim p_t(x_1|x)}[\phi_{t\rightarrow r}(x|x_1)]$. It suffices to show that $\phi_{t\rightarrow r}^{(1)}(x) \neq \phi_{t\rightarrow r}^{(2)}(x)$. To this end, it is sufficient to prove that $\frac{d}{dt}\phi_{t\rightarrow r}^{(1)}(x) \neq \frac{d}{dt}\phi_{t\rightarrow r}^{(2)}(x)$ at $t = 0$.

$$
\begin{aligned}
\frac{d}{dr}\phi_{t\rightarrow r}^{(2)}(x) &= \frac{d}{dr}\int_{x_1}\phi_{t\rightarrow r}(x|x_1)p_t(x_1|x)dx_1 \\
&= \int_{x_1}\frac{d}{dr}\phi_{t\rightarrow r}(x|x_1)p_t(x_1|x)dx_1 \\
&= \int_{x_1} u_r(\phi_{t\rightarrow r}(x|x_1)|x_1)\frac{p_{data}(x_1)p_t(x|x_1)}{p_t(x)}dx_1 \\
&= \int_{x_1} u_r(\phi_{t\rightarrow r}(x|x_1)|x_1)p_{data}(x_1)dx_1 \\
&= \int_{x_1} (x_1 - x)p_{data}(x_1)dx_1 \\
&= \mathbb{E}_{x_1\sim p_{data}(x_1)}[x_1] - x
\end{aligned}
\tag{18}
$$

Consequently, if $\phi_{t\rightarrow r}^{(1)}(x) = \phi_{t\rightarrow r}^{(2)}(x)$ we must have $\frac{d}{dr}\phi_r(x) = \mathbb{E}_{x_1\sim p_{data}(x_1)}[x_1] - x$, $\phi_r(x) = (\mathbb{E}_{x_1\sim p_{data}(x_1)}[x_1] - x)r + x$, which implies $p_{data} = (\phi_1)_\sharp p_0 = \delta_{\mathbb{E}_{x_1\sim p_{data}(x_1)}[x_1]}$, where $\delta$ denotes the Dirac distribution at a single point. $\square$

## B.2. Lemma 1 and its Proof

**Lemma 1.** *With $M_g = \sqrt{\mathbb{E}_{t,r,x\sim p_t(x)}[\frac{1}{m}\|\nabla_\theta u_{t\rightarrow r}^\theta(x)\|_2^2]} < +\infty$ holds in Assumption 1, our Euler Mean Flow loss $\mathcal{L}^E(\theta)$ and the approximated trajectory consistency loss $\mathcal{L}^{\tilde{C}}(\theta)$ satisfy*

$$
MSE(\nabla\mathcal{L}^E(\theta), \nabla\mathcal{L}^{\tilde{C}}(\theta)) \leq M_g\sqrt{\mathbb{E}_{t,r,x\sim p_t(x)}[\|u_{t\rightarrow t}^\theta(x) - u_t(x)\|^2]}
\tag{19}
$$

*where* MSE *denotes the mean squared error. Consequently, during training, if $\|u_{t\rightarrow t}^\theta(x) - u_t(x)\|^2 \rightarrow 0$, then $\mathcal{L}^E(\theta)$ and $\mathcal{L}^{\tilde{C}}(\theta)$ share the same optimal target at $\theta$. The term $u_t(x)$ denotes the reference velocity at $x$, defined as $u_t(x) = \mathbb{E}_{x_1\sim p(x_1|x)}[u_t(x|x_1)]$, which is intractable to compute analytically.*

*Here, the approximated trajectory consistency loss $\mathcal{L}^{\tilde{C}}(\theta)$ are defined as*

$$
\begin{aligned}
\mathcal{L}^{\tilde{C}}(\theta) = \mathbb{E}_{t,r,x\sim p_t(x),x'=sg(\Delta t u_{t\rightarrow t}^\theta(x))+x} \\
[\|u_{t\rightarrow r}^\theta(x) - (u_{t\rightarrow t}^\theta(x) + (r - t - \Delta t) \\
sg(\frac{u_{t+\Delta t\rightarrow r}^\theta(x') - u_{t\rightarrow r}^\theta(x)}{\Delta t}))\|^2]
\end{aligned}
\tag{20}
$$

*It is straightforward to verify that the loss $\mathcal{L}^{\tilde{C}}(\theta)$ is the mean-velocity formulation of $\mathcal{L}^C(\theta)$ under the local linear approximation in Equation 10, expressed via $u_{t\rightarrow r}(x) = \frac{\phi_{t\rightarrow r}(x) - x}{r - t}$, and differs by a temporal scaling factor $1/\Delta t$.*

The above lemma links the surrogate Euler Mean Flow loss $\mathcal{L}^E(\theta)$ to the approximated trajectory consistency loss $\mathcal{L}^{\tilde{C}}(\theta)$. Building on this result, we can further relate $\mathcal{L}^E(\theta)$ to the original trajectory consistency objective $\mathcal{L}^C(\theta)$ thereby showing that $\mathcal{L}^E(\theta)$ serves as a valid surrogate for the trajectory consistency objective.

*Proof.* We first define the reference regression loss $\mathcal{L}^R(\theta)$ as

$$
\begin{aligned}
\mathcal{L}^R(\theta) = & \mathbb{E}_{t,r,x\sim p_t(x),x'=sg(\Delta t u_{t\to t}^\theta(x))+x} \\
& [\|u_{t\to r}^\theta(x) - (u_t(x) + (r - t - \Delta t) \\
& sg(\frac{u_{t+\Delta t\to r}^\theta(x') - u_{t\to r}^\theta(x)}{\Delta t}))\|^2]
\end{aligned}
\tag{21}
$$

Let $B(\theta; r, t, x) = (r - t - \Delta t)sg(\frac{u_{t+\Delta t\to r}^\theta(x') - u_{t\to r}^\theta(x)}{\Delta t})$. Since $B(\theta; r, t, x)$ contains the stop-gradient operator $sg(\cdot)$, it satisfies $\nabla_\theta B(\theta; r, t, x) = 0$. Using $B(\theta; r, t, x)$, the Euler Mean Flow loss $\mathcal{L}^E(\theta)$, the reference regression loss $\mathcal{L}^R(\theta)$, and the approximated trajectory consistency loss $\mathcal{L}^{\tilde{C}}(\theta)$ can be written as

$$
\begin{aligned}
\mathcal{L}^E(\theta) &= \mathbb{E}_{t,r,x_1\sim p_{data},x\sim p_t(x|x_1)}[\|u_{t\to r}^\theta(x) - (u_t(x|x_1) + B(\theta; r, t, x))\|^2] \\
\mathcal{L}^R(\theta) &= \mathbb{E}_{t,r,x\sim p_t(x)}[\|u_{t\to r}^\theta(x) - (u_t(x) + B(\theta; r, t, x))\|^2] \\
\mathcal{L}^{\tilde{C}}(\theta) &= \mathbb{E}_{t,r,x\sim p_t(x)}[\|u_{t\to r}^\theta(x) - (u_{t\to t}^\theta(x) + B(\theta; r, t, x))\|^2]
\end{aligned}
\tag{22}
$$

We first show that the Euler Mean Flow loss $\mathcal{L}^E(\theta)$ and the reference regression loss $\mathcal{L}^R(\theta)$ satisfy $\nabla_\theta \mathcal{L}^E(\theta) = \nabla_\theta \mathcal{L}^R(\theta)$. Expanding $\nabla \mathcal{L}^E(\theta)$, we obtain

$$
\begin{aligned}
\nabla_\theta \mathcal{L}^E(\theta) &= \mathbb{E}_{t,r,x_1\sim p_{data},x\sim p_t(x|x_1),}[\nabla\|u_{t\to r}^\theta(x) - (u_t(x|x_1) + B(\theta; r, t, x))\|^2] \\
&\overset{\nabla B(\theta;r,t,x)=0}{=} \mathbb{E}_{t,r,x_1\sim p_{data},x\sim p_t(x|x_1)}[\nabla u_{t\to r}^\theta(x)(u_{t\to r}^\theta(x) - u_t(x|x_1) - B(\theta; r, t, x))] \\
&\overset{\nabla B(\theta;r,t,x)=0}{=} \mathbb{E}_{t,r,x_1\sim p_{data},x\sim p_t(x|x_1)}[\nabla u_{t\to r}^\theta(x)(u_{t\to r}^\theta(x) - u_t(x|x_1) - B(\theta; r, t, x))]
\end{aligned}
\tag{23}
$$

where $\mathbb{E}_{t,r,x_1\sim p_{data},x\sim p_t(x|x_1)}[\nabla u_{t\to r}^\theta(x)(u_{t\to r}^\theta(x) - B(\theta; r, t, x))]$ can be computed as:

$$
\begin{aligned}
&\mathbb{E}_{t,r,x_1\sim p_{data},x\sim p_t(x|x_1)}[\nabla u_{t\to r}^\theta(x)(u_{t\to r}^\theta(x) - B(\theta; r, t, x))] \\
&= \int_{t,r}\int_{x_1}\int_x (\nabla u_{t\to r}^\theta(x)(u_{t\to r}^\theta(x) - B(\theta; r, t, x)))p(x|x_1)p_{data}(x_1)p(t, r)dxdx_1dtdr \\
&= \int_{t,r}\int_x (\nabla u_{t\to r}^\theta(x)(u_{t\to r}^\theta(x) - B(\theta; r, t, x)))\int_{x_1} p(x|x_1)p_{data}(x_1)dx_1 p(t, r)dxdtdr \\
&= \int_{t,r}\int_x (\nabla u_{t\to r}^\theta(x)(u_{t\to r}^\theta(x) - B(\theta; r, t, x)))p(x)p(t, r)dxdtdr \\
&= \mathbb{E}_{t,r,x\sim p_t(x)}[\nabla u_{t\to r}^\theta(x)(u_{t\to r}^\theta(x) - B(\theta; r, t, x))]
\end{aligned}
\tag{24}
$$

And $\mathbb{E}_{t,r,x_1\sim p_{data},x\sim p_t(x|x_1)}[\nabla u_{t\to r}^\theta(x)u_t(x|x_1)]$ can be calculated as

$$
\begin{aligned}
&\mathbb{E}_{t,r,x_1\sim p_{data},x\sim p_t(x|x_1)}[\nabla u_{t\to r}^\theta(x)u_t(x|x_1)] \\
&= \int_{t,r}\int_{x_1}\int_x \nabla u_{t\to r}^\theta(x)u_t(x|x_1)p(x|x_1)p_{data}(x_1)p(t, r)dxdx_1dtdr \\
&= \int_{t,r}\int_x \nabla u_{t\to r}^\theta(x)(\int_{x_1} u_t(x|x_1)p(x|x_1)p_{data}(x_1)dx_1)p(t, r)dxdtdr \\
&= \int_{t,r}\int_x \nabla u_{t\to r}^\theta(x)(\int_{x_1} u_t(x|x_1)p(x_1|x)p(x)dx_1)p(t, r)dxdtdr \\
&= \int_{t,r}\int_x \nabla u_{t\to r}^\theta(x)p(x)u_t(x)p(t, r)dxdtdr \\
&= \mathbb{E}_{t,r,x\sim p_t(x)}[\nabla u_{t\to r}^\theta(x)u_t(x)]
\end{aligned}
\tag{25}
$$

Therefore, we have

$$
\begin{aligned}
\nabla_\theta \mathcal{L}^E(\theta) &= \mathbb{E}_{t,r,x_1\sim p_{data},x\sim p_t(x|x_1)}[\nabla u_{t\to r}^\theta(x)(u_{t\to r}^\theta(x) - u_t(x|x_1) - B(\theta; r, t, x))] \\
&= \mathbb{E}_{t,r,x\sim p_t(x)}[\nabla u_{t\to r}^\theta(x)(u_{t\to r}^\theta(x) - u_t(x) - B(\theta; r, t, x))] \\
&= \nabla_\theta \mathcal{L}^R(\theta)
\end{aligned}
\tag{26}
$$

We then calculate the difference between $\nabla_\theta \mathcal{L}^R(\theta)$ and $\nabla_\theta \mathcal{L}^{\tilde{C}}(\theta)$ as

$$
\begin{aligned}
\nabla_\theta \mathcal{L}^R(\theta) - \nabla_\theta \mathcal{L}^{\tilde{C}}(\theta) &= \mathbb{E}_{t,r,x \sim p_t(x)}[\nabla_\theta u_{t\to r}^\theta(x)(u_{t\to r}^\theta(x) - u_t(x) - B(\theta;r,t,x)) \\
&\quad - \nabla_\theta u_{t\to r}^\theta(x)(u_{t\to r}^\theta(x) - u_{t\to t}^\theta(x) - B(\theta;r,t,x))\|^2] \\
&= \mathbb{E}_{t,r,x \sim p_t(x)}[\nabla_\theta u_{t\to r}^\theta(x)(u_{t\to t}^\theta(x) - u_t(x))]
\end{aligned}
\tag{27}
$$

Applying the Cauchy-Schwarz inequality and using the assumption $M_g = \sqrt{\mathbb{E}_{t,r,x\sim p_t(x)}[\frac{1}{m}\|\nabla_\theta u_{t\to r}^\theta(x)\|_2^2]} < +\infty$ in Assumption 1, we further obtain the following bound:

$$
\begin{aligned}
MSE(\nabla_\theta \mathcal{L}^R(\theta), \nabla_\theta \mathcal{L}^{\tilde{C}}(\theta)) &= \frac{1}{\sqrt{m}}\|\mathbb{E}_{t,r,x\sim p_t(x)}[\nabla_\theta u_{t\to r}^\theta(x)(u_{t\to t}^\theta(x) - u_t(x))]\| \\
&\leq \frac{1}{\sqrt{m}}\mathbb{E}_{t,r,x\sim p_t(x)}[\|\nabla_\theta u_{t\to r}^\theta(x)(u_{t\to t}^\theta(x) - u_t(x))\|] \\
&\leq \frac{1}{\sqrt{m}}\mathbb{E}_{t,r,x\sim p_t(x)}[\|\nabla_\theta u_{t\to r}^\theta(x)\|_2\|u_{t\to t}^\theta(x) - u_t(x)\|] \\
&\leq \sqrt{\mathbb{E}_{t,r,x\sim p_t(x)}[\frac{1}{m}\|\nabla_\theta u_{t\to r}^\theta(x)\|_2^2]\mathbb{E}_{t,r,x\sim p_t(x)}[\|u_{t\to t}^\theta(x) - u_t(x)\|^2]} \\
&\leq M_g\sqrt{\mathbb{E}_{t,r,x\sim p_t(x)}[\|u_{t\to t}^\theta(x) - u_t(x)\|^2]}
\end{aligned}
\tag{28}
$$

Combine Equation 45 and Equation 47, we have

$$
MSE(\nabla \mathcal{L}^E(\theta), \nabla \mathcal{L}^{\tilde{C}}(\theta)) \leq M_g\sqrt{\mathbb{E}_{t,r,x\sim p_t(x)}[\|u_{t\to t}^\theta(x) - u_t(x)\|^2]}
\tag{29}
$$

$\square$

## B.3. Proof of Theorem 4.3

**Theorem 4.3** (Surrogate Loss Validity) With $M_g = \sqrt{\mathbb{E}_{t,r,x\sim p_t(x)}[\frac{1}{m}\|\nabla_\theta u_{t\to r}^\theta(x)\|_2^2]} < +\infty$, $M_x = \sqrt{\mathbb{E}_{t,r,x\sim p_t(x)}[\frac{1}{m}\|\partial_x u_{t+\Delta t\to r}^\theta(x')\|_2^2]} < +\infty$, and $M_t = \sqrt{\mathbb{E}_{t,r,x\sim p_t(x)}[\|\partial_s u_{t\to s}^\theta|_{s=t}\|^2]} < +\infty$ hold in Assumption 1, Our Euler Mean Flow loss $\mathcal{L}^E(\theta)$ and the trajectory consistency loss $\mathcal{L}^C(\theta)$ satisfy

$$
MSE(\nabla_\theta L^E(\theta), \nabla_\theta L^C(\theta)) \leq M_g\sqrt{\mathbb{E}_{t,r,x\sim p_t(x)}[\|u_{t\to t}^\theta(x) - u_t(x)\|^2]} + (M_g M_t + M_x M_t)\Delta t + O(\Delta t^2)
\tag{30}
$$

*Proof.* We define $C(\theta;r,t,x) = sg(\frac{u_{t+\Delta t\to r}^\theta(x') - u_{t\to r}^\theta(x)}{\Delta t})$, $x' = x + u_{t\to t}^\theta(x)\Delta t$ and $D(\theta;r,t,x) = sg(\frac{u_{t+\Delta t\to r}^\theta(x'') - u_{t\to r}^\theta(x)}{\Delta t})$, $x'' = x + u_{t\to t+\Delta t}^\theta(x)\Delta t$. With these definitions, the approximated trajectory consistency loss $\mathcal{L}^{\tilde{C}}(\theta)$ and the trajectory consistency loss $\mathcal{L}^C(\theta)$ can be written as

$$
\begin{aligned}
\mathcal{L}^{\tilde{C}}(\theta) &= \mathbb{E}_{t,r,x\sim p_t(x)}[\|u_{t\to r}^\theta(x) - sg(u_{t\to t}^\theta(x) + (r - t - \Delta t)C(\theta;r,t,x))\|^2] \\
\mathcal{L}^C(\theta) &= \mathbb{E}_{t,r,x\sim p_t(x)}[\|u_{t\to r}^\theta(x) - sg(u_{t\to t+\Delta t}^\theta(x) + (r - t - \Delta t)D(\theta;r,t,x))\|^2]
\end{aligned}
\tag{31}
$$

We now analyze the difference $\nabla_\theta \mathcal{L}^{\tilde{C}}(\theta) - \nabla_\theta \mathcal{L}^C(\theta)$ between the gradients of these two objectives. A direct computation yields

$$
\begin{aligned}
\nabla \mathcal{L}_\theta^{\tilde{C}}(\theta) - \nabla \mathcal{L}_\theta^C(\theta) &= \mathbb{E}_{t,r,x\sim p_t(x)}[\nabla_\theta u_{t\to r}^\theta(x)(u_{t\to r}^\theta(x) - (u_t^\theta(x) + (r - t - \Delta t)C(\theta;r,t,x)))] \\
&\quad - \mathbb{E}_{t,r,x\sim p_t(x)}[\nabla_\theta u_{t\to r}^\theta(x)(u_{t\to r}^\theta(x) - (u_{t\to t+\Delta t}^\theta(x) + (r - t - \Delta t)D(\theta;r,t,x)))] \\
&= \mathbb{E}_{t,r,x\sim p_t(x)}[\nabla_\theta u_{t\to r}^\theta(x)((u_{t\to t+\Delta t}^\theta(x) - u_{t\to t}^\theta(x)) + (r - t - \Delta t)(D(\theta;r,t,x) - C(\theta;r,t,x)))]
\end{aligned}
\tag{32}
$$

We first bound the difference $D(\theta; r, t, x) - C(\theta; r, t, x)$. By definition,

$$
\begin{aligned}
\|D(\theta; r, t, x) - C(\theta; r, t, x)\| &= \|\frac{u_{t+\Delta t \to r}^\theta(x') - u_{t\to r}^\theta(x)}{\Delta t} - \frac{u_{t+\Delta t \to r}^\theta(x'') - u_{t\to r}^\theta(x)}{\Delta t}\| \\
&= \frac{1}{\Delta t}\|u_{t+\Delta t \to r}^\theta(x') - u_{t+\Delta t \to r}^\theta(x'')\| \\
&= \frac{1}{\Delta t}\|\partial_x u_{t+\Delta t \to r}^\theta(x')(x'' - x')\| \\
&= \frac{1}{\Delta t}\|\partial_x u_{t+\Delta t \to r}^\theta(x')(u_{t \to t+\Delta t}^\theta(x) - u_{t \to t}^\theta(x))\Delta t\| \\
&= \|\partial_x u_{t+\Delta t \to r}^\theta(x')(u_{t \to t+\Delta t}^\theta(x) - u_{t \to t}^\theta(x))\| \\
&\leq \|\partial_x u_{t+\Delta t \to r}^\theta(x')\|_2 \|u_{t \to t+\Delta t}^\theta(x) - u_{t \to t}^\theta(x)\|
\end{aligned}
\tag{33}
$$

Next, the difference $u_{t \to t+\Delta t}^\theta(x) - u_{t \to t}^\theta(x)$ admits a first-order expansion:

$$
\begin{aligned}
\|u_{t \to t+\Delta t}^\theta(x) - u_{t \to t}^\theta(x)\| &= \|u_{t \to t}^\theta(x) + \Delta t \partial_s u_{t \to s}^\theta|_{s=t} + O(\Delta t^2) - u_{t \to t}^\theta(x)\| \\
&\leq \Delta t \|\partial_s u_{t \to s}^\theta|_{s=t}\| + O(\Delta t^2)
\end{aligned}
\tag{34}
$$

Combining the above estimates, we can bound $\|\nabla_\theta \mathcal{L}^{\tilde{C}}(\theta) - \nabla_\theta \mathcal{L}^C(\theta)\|$ as

$$
MSE(\nabla_\theta \mathcal{L}^{\tilde{C}}(\theta), \nabla_\theta \mathcal{L}^C(\theta))
$$

$$
\leq \frac{1}{\sqrt{m}} \mathbb{E}_{t,r,x\sim p_t(x)} \|\nabla_\theta u_{t\to r}^\theta(x)((u_{t\to t+\Delta t}^\theta(x) - u_{t\to t}^\theta(x)) + (r-t-\Delta t)(D(\theta;r,t,x) - C(\theta;r,t,x)))\|
$$

$$
\leq \frac{1}{\sqrt{m}} \mathbb{E}_{t,r,x\sim p_t(x)} [\|\nabla_\theta u_{t\to r}^\theta(x)(u_{t\to t+\Delta t}^\theta(x) - u_{t\to t}^\theta(x))\| + (r-t-\Delta t)\|D(\theta;r,t,x) - C(\theta;r,t,x)\|]
$$

$$
\leq \frac{1}{\sqrt{m}} \mathbb{E}_{t,r,x\sim p_t(x)} [\|\nabla_\theta u_{t\to r}^\theta(x)\|_2 \|u_{t\to t+\Delta t}^\theta(x) - u_{t\to t}^\theta(x)\| + (r-t-\Delta t)\|\partial_x u_{t+\Delta t \to r}^\theta(x')\|_2 \|u_{t\to t+\Delta t}^\theta(x) - u_{t\to t}^\theta(x)\|]
$$

$$
\leq \frac{1}{\sqrt{m}} \mathbb{E}_{t,r,x\sim p_t(x)} [\|\nabla_\theta u_{t\to r}^\theta(x)\|_2 \|u_{t\to t+\Delta t}^\theta(x) - u_{t\to t}^\theta(x)\|] + \mathbb{E}_{t,r,x\sim p_t(x)} [\|\partial_x u_{t+\Delta t \to r}^\theta(x')\|_2 \|u_{t\to t+\Delta t}^\theta(x) - u_{t\to t}^\theta(x)\|]
$$

$$
\leq \sqrt{\mathbb{E}_{t,r,x\sim p_t(x)}[\frac{1}{m}\|\nabla_\theta u_{t\to r}^\theta(x)\|_2^2] \mathbb{E}_{t,r,x\sim p_t(x)}[\|u_{t\to t+\Delta t}^\theta(x) - u_{t\to t}^\theta(x)\|^2]}
$$

$$
+ \sqrt{\mathbb{E}_{t,r,x\sim p_t(x)}[\frac{1}{m}\|\partial_x u_{t+\Delta t \to r}^\theta(x')\|_2^2] \mathbb{E}_{t,r,x\sim p_t(x)}[\|u_{t\to t+\Delta t}^\theta(x) - u_{t\to t}^\theta(x)\|^2]}
$$

$$
\leq (\sqrt{\mathbb{E}_{t,r,x\sim p_t(x)}[\frac{1}{m}\|\nabla_\theta u_{t\to r}^\theta(x)\|_2^2] \mathbb{E}_{t,r,x\sim p_t(x)}[\|\partial_s u_{t\to s}^\theta|_{s=t}\|^2}
$$

$$
+ \sqrt{\mathbb{E}_{t,r,x\sim p_t(x)}[\frac{1}{m}\|\partial_x u_{t+\Delta t \to r}^\theta(x')\|_2^2] \mathbb{E}_{t,r,x\sim p_t(x)}[\|\partial_s u_{t\to s}^\theta|_{s=t}\|^2]}) \Delta t + O(\Delta t^2)
$$

$$
= (M_g M_t + M_x M_t)\Delta t + O(\Delta t^2)
$$

$$
\tag{35}
$$

where $M_g = \sqrt{\mathbb{E}_{t,r,x\sim p_t(x)}[\frac{1}{m}\|\nabla_\theta u_{t\to r}^\theta(x)\|_2^2]} < +\infty$, $M_x = \sqrt{\mathbb{E}_{t,r,x\sim p_t(x)}[\frac{1}{m}\|\partial_x u_{t+\Delta t \to r}^\theta(x')\|_2^2]} < +\infty$, and $M_t = \sqrt{\mathbb{E}_{t,r,x\sim p_t(x)}[\|\partial_s u_{t\to s}^\theta|_{s=t}\|^2]} < +\infty$ as Assumption 1.

Combine with Lemma 1, we have

$$
\begin{aligned}
MSE(\nabla_\theta L^E(\theta), \nabla_\theta L^C(\theta)) &\leq MSE(\nabla_\theta L^E(\theta), \nabla_\theta L^{\tilde{C}}(\theta)) + MSE(\nabla_\theta L^{\tilde{C}}(\theta), \nabla_\theta L^C(\theta)) \\
&\leq M_g \sqrt{\mathbb{E}_{t,r,x\sim p_t(x)}[\|u_{t\to t}^\theta(x) - u_t(x)\|^2]} + (M_g M_t + M_x M_t)\Delta t + O(\Delta t^2)
\end{aligned}
\tag{36}
$$

$\square$

## B.4. Derivation of Equation 13

Substituting this relation $\tilde{x}_{t\to r}(x) = (1-t)\,u_{t\to r}(x) + x$ and $x_s = \phi_{t\to s}(x_t) = \frac{s-t}{1-t}(\tilde{x}_{t\to s}(x_t) - x_t) + x_t$ into Equation 8, we obtain

$$(r-t)u_{t\to r}(x_t) = (s-t)u_{t\to s}(x_t) + (r-s)u_{s\to r}(x_s)$$

$$(r-t)(\frac{\tilde{x}_{t\to r}(x_t) - x_t}{1-t}) = (s-t)\frac{\tilde{x}_{t\to s}(x_t) - x_t}{1-t} + (r-s)\frac{\tilde{x}_{s\to r}(x_s) - x_s}{1-s}$$

$$\tilde{x}_{t\to r}(x_t) = x_t + \frac{s-t}{r-t}(\tilde{x}_{t\to s}(x_t) - x_t) + \frac{(1-t)(r-s)}{(1-s)(r-t)}(\tilde{x}_{s\to r}(x_s) - x_s)$$

$$\tilde{x}_{t\to r}(x_t) = x_t + \frac{s-t}{r-t}(\tilde{x}_{t\to s}(x_t) - x_t) + \frac{(1-t)(r-s)}{(1-s)(r-t)}(\tilde{x}_{s\to r}(x_s) - \frac{s-t}{1-t}(\tilde{x}_{t\to s}(x_t) - x_t) - x_t)$$

$$\tilde{x}_{t\to r}(x_t) = x_t + \frac{s-t}{r-t}(\tilde{x}_{t\to s}(x_t) - x_t) + \frac{(1-t)(r-s)}{(1-s)(r-t)}(\tilde{x}_{s\to r}(x_s) - \frac{s-t}{1-t}\tilde{x}_{t\to s}(x_t) - \frac{1-s}{1-t}x_t)$$

$$\tilde{x}_{t\to r}(x_t) = \frac{s-t}{r-t}\tilde{x}_{t\to s}(x_t) + \frac{(1-t)(r-s)}{(1-s)(r-t)}(\tilde{x}_{s\to r}(x_s) - \frac{s-t}{1-t}\tilde{x}_{t\to s}(x_t))$$

$$\tilde{x}_{t\to r}(x_t) = \frac{1-r}{1-s}\frac{s-t}{r-t}\tilde{x}_{t\to s}(x_t) + \frac{(1-t)(r-s)}{(1-s)(r-t)}\tilde{x}_{s\to r}(x_s)$$

$$\frac{(s-t)(1-r)}{(1-s)(r-t)}\tilde{x}_{t\to r}(x_t) = \frac{1-r}{1-s}\frac{s-t}{r-t}\tilde{x}_{t\to s}(x_t) + \frac{(1-t)(r-s)}{(1-s)(r-t)}(\tilde{x}_{s\to r}(x_s) - \tilde{x}_{t\to r}(x_t))$$

$$\tilde{x}_{t\to r}(x_t) = \tilde{x}_{t\to s}(x_t) + (r-s)\frac{(1-t)}{(1-r)}\frac{\tilde{x}_{s\to r}(x_s) - \tilde{x}_{t\to r}(x_t)}{s-t} \tag{37}$$

## B.5. Lemma 2 and its proof

**Lemma 2.** *With $M'_g = \sqrt{\mathbb{E}_{t,r,x\sim p_t(x)}[\frac{1}{m}\|\nabla_\theta \tilde{x}^\theta_{t\to r}(x)\|_2^2]} < +\infty$ holds in Assumption 2, $x_1$-prediction Euler Mean Flow loss $\mathcal{L}^{E'}(\theta)$ and the approximated $x_1$-prediction trajectory consistency loss $\mathcal{L}^{\tilde{C}'}(\theta)$ satisfy*

$$MSE(\nabla\mathcal{L}^{E'}(\theta), \nabla\mathcal{L}^{\tilde{C}'}(\theta)) \leq M'_g\sqrt{\mathbb{E}_{t,r,x\sim p_t(x)}[\|\tilde{x}^\theta_{t\to t}(x) - \tilde{x}_t(x)\|^2]} \tag{38}$$

*Consequently, during training, if $\|\tilde{x}^\theta_{t\to t}(x) - \tilde{x}_t(x)\|^2 \to 0$, then $\mathcal{L}^{E'}(\theta)$ and $\mathcal{L}^{\tilde{C}'}(\theta)$ share the same optimal target at $\theta$. The term $\tilde{x}_t(x)$ denotes the reference instantaneous velocity at $x$, defined as $\tilde{x}_t(x) = \mathbb{E}_{x_1\sim p(x_1|x)}[\tilde{x}_t(x|x_1)]$, which is generally intractable to compute analytically.*

*Here, the approximated $x_1$-prediction trajectory consistency loss $\mathcal{L}^{\tilde{C}'}(\theta)$ are defined as*

$$\mathcal{L}^{\tilde{C}'}(\theta) = \mathbb{E}_{t,r,x\sim p_t(x),x'=sg(\Delta t\frac{\tilde{x}^\theta_{t\to t}(x)-x}{1-t})+x}$$
$$[\|\tilde{x}^\theta_{t\to r}(x) - (\tilde{x}^\theta_{t\to t}(x) + (r-t-\Delta t)$$
$$\frac{1-t}{1-r}sg(\frac{\tilde{x}^\theta_{t+\Delta t\to r}(x') - \tilde{x}^\theta_{t\to r}(x)}{\Delta t}))\|^2] \tag{39}$$

*It is straightforward to verify that the loss $\mathcal{L}^{\tilde{C}'}(\theta)$ is the mean-velocity formulation of $\mathcal{L}^{C'}(\theta)$ under the local linear approximation in Equation 10, expressed via $\tilde{x}_{t\to r}(x) = (1-t)\frac{\phi_{t\to r}(x)-x}{r-t} + x$, and differs by a temporal scaling factor $\frac{(1-t-\Delta t)}{\Delta t(1-t)(1-r)}$.*

*Proof.* We first define the reference regression loss $\mathcal{L}^R(\theta)$ as

$$\mathcal{L}^{R'}(\theta) = \mathbb{E}_{t,r,x\sim p_t(x),x'=sg(\Delta t\frac{\tilde{x}^\theta_{t\to t}(x)-x}{1-t})+x}$$
$$[\|\tilde{x}^\theta_{t\to r}(x) - (\tilde{x}_t(x) + (r-t-\Delta t)$$
$$\frac{1-t}{1-r}sg(\frac{\tilde{x}^\theta_{t+\Delta t\to r}(x') - \tilde{x}^\theta_{t\to r}(x)}{\Delta t}))\|^2] \tag{40}$$

Let $B'(\theta; r, t, x) = (r - t - \Delta t)\frac{1-t}{1-r}sg(\frac{\tilde{x}^\theta_{t+\Delta t \to r}(x') - \tilde{x}^\theta_{t \to r}(x)}{\Delta t})$. Since $B'(\theta; r, t, x)$ contains the stop-gradient operator $sg(\cdot)$, it satisfies $\nabla_\theta B'(\theta; r, t, x) = 0$. Using $B'(\theta; r, t, x)$, the $x_1$-prediction Euler Mean Flow loss $\mathcal{L}^{E'}(\theta)$, the $x_1$-prediction reference regression loss $\mathcal{L}^{R'}(\theta)$, and the approximated trajectory consistency loss $\mathcal{L}^{\tilde{C}'}(\theta)$ can be written as

$$\mathcal{L}^{E'}(\theta) = \mathbb{E}_{t,r,x_1 \sim p_{data}, x \sim p_t(x|x_1)}[\|\tilde{x}^\theta_{t \to r}(x) - (\tilde{x}_t(x|x_1) + B'(\theta; r, t, x))\|^2]$$

$$\mathcal{L}^{R'}(\theta) = \mathbb{E}_{t,r,x \sim p_t(x)}[\|\tilde{x}^\theta_{t \to r}(x) - (\tilde{x}_t(x) + B'(\theta; r, t, x))\|^2] \tag{41}$$

$$\mathcal{L}^{\tilde{C}'}(\theta) = \mathbb{E}_{t,r,x \sim p_t(x)}[\|\tilde{x}^\theta_{t \to r}(x) - (\tilde{x}^\theta_{t \to t}(x) + B'(\theta; r, t, x))\|^2]$$

We first show that the $x_1$-prediction Euler Mean Flow loss $\mathcal{L}^{E'}(\theta)$ and $x_1$-prediction the reference regression loss $\mathcal{L}^{R'}(\theta)$ satisfy $\nabla_\theta \mathcal{L}^{E'}(\theta) = \nabla_\theta \mathcal{L}^{R'}(\theta)$. Expanding $\nabla_\theta \mathcal{L}^{E'}(\theta)$, we obtain

$$\nabla_\theta \mathcal{L}^{E'}(\theta) = \mathbb{E}_{t,r,x_1 \sim p_{data}, x \sim p_t(x|x_1),}[\nabla_\theta\|\tilde{x}^\theta_{t \to r}(x) - (\tilde{x}_t(x|x_1) + B'(\theta; r, t, x))\|^2]$$

$$\overset{\nabla_\theta B'(\theta;r,t,x)=0}{=} \mathbb{E}_{t,r,x_1 \sim p_{data}, x \sim p_t(x|x_1)}[\nabla_\theta\tilde{x}^\theta_{t \to r}(x)(\tilde{x}^\theta_{t \to r}(x) - \tilde{x}_t(x|x_1) - B'(\theta; r, t, x))] \tag{42}$$

$$\overset{\nabla_\theta B'(\theta;r,t,x)=0}{=} \mathbb{E}_{t,r,x_1 \sim p_{data}, x \sim p_t(x|x_1)}[\nabla_\theta\tilde{x}^\theta_{t \to r}(x)(\tilde{x}^\theta_{t \to r}(x) - \tilde{x}_t(x|x_1) - B'(\theta; r, t, x))]$$

where $\mathbb{E}_{t,r,x_1 \sim p_{data}, x \sim p_t(x|x_1)}[\nabla_\theta\tilde{x}^\theta_{t \to r}(x)(\tilde{x}^\theta_{t \to r}(x) - B'(\theta; r, t, x))]$ can be computed as:

$$\mathbb{E}_{t,r,x_1 \sim p_{data}, x \sim p_t(x|x_1)}[\nabla_\theta\tilde{x}^\theta_{t \to r}(x)(\tilde{x}^\theta_{t \to r}(x) - B'(\theta; r, t, x))]$$

$$= \int_{t,r} \int_{x_1} \int_x (\nabla_\theta\tilde{x}^\theta_{t \to r}(x)(\tilde{x}^\theta_{t \to r}(x) - B'(\theta; r, t, x)))p(x|x_1)p_{data}(x_1)p(t,r)dxdx_1dtdr$$

$$= \int_{t,r} \int_x (\nabla_\theta\tilde{x}^\theta_{t \to r}(x)(\tilde{x}^\theta_{t \to r}(x) - B'(\theta; r, t, x))) \int_{x_1} p(x|x_1)p_{data}(x_1)dx_1 p(t,r)dxdtdr \tag{43}$$

$$= \int_{t,r} \int_x (\nabla_\theta\tilde{x}^\theta_{t \to r}(x)(\tilde{x}^\theta_{t \to r}(x) - B'(\theta; r, t, x)))p(x)p(t,r)dxdtdr$$

$$= \mathbb{E}_{t,r,x \sim p_t(x)}[\nabla\tilde{x}^\theta_{t \to r}(x)(\tilde{x}^\theta_{t \to r}(x) - B'(\theta; r, t, x))]$$

And $\mathbb{E}_{t,r,x_1 \sim p_{data}, x \sim p_t(x|x_1)}[\nabla\tilde{x}^\theta_{t \to r}(x)\tilde{x}_t(x|x_1)]$ can be calculated as

$$\mathbb{E}_{t,r,x_1 \sim p_{data}, x \sim p_t(x|x_1)}[\nabla_\theta\tilde{x}^\theta_{t \to r}(x)\tilde{x}_t(x|x_1)]$$

$$= \int_{t,r} \int_{x_1} \int_x \nabla_\theta\tilde{x}^\theta_{t \to r}(x)\tilde{x}_t(x|x_1)p(x|x_1)p_{data}(x_1)p(t,r)dxdx_1dtdr$$

$$= \int_{t,r} \int_x \nabla_\theta\tilde{x}^\theta_{t \to r}(x)(\int_{x_1} \tilde{x}_t(x|x_1)p(x|x_1)p_{data}(x_1)dx_1)p(t,r)dxdtdr$$

$$= \int_{t,r} \int_x \nabla_\theta\tilde{x}^\theta_{t \to r}(x)(\int_{x_1} \tilde{x}_t(x|x_1)p(x_1|x)p(x)dx_1)p(t,r)dxdtdr \tag{44}$$

$$= \int_{t,r} \int_x \nabla_\theta\tilde{x}^\theta_{t \to r}(x)p(x)\tilde{x}_t(x)p(t,r)dxdtdr$$

$$= \mathbb{E}_{t,r,x \sim p_t(x)}[\nabla_\theta\tilde{x}^\theta_{t \to r}(x)\tilde{x}_t(x)]$$

Therefore, we have

$$\nabla_\theta \mathcal{L}^{E'}(\theta) = \mathbb{E}_{t,r,x_1 \sim p_{data}, x \sim p_t(x|x_1)}[\nabla\tilde{x}^\theta_{t \to r}(x)(\tilde{x}^\theta_{t \to r}(x) - \tilde{x}_t(x|x_1) - B(\theta; r, t, x))]$$

$$= \mathbb{E}_{t,r,x \sim p_t(x)}[\nabla\tilde{x}^\theta_{t \to r}(x)(\tilde{x}^\theta_{t \to r}(x) - \tilde{x}_t(x) - B(\theta; r, t, x))] \tag{45}$$

$$= \nabla_\theta \mathcal{L}^{R'}(\theta)$$

We then calculate the difference between $\nabla_\theta \mathcal{L}^{R'}(\theta)$ and $\nabla_\theta \mathcal{L}^{\tilde{C}'}(\theta)$ as

$$\nabla_\theta \mathcal{L}^{R'}(\theta) - \nabla_\theta \mathcal{L}^{\tilde{C}'}(\theta) = \mathbb{E}_{t,r,x \sim p_t(x)}[\nabla_\theta\tilde{x}^\theta_{t \to r}(x)(\tilde{x}^\theta_{t \to r}(x) - \tilde{x}_t(x) - B(\theta; r, t, x))$$

$$- \nabla_\theta\tilde{x}^\theta_{t \to r}(x)(\tilde{x}^\theta_{t \to r}(x) - \tilde{x}^\theta_{t \to t}(x) - B(\theta; r, t, x))\|^2] \tag{46}$$

$$= \mathbb{E}_{t,r,x \sim p_t(x)}[\nabla_\theta\tilde{x}^\theta_{t \to r}(x)(\tilde{x}^\theta_{t \to t}(x) - \tilde{x}_t(x))]$$

Applying the Cauchy-Schwarz inequality and using the assumption $M_g' = \sqrt{\mathbb{E}_{t,r,x \sim p_t(x)}[\frac{1}{m}\|\nabla_\theta \tilde{x}_{t \to r}^\theta(x)\|^2]} < +\infty$ in Assumption 2, we further obtain the following bound:

$$
\begin{aligned}
MSE(\nabla_\theta \mathcal{L}^{R'}(\theta), \nabla_\theta \mathcal{L}^{\tilde{C}'}(\theta)) &= \frac{1}{\sqrt{m}}\|\mathbb{E}_{t,r,x \sim p_t(x)}[\nabla_\theta \tilde{x}_{t \to r}^\theta(x)(\tilde{x}_{t \to t}^\theta(x) - \tilde{x}_t(x))]\| \\
&\leq \frac{1}{\sqrt{m}}\mathbb{E}_{t,r,x \sim p_t(x)}[\|\nabla_\theta \tilde{x}_{t \to r}^\theta(x)(\tilde{x}_{t \to t}^\theta(x) - \tilde{x}_t(x))\|] \\
&\leq \frac{1}{\sqrt{m}}\mathbb{E}_{t,r,x \sim p_t(x)}[\|\nabla_\theta \tilde{x}_{t \to r}^\theta(x)\|_2\|\tilde{x}_{t \to t}^\theta(x) - \tilde{x}_t(x)\|] \\
&\leq \sqrt{\mathbb{E}_{t,r,x \sim p_t(x)}[\frac{1}{m}\|\nabla_\theta \tilde{x}_{t \to r}^\theta(x)\|_2^2]\mathbb{E}_{t,r,x \sim p_t(x)}[\|\tilde{x}_{t \to t}^\theta(x) - \tilde{x}_t(x)\|^2]} \\
&\leq M_g'\sqrt{\mathbb{E}_{t,r,x \sim p_t(x)}[\|\tilde{x}_{t \to t}^\theta(x) - \tilde{x}_t(x)\|^2]}
\end{aligned}
\tag{47}
$$

Combine Equation 45 and Equation 47, we have

$$
MSE(\nabla \mathcal{L}^{E'}(\theta), \nabla \mathcal{L}^{\tilde{C}'}(\theta)) \leq M_g'\sqrt{\mathbb{E}_{t,r,x \sim p_t(x)}[\|\tilde{x}_{t \to t}^\theta(x) - \tilde{x}_t(x)\|^2]}
\tag{48}
$$

$\square$

## B.6. Proof of Theorem 4.4

**Theorem 4.4** (Surrogate Loss Validity for $x_1$-Prediction) With $M_g' = \sqrt{\mathbb{E}_{t,r,x \sim p_t(x)}[\frac{1}{m}\|\nabla_\theta u_{t \to r}^\theta(x)\|_2^2]} < +\infty$, $M_x' = \sqrt{\mathbb{E}_{t,r,x \sim p_t(x)}[\frac{1}{m}\|\partial_x u_{t+\Delta t \to r}^\theta(x')\|_2^2]} < +\infty$, and $M_t' = \sqrt{\mathbb{E}_{t,r,x \sim p_t(x)}[\|\partial_s u_{t \to s}^\theta|_{s=t}\|^2]} < +\infty$ hold in Assumption 2 and Lemma 2, our Euler Mean Flow loss $\mathcal{L}^{E'}(\theta)$ and the trajectory consistency loss $\mathcal{L}^{C'}(\theta)$ satisfy

$$
MSE(\nabla_\theta L^{E'}(\theta), \nabla_\theta L^{C'}(\theta)) \leq M_g'\sqrt{\mathbb{E}_{t,r,x \sim p_t(x)}[\|\tilde{x}_{t \to t}^\theta(x) - \tilde{x}_t(x)\|^2]} + (M_g'M_t' + M_x'M_t')\Delta t + O(\Delta t^2)
\tag{49}
$$

*Proof.* We define $C'(\theta; r, t, x) = sg(\frac{\tilde{x}_{t+\Delta t \to r}^\theta(x') - \tilde{x}_{t \to r}^\theta(x)}{\Delta t})$, $x' = x + \frac{\tilde{x}_{t \to t}^\theta(x) - x}{1-t}\Delta t$ and $D'(\theta; r, t, x) = sg(\frac{\tilde{x}_{t+\Delta t \to r}^\theta(x'') - \tilde{x}_{t \to r}^\theta(x)}{\Delta t})$, $x'' = x + \frac{\tilde{x}_{t \to t+\Delta t}^\theta(x) - x}{1-t}\Delta t$. With these definitions, the approximated trajectory consistency loss $\mathcal{L}^{\tilde{C}'}(\theta)$ and the trajectory consistency loss $\mathcal{L}^{C'}(\theta)$ can be written as

$$
\begin{aligned}
\mathcal{L}^{\tilde{C}'}(\theta) &= \mathbb{E}_{t,r,x \sim p_t(x)}[\|\tilde{x}_{t \to r}^\theta(x) - sg(\tilde{x}_{t \to t}^\theta(x) + (r - t - \Delta t)\frac{1-t}{1-r}C(\theta; r, t, x))\|^2] \\
\mathcal{L}^{C'}(\theta) &= \mathbb{E}_{t,r,x \sim p_t(x)}[\|\tilde{x}_{t \to r}^\theta(x) - sg(\tilde{x}_{t \to t+\Delta t}^\theta(x) + (r - t - \Delta t)\frac{1-t}{1-r}D(\theta; r, t, x))\|^2]
\end{aligned}
\tag{50}
$$

We now analyze the difference $\nabla_\theta \mathcal{L}^{\tilde{C}'}(\theta) - \nabla_\theta \mathcal{L}^{C'}(\theta)$ between the gradients of these two objectives. A direct computation yields

$$
\begin{aligned}
\nabla \mathcal{L}_\theta^{\tilde{C}'}(\theta) - \nabla \mathcal{L}_\theta^{C'}(\theta) &= \mathbb{E}_{t,r,x \sim p_t(x)}[\nabla_\theta \tilde{x}_{t \to r}^\theta(x)(\tilde{x}_{t \to r}^\theta(x) - (\tilde{x}_t^\theta(x) + (r - t - \Delta t)\frac{1-t}{1-r}C'(\theta; r, t, x))] \\
&\quad - \mathbb{E}_{t,r,x \sim p_t(x)}[\nabla_\theta \tilde{x}_{t \to r}^\theta(x)(\tilde{x}_{t \to r}^\theta(x) - (\tilde{x}_{t \to t+\Delta t}^\theta(x) + (r - t - \Delta t)\frac{1-t}{1-r}D'(\theta; r, t, x))] \\
&= \mathbb{E}_{t,r,x \sim p_t(x)}[\nabla_\theta \tilde{x}_{t \to r}^\theta(x)((\tilde{x}_{t \to t+\Delta t}^\theta(x) - \tilde{x}_{t \to t}^\theta(x)) + (r - t - \Delta t)\frac{1-t}{1-r}(D'(\theta; r, t, x) - C'(\theta; r, t, x)))]
\end{aligned}
\tag{51}
$$

We first bound the difference $D'(\theta; r, t, x) - C'(\theta; r, t, x)$. By definition,

$$
\begin{aligned}
\|D'(\theta; r, t, x) - C'(\theta; r, t, x)\| &= \|\frac{\tilde{x}^\theta_{t+\Delta t \to r}(x') - \tilde{x}^\theta_{t \to r}(x)}{\Delta t} - \frac{\tilde{x}^\theta_{t+\Delta t \to r}(x'') - \tilde{x}^\theta_{t \to r}(x)}{\Delta t}\| \\
&= \frac{1}{\Delta t}\|\tilde{x}^\theta_{t+\Delta t \to r}(x') - \tilde{x}^\theta_{t+\Delta t \to r}(x'')\| \\
&= \frac{1}{\Delta t}\|\partial_x \tilde{x}^\theta_{t+\Delta t \to r}(x')(x'' - x')\| \\
&= \frac{1}{\Delta t}\|\partial_x \tilde{x}^\theta_{t+\Delta t \to r}(x')(\frac{\tilde{x}^\theta_{t \to t+\Delta t}(x) - \tilde{x}^\theta_{t \to t}(x)}{1-t})\Delta t\| \\
&= \|\partial_x \tilde{x}^\theta_{t+\Delta t \to r}(x')(\frac{\tilde{x}^\theta_{t \to t+\Delta t}(x) - \tilde{x}^\theta_{t \to t}(x)}{1-t})\| \\
&\leq \frac{1}{1-t}\|\partial_x \tilde{x}^\theta_{t+\Delta t \to r}(x')\|_2\|\tilde{x}^\theta_{t \to t+\Delta t}(x) - \tilde{x}^\theta_{t \to t}(x)\|
\end{aligned}
\tag{52}
$$

Next, the difference $\tilde{x}^\theta_{t \to t+\Delta t}(x) - \tilde{x}^\theta_{t \to t}(x)$ admits a first-order expansion:

$$
\begin{aligned}
\|\tilde{x}^\theta_{t \to t+\Delta t}(x) - \tilde{x}^\theta_{t \to t}(x)\| &= \|\tilde{x}^\theta_{t \to t}(x) + \Delta t \partial_s \tilde{x}^\theta_{t \to s}|_{s=t} + O(\Delta t^2) - \tilde{x}^\theta_{t \to t}(x)\| \\
&\leq \Delta t\|\partial_s \tilde{x}^\theta_{t \to s}|_{s=t}\| + O(\Delta t^2)
\end{aligned}
\tag{53}
$$

Combining the above estimates, we can bound $MSE(\nabla_\theta \mathcal{L}^{\tilde{C}'}(\theta), \nabla_\theta \mathcal{L}^{C'}(\theta))$ as

$$
\begin{aligned}
&MSE(\nabla_\theta \mathcal{L}^{\tilde{C}'}(\theta), \nabla_\theta \mathcal{L}^{C'}(\theta)) \\
&\leq \frac{1}{\sqrt{m}}\mathbb{E}_{t,r,x \sim p_t(x)}\|\nabla_\theta \tilde{x}^\theta_{t \to r}(x)((\tilde{x}^\theta_{t \to t+\Delta t}(x) - \tilde{x}^\theta_{t \to t}(x)) + (r-t-\Delta t)\frac{1-t}{1-r}(D'(\theta; r, t, x) - C'(\theta; r, t, x)))\| \\
&\leq \frac{1}{\sqrt{m}}\mathbb{E}_{t,r,x \sim p_t(x)}[\|\nabla_\theta \tilde{x}^\theta_{t \to r}(x)(\tilde{x}^\theta_{t \to t+\Delta t}(x) - \tilde{x}^\theta_{t \to t}(x))\| + \frac{1}{\sqrt{m}}(r-t-\Delta t)\frac{1-t}{1-r}\|D'(\theta; r, t, x) - C'(\theta; r, t, x)\|] \\
&\leq \frac{1}{\sqrt{m}}\mathbb{E}_{t,r,x \sim p_t(x)}[\|\nabla_\theta \tilde{x}^\theta_{t \to r}(x)\|_2\|\tilde{x}^\theta_{t \to t+\Delta t}(x) - \tilde{x}^\theta_{t \to t}(x)\| + \frac{1}{\sqrt{m}}(r-t-\Delta t)\frac{1}{1-r}\|\partial_x \tilde{x}^\theta_{t+\Delta t \to r}(x')\|_2\|\tilde{x}^\theta_{t \to t+\Delta t}(x) - \tilde{x}^\theta_{t \to t}(x)\|] \\
&\leq \frac{1}{\sqrt{m}}\mathbb{E}_{t,r,x \sim p_t(x)}[\|\nabla_\theta \tilde{x}^\theta_{t \to r}(x)\|_2\|\tilde{x}^\theta_{t \to t+\Delta t}(x) - \tilde{x}^\theta_{t \to t}(x)\|] + \frac{1}{\sqrt{m}}\frac{1}{1-r}\mathbb{E}_{t,r,x \sim p_t(x)}[\|\partial_x \tilde{x}^\theta_{t+\Delta t \to r}(x')\|_2\|\tilde{x}^\theta_{t \to t+\Delta t}(x) - \tilde{x}^\theta_{t \to t}(x)\|] \\
&\leq \sqrt{\mathbb{E}_{t,r,x \sim p_t(x)}[\frac{1}{m}\|\nabla_\theta \tilde{x}^\theta_{t \to r}(x)\|_2^2]\mathbb{E}_{t,r,x \sim p_t(x)}[\|\tilde{x}^\theta_{t \to t+\Delta t}(x) - \tilde{x}^\theta_{t \to t}(x)\|^2]} \\
&+ \frac{1}{1-r}\sqrt{\mathbb{E}_{t,r,x \sim p_t(x)}[\frac{1}{m}\|\partial_x \tilde{x}^\theta_{t+\Delta t \to r}(x')\|_2^2]\mathbb{E}_{t,r,x \sim p_t(x)}[\|\tilde{x}^\theta_{t \to t+\Delta t}(x) - \tilde{x}^\theta_{t \to t}(x)\|^2]} \\
&\leq (\sqrt{\mathbb{E}_{t,r,x \sim p_t(x)}[\frac{1}{m}\|\nabla_\theta \tilde{x}^\theta_{t \to r}(x)\|_2^2]\mathbb{E}_{t,r,x \sim p_t(x)}[\|\partial_s \tilde{x}^\theta_{t \to s}|_{s=t}\|^2}} \\
&+ \frac{1}{1-r}\sqrt{\mathbb{E}_{t,r,x \sim p_t(x)}[\frac{1}{m}\|\partial_x \tilde{x}^\theta_{t+\Delta t \to r}(x')\|_2^2]\mathbb{E}_{t,r,x \sim p_t(x)}[\|\partial_s \tilde{x}^\theta_{t \to s}|_{s=t}\|^2]})\Delta t + O(\Delta t^2) \\
&= (M_g M_t + M_x M_t)\Delta t + O(\Delta t^2)
\end{aligned}
\tag{54}
$$

where $M'_g = \sqrt{\mathbb{E}_{t,r,x \sim p_t(x)}[\frac{1}{m}\|\nabla_\theta \tilde{x}^\theta_{t \to r}(x)\|_2^2]} < +\infty$, $M_x = \frac{1}{1-r}\sqrt{\mathbb{E}_{t,r,x \sim p_t(x)}[\frac{1}{m}\|\partial_x \tilde{x}^\theta_{t+\Delta t \to r}(x')\|_2^2]} < +\infty$, and $M'_t = \sqrt{\mathbb{E}_{t,r,x \sim p_t(x)}[\|\partial_s \tilde{x}^\theta_{t \to s}|_{s=t}\|^2]} < +\infty$ as Assumption 2.

Combine with Lemma 2, we have

$$
\begin{aligned}
MSE(\nabla_\theta L^{E'}(\theta), \nabla_\theta L^{C'}(\theta)) &\leq MSE(\nabla_\theta L^{E'}(\theta) - \nabla_\theta L^{\tilde{C}'}(\theta)) + MSE(\nabla_\theta L^{\tilde{C}'}(\theta) - \nabla_\theta L^{C'}(\theta)) \\
&\leq M'_g\sqrt{\mathbb{E}_{t,r,x \sim p_t(x)}[\|\tilde{x}^\theta_{t \to t}(x) - \tilde{x}_t(x)\|^2]} + (M'_g M'_t + M'_x M'_t)\Delta t + O(\Delta t^2)
\end{aligned}
\tag{55}
$$

$\square$

# C. Model Architecture and Details of Dataset, Training, Sampling and Results

**Algorithm 2** Euler Mean Flow: Sampling
*Highlighted parts are used for conditional generation.*

**Require:** parameters $\theta$, learning rate $\eta$, noise sampler $\mathcal{N}$
1: **repeat**
2:      Sample $x_0 \sim \mathcal{N}$
3:      **if** $u$-prediction **then**
4:         $x_1 = u_{0\to1}^\theta(x, C) + x$
     **else if** $x_1$-prediction **then**
5:         $x_1 = \tilde{x}_{0\to1}^\theta(x, C)$
6:      **end if**
7: **until** convergence

## C.1. Algorithm Details

**Classifier-Free Guidance (CFG)** For conditional generation, we follow (Geng et al., 2025a) and apply classifier-free guidance (CFG) during training by modifying the conditional field. Specifically, $u(x|x_1)$ is replaced by $wu(x|x_1) + (1 - w - k)u_{t\to t}^\theta(x_t, C_0) + ku_{t\to t}^\theta(x_t, C)$, where $C$ is the label of $x_1$ and $C_0$ denotes the null label. The effective guidance scale is $w' = \frac{w}{1-k}$. Unconditional capability for $u_{t\to t}^\theta(x_t, C_0)$ is enabled by dropping labels with probability $p = 0.1$ during training.

**Time Sampler** Following (Geng et al., 2025a), we independently sample $t, r \sim \mathcal{T}_1$ and swap them if $t > r$, forming the sampler $\mathcal{T}$. We use $\mathcal{T}_1 = \mathcal{U}[0, 1]$ by default and a log-normal distribution for ImageNet. In addition, a fraction $\alpha$ of samples is constructed with $r = t$, corresponding to training the instantaneous model, e.g., $u_{t\to t}^\theta$. This ensures that the validity condition $\|u_{t\to t}^\theta(x) - u_t(x)\| \to 0$ required by Theorem 4.3 for $\mathcal{L}^{EFM}$ is satisfied.

**Adaptive Loss** For Training stability, we follow (Geng et al., 2025a) and adopt an adaptive loss (Geng et al., 2024) to reweight loss as $w\|\Delta\|_2^2$, $w = \frac{1}{(\|\Delta\|_2^2 + c)^p}$ to stabilize learning, where $\Delta$ denotes the discrepancy in the loss.

## C.2. Latent Space Image Generation

**Model** We adopt a Diffusion Transformer (DiT) (Peebles & Xie, 2023) architecture with DiT-B/2 configuration as our backbone for image generation. The input image $x$ is first encoded as a latent $z$ by a pretrained variational autoencoder (VAE) model (Kingma & Welling, 2022) from Stable Diffusion (Rombach et al., 2022a). For $256 \times 256 \times 3$ images, the shape of $z$ is $32 \times 32 \times 4$. The latent $z$ is first partitioned into non-overlapping patches of size $2 \times 2$, resulting in a token sequence. Each patch is linearly projected into a $D$-dimensional embedding space, where $D = 768$ for DiT-B/2. The backbone consists of a stack of Transformer blocks with multi-head self-attention and MLP layers. For conditioning, we follow the AdaLN-Zero design introduced in DiT. Specifically, the time embeddings for $t$ and $r$, together with optional class embeddings for conditional generation, are first projected through a small MLP and then used to modulate the Transformer blocks via adaptive layer normalization. See Table 8 and Table 9 for hyperparameters in detail.

| | | | | | | | |
|---|---|---|---|---|---|---|---|
| Batch Size | 64 | | Batch Size | 256 | | Batch Size | 64 |
| Training Steps | 400K | | Training Steps | 800K | | Training Steps | 600K |
| Classifier Free Guidance | - | | Classifier Free Guidance | 2.5 | | Classifier Free Guidance | - |
| Class Dropout Probability | - | | Class Dropout Probability | 0.1 | | Class Dropout Probability | - |
| EMA Ratio | 0.9999 | | EMA Ratio | 0.9999 | | EMA Ratio | 0.9999 |
| Optimizer | Adam | | Optimizer | Adam | | Optimizer | Adam |
| Learning Rate | 1e-4 | | Learning Rate | 1e-4 | | Learning Rate | 1e-4 |
| Weight Decay | 0 | | Weight Decay | 0 | | Weight Decay | 0 |
| Patch Size | 2 | | Patch Size | 2 | | Patch Size | 16 |
| Backbone | DiT-B/2 | | Backbone | DiT-B/2 | | Backbone | JiT-B/16 |

*Table 8.* Latent-Based CelebA-HQ.     *Table 9.* Latent-Based ImageNet.     *Table 10.* Pixel-Based CelebA-HQ.

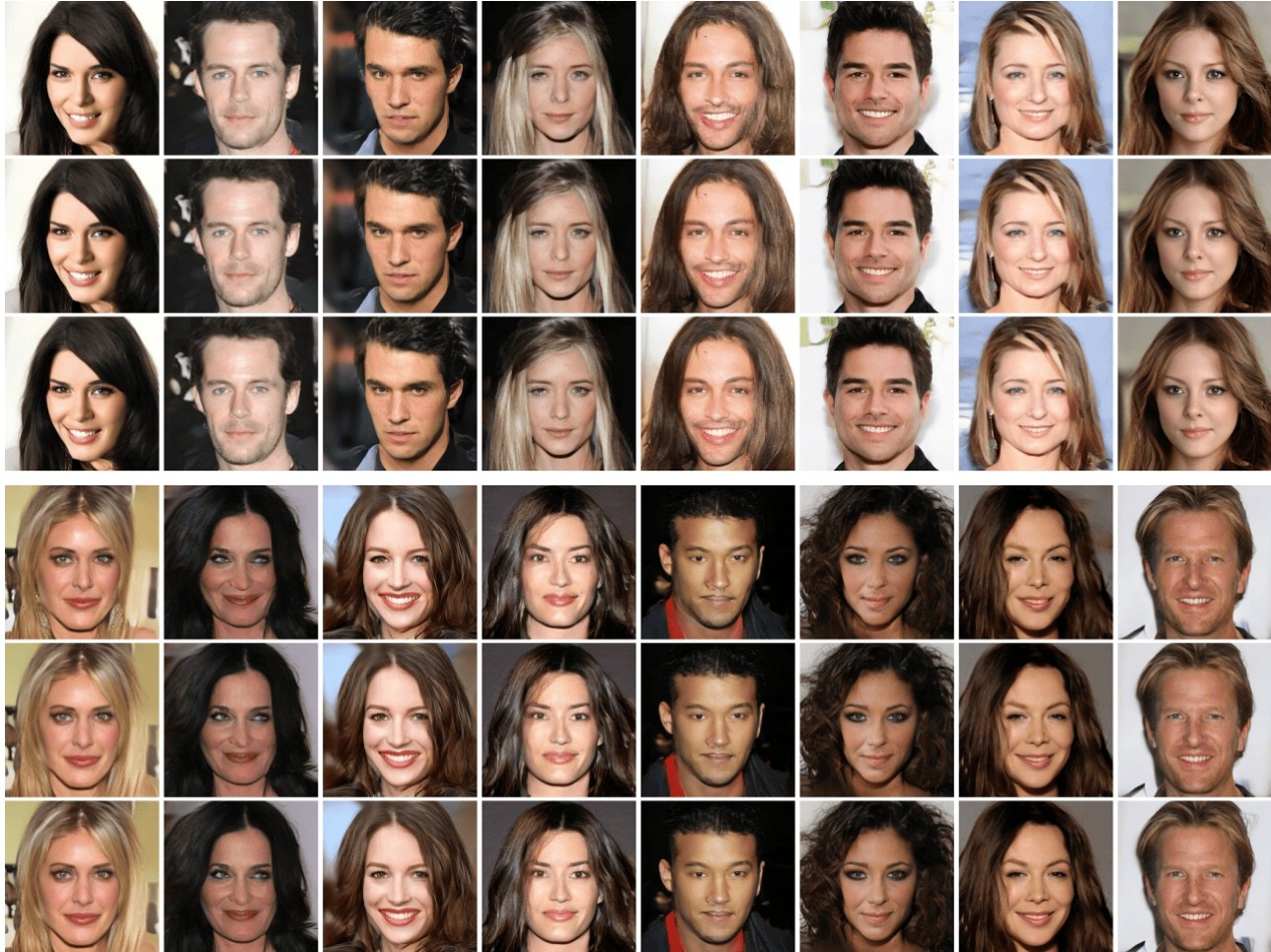

*Figure 5.* Latent image unconditional generation result trained on CelebA-HQ dataset (Liu et al., 2015). First, second and third rows shows 1-step, 2-steps and 4-steps generation respectively.

**Datasets** We trained DiT-B/2 on two image datasets: ImageNet-1000 (Deng et al., 2009) and CelebA-HQ (Liu et al., 2015). ImageNet-1000 contains approximately 1.28M training images and 50K validation images spanning 1,000 object categories. CelebA-HQ contains 30,000 high-resolution human face images derived from CelebA. All dataset are resized to a resolution of $256 \times 256$.

**Metric** To evaluate generative performance, we generate 50K samples for each trained model and compare them against the corresponding real datasets. We report the Fréchet Inception Distance (FID) (Heusel et al., 2017) computed using Inception-V3 features. We follow the same evaluation protocol as in (Geng et al., 2025a) for FID computation.

**Result** For qualitative evaluation, we present unconditional 1-, 2-, and 4-step generation results on CelebA-HQ in Figures 5 and 6. We also show conditional 1-, 2-, and 4-step generation results on ImageNet in Figures 24–25, using the image category as the guidance condition. In both cases, we see our 1-step generation result give reasonably good result comparing against few-step generations. For quantitative comparison, we report FID scores for both datasets in Table 3 where our method achieves the best result overall.

### C.3. Pixel Space Image Generation

**Model** We conduct pixel-space image generation experiments using the Just Image Transformers (JiT) architecture (Li & He, 2025), training the JiT-B/16 model on CelebA-HQ. Conceptually, JiT is a plain Vision Transformer (ViT) applied to patches of pixels without latent encoding. An input image of resolution $H \times W \times C$ is divided into non-overlapping $p \times p$ patches, producing a sequence of patch tokens. To ensure sufficient capacity to model high-dimensional images, JiT uses

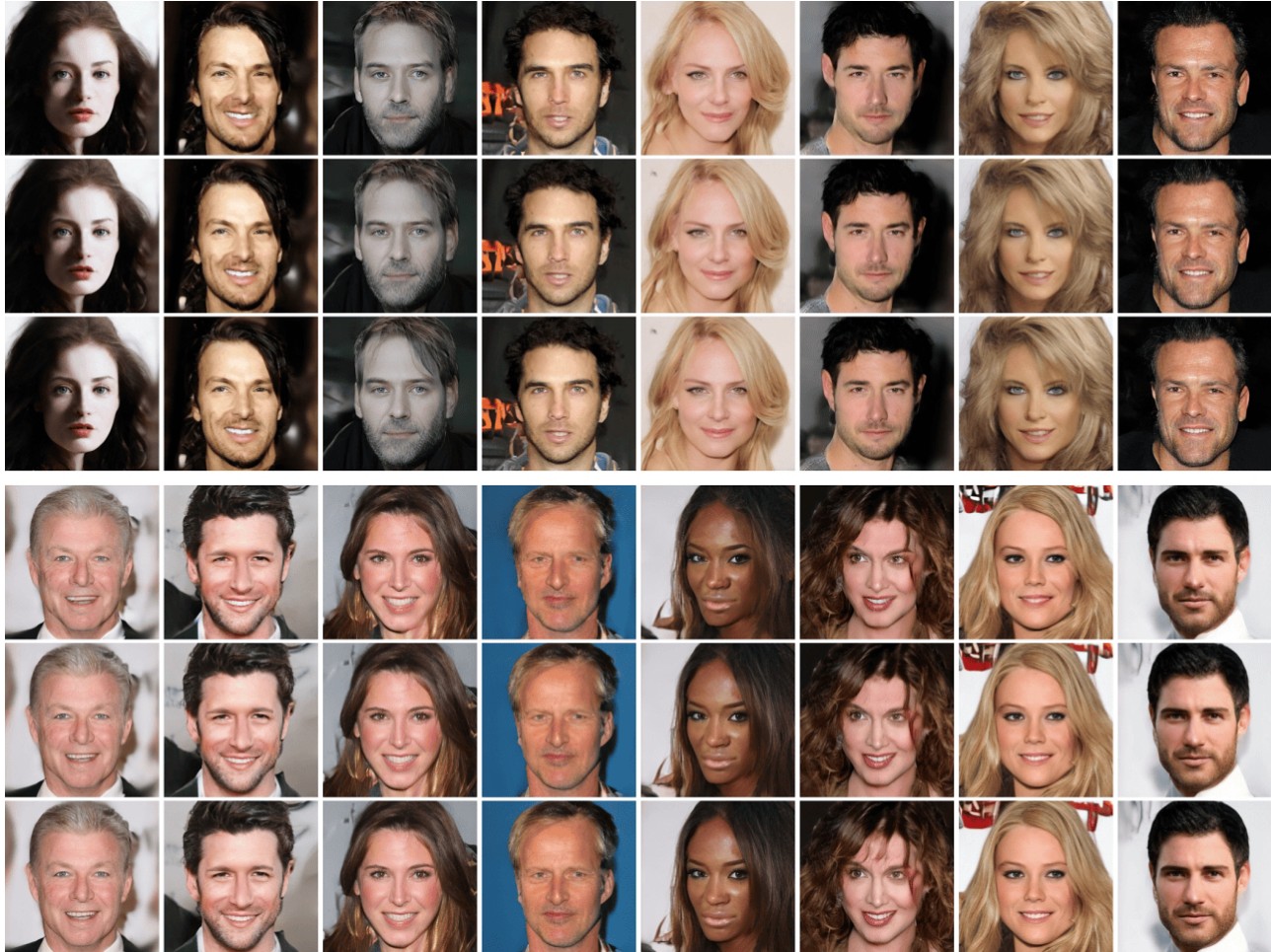

*Figure 6.* Latent image unconditional generation result trained on CelebA-HQ dataset (Liu et al., 2015). First, second and third rows shows 1-step, 2-steps and 4-steps generation respectively.

large patch size ($p = 16$) to balance spatial token length and per-token dimensionality. Each patch token, of dimensionality $p^2C$, is linearly embedded and combined with sinusoidal positional embeddings before being processed by a stack of Transformer blocks. For conditioning on time $t$, $r$ and class information (when applicable), JiT uses AdaLN-Zero similar to DiT. The output tokens are projected back to patch RGB values to reconstruct the full high-resolution image. See Table 10 for hyperparameters in detail.

**Result**    Unconditional pixel-space generation results using the JiT architecture combined with our EMF method, trained on CelebA-HQ with the $x_1$-prediction objective, are shown in Figures 8 and 9. Our method maintains consistent visual quality across 1-, 2-, and 4-step sampling. We further verify that the $x_1$-prediction objective is essential: when trained with the $u$-prediction objective, the generated images remain noisy even as the number of inference steps increases (Figure 7). Quantitative results are reported in Table 6.

### C.4. Functional Image Generation

**Model**    We build upon the Infty-Diff architecture (Bond-Taylor & Willcocks, 2024), which models both inputs and outputs as continuous image functions represented by randomly sampled pixel coordinates. As shown in Figure 12, the network adopts a hybrid sparse–dense design composed of a Sparse Neural Operator and a Dense U-Net to support learning from sparse functional observations. The Sparse Neural Operator first embeds irregularly sampled pixels into feature vectors. These features are interpolated onto a coarse dense grid using KNN interpolation with neighborhood size 3, enabling subsequent dense processing. A U-Net is then applied on a $128 \times 128$ grid for $256 \times 256$ images, with 128 base channels and five resolution stages with channel multipliers $[1, 2, 4, 8, 8]$. Self-attention blocks are inserted at the $16 \times 16$ and $8 \times 8$

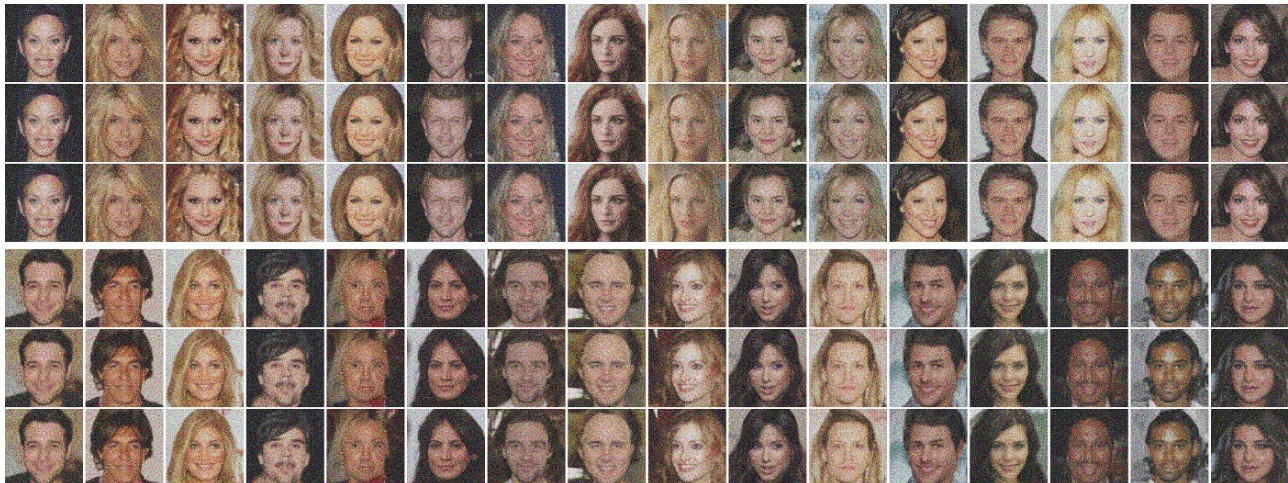

*Figure 7.* We show u-prediction of JiT will fail to generate clean images

*Figure 8.* JiT image unconditional generation result trained on CelebA-HQ dataset (Liu et al., 2015). First, second and third rows shows 1-step, 2-steps and 4-steps generation respectively.

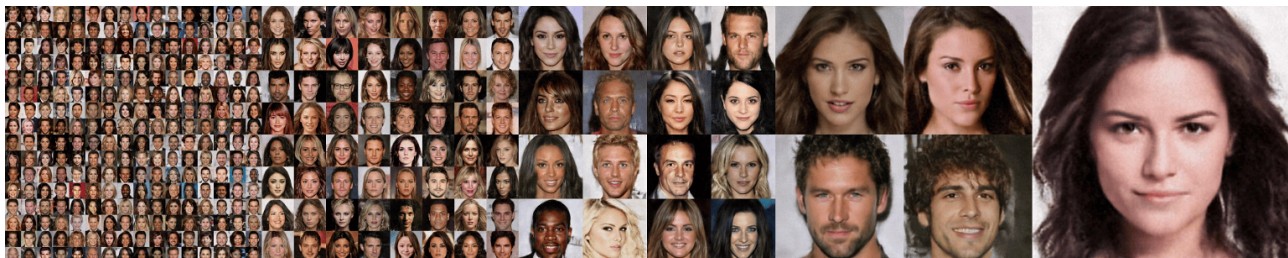

*Figure 9.* JiT image unconditional generation result trained on CelebA-HQ dataset (Liu et al., 2015). First, second and third rows shows 1-step, 2-steps and 4-steps generation respectively.

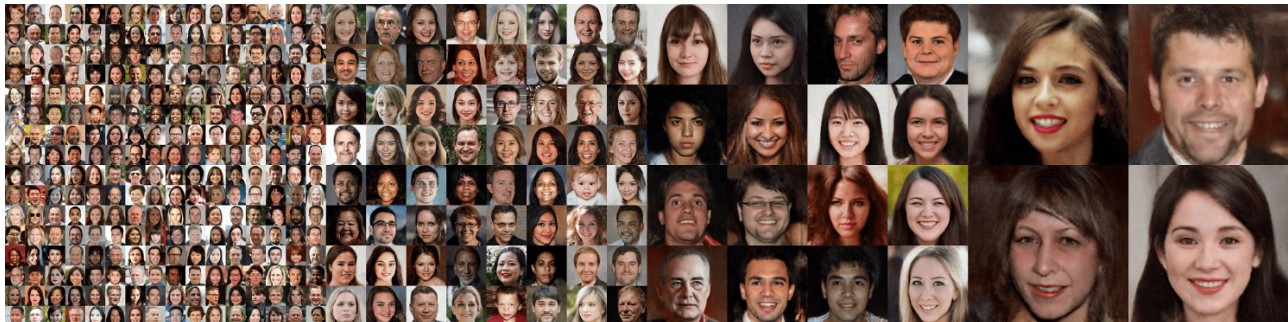

*Figure 10.* 1-step functional generation for images on CelebA-HQ (Liu et al., 2015) dataset. From left to right, we show images generated at 64, 128, 256, 512 and 1024n resolution respectively.

*Figure 11.* 1-step functional generation for images on FFHQ (Karras et al., 2019) dataset. From left to right, we show images generated at 64, 128, 256 and 512 resolution respectively.

resolutions to enhance global context modeling. The dense features are subsequently mapped back to the original coordinate set via inverse KNN interpolation and further refined by a second Sparse Neural Operator, with a residual connection applied to the initial sparse features.

Following Infty-Diff, we implement the Sparse Neural Operator using linear-kernel sparse convolutions with TorchSparse for efficiency. Each Sparse Operator module is composed of five convolutional layers in sequence. It begins with a pointwise convolution, followed by three linear-kernel operator layers. Each operator layer applies a sparse depthwise convolution with 64 channels and a kernel size of 7 (for $256 \times 256$-resolution images), and is followed by two pointwise convolutions with 128 hidden channels to mix channel-wise information. A final pointwise convolution projects the features to the output dimension. Time conditioning is incorporated in both the sparse and dense components using sinusoidal positional

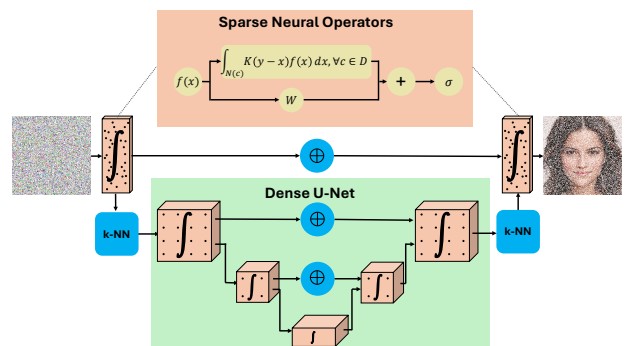

Figure 12. The network design of Infty-Diff.

embeddings (Vaswani et al., 2017), following the Mean Flow formulation (Geng et al., 2025a). The embeddings of $t$ and $r$ are summed and injected in place of the original time conditioning used in Infty-Diff. The resulting model contains approximately 420M trainable parameters.

**Dataset** We conduct experiments on two image datasets: FFHQ (Karras et al., 2019) and CelebA-HQ. FFHQ contains 70000 diverse face images. All images are resized to $256 \times 256$. Following Infty-Diff (Bond-Taylor & Willcocks, 2024), we randomly sample 25% of image pixels during training to evaluate functional-based generation.

**Result** For 2D functional image generation, we present qualitative results in Figure 11 for 1-step unconditional generation on FFHQ, and in Figure 10 for 1-step unconditional generation on CelebA-HQ. Following Infty-Diff, we use $\text{FID}_{\text{CLIP}}$ (Kynkäänniemi et al., 2022) metric to assess function-based generative methods. Because our model generates a continuous function that represents an image, the output is resolution-agnostic. We therefore visualize samples at multiple resolutions, ranging from 64 to 512 on FFHQ and from 64 to 1024 on CelebA-HQ. For quantitative evaluation, Table 11 compares our method against prior approaches; despite using a single sampling step, our results are comparable to multi-step methods such as $\infty$-Diff.

### C.5. SDF Generation

**Model** We adopt the Functional Diffusion architecture (Zhang & Wonka, 2024) for signed distance field (SDF) generation. A SDF represents a shape as a continuous scalar function whose value at each spatial location equals the signed distance to the closest surface, with the sign indicating whether the point lies inside or outside the shape. Both inputs and outputs of the model are specified by randomly sampled points and their corresponding function values, rather than fixed grids. Concretely, the input function $f_c$ is given by a set of context points $\{x_c^i\}_{i=1}^n$ with values $v_c^i = f_c(x_c^i)$, while the output function $f_q$ is queried at locations $\{x_q^j\}_{j=1}^m$ to produce values $\{v_q^j\}_{j=1}^m$. This formulation naturally supports mismatched context and query sets, enabling flexible functional mappings. Following (Zhang & Wonka, 2024), the context set is evenly divided into $d$ disjoint groups. As shown in Figure 13, each group is processed sequentially by an attention block composed of cross-attention followed by self-attention. The cross-attention uses a latent vector to aggregate information from each context group, where the latent is initialized as a learnable variable representing the underlying function and is propagated across blocks. Context points are embedded by combining Fourier positional encodings of spatial coordinates

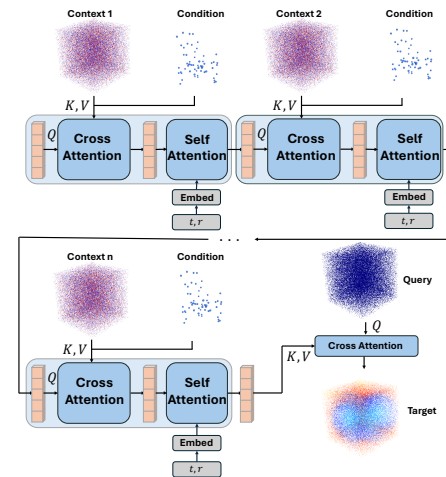

Figure 13. The network design of Functional Diffusion.

with embeddings of function values, and further concatenated with conditional embeddings. In our experiments, conditioning is provided by 64 partially observed surface points.

*Table 11.* Evaluation of FID$_{\text{CLIP}}$ (Kynkäänniemi et al., 2022) against previous infinite-dimensional approaches trained on coordinate subsets. The best results for the 1-step and multi-step settings are highlighted in bold. $^*$ indicates missing entries, where FID scores are reported instead of FID$_{\text{CLIP}}$.

| Method | Step | CelebAHQ-64 | CelebAHQ-128 | FFHQ-256 |
|---|---|---|---|---|
| D2F (Dupont et al., 2022a) | 1 | 40.4$^*$ | – | – |
| GEM (Du et al., 2021) | 1 | 14.65 | 23.73 | 35.62 |
| GASP (Dupont et al., 2022b) | 1 | 9.29 | 27.31 | 24.37 |
| EMF (Ours) | 1 | **4.32** | **8.86** | **15.0** |
| ∞-Diff (Bond-Taylor & Willcocks, 2024) | 100 | **4.57** | **3.02** | **3.87** |
| DPF (Zhuang et al., 2023) | 1000 | 13.21$^*$ | – | – |

**Dataset**   We follow the surface reconstruction setting of Functional Diffusion (Zhang & Wonka, 2024), where the model reconstructs a complete surface from 64 observed points sampled on a target shape. The generative process is conditioned on these surface points and predicts the full SDF starting from noise. All experiments are conducted on the ShapeNet-CoreV2 dataset (Chang et al., 2015), which contains approximately 57000 3D models spanning 55 object categories. Using the same preprocessing pipeline as prior work (Zhang & Wonka, 2024; Zhang et al., 2023; 2022), each mesh is converted into an SDF defined over the domain $[0, 1]^3$. For each shape, we uniformly sample $n = 49152$ points to form the context set and their SDF values, and independently sample $m = 2048$ points as query locations with corresponding SDF values. In addition, a separate set of 64 surface points near the zero-level set is sampled and used as conditional input.

**Metrics**   We evaluate reconstructed SDF quality using Chamfer Distance, F-score, and Boundary Loss, following prior work (Zhang & Wonka, 2024; Zhang et al., 2023; 2022). Chamfer Distance (CD) and F-score are computed by uniformly sampling 50K points from each reconstructed surface. F-Score evaluates surface reconstruction quality by measuring the precision–recall trade-off between generated and ground-truth surface points under a fixed distance threshold. It quantifies how well the predicted surface aligns with the true surface by penalizing both missing regions and spurious geometry. Boundary Loss measures SDF accuracy near the surface boundary and is defined as $\text{Boundary}(f) = \frac{1}{|\mathcal{E}_\Omega|} \sum_{i \in \mathcal{E}_\Omega} |f(\mathbf{x}_i) - q(\mathbf{x}_i)|^2$, where $\mathcal{E}_\Omega$ denotes points sampled near the zero-level set, $f$ is the predicted SDF, and $q$ is the ground-truth SDF. This metric is computed using 100K boundary samples. We use the same train/test split as (Zhang & Wonka, 2024) for our experiment.

**Result**   For SDF generation, we train our model on ShapeNet using only 64 surface points as conditioning input. This sparse-conditioning setting is challenging, particularly for single-step generation. Qualitative results for 1-step conditional generation are shown in Figures 14 and 15. For quantitative evaluation, Table 7 reports 3D reconstruction metrics; our method achieves quality comparable to multi-step approaches and consistently outperforms the original mean flow method.

### C.6. Point Cloud Generation

**Model**   We adopt the Latent Point Diffusion Model (LION) architecture (Vahdat et al., 2022) for point cloud generation, which performs generative modeling in a structured latent space derived from point clouds. As shown Figure 16 in The model builds upon a variational autoencoder that encodes each shape into a hierarchical latent representation consisting of a global shape latent and a point-structured latent point cloud, capturing coarse structure and fine-grained geometry, respectively. The encoder, decoder, and latent point diffusion modules are implemented with Point-Voxel CNNs (PVCNNs) (Liu et al., 2019), following the design of (Zhou et al., 2021). The global latent diffusion model is parameterized by a ResNet-style network composed of fully connected layers, imple-

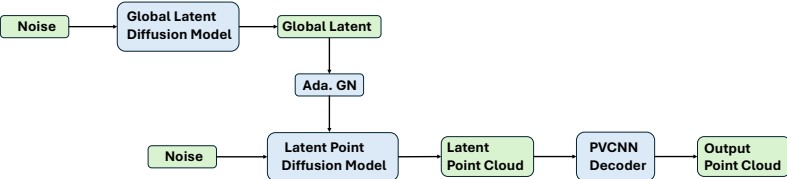

*Figure 16.* The network design of LION.

mented as $1 \times 1$ convolutions. Conditioning on the global latent is injected into the PVCNN layers to generate point-structured latent point cloud through adaptive Group Normalization. For modeling the point-structured latent representations, we further adopt a modified DiT-3D backbone based on (Wang et al., 2025), which provides stronger modeling capacity and improved scalability. Finally, the decoder maps the generated latent representation back to the 3D space, yielding the output point cloud.

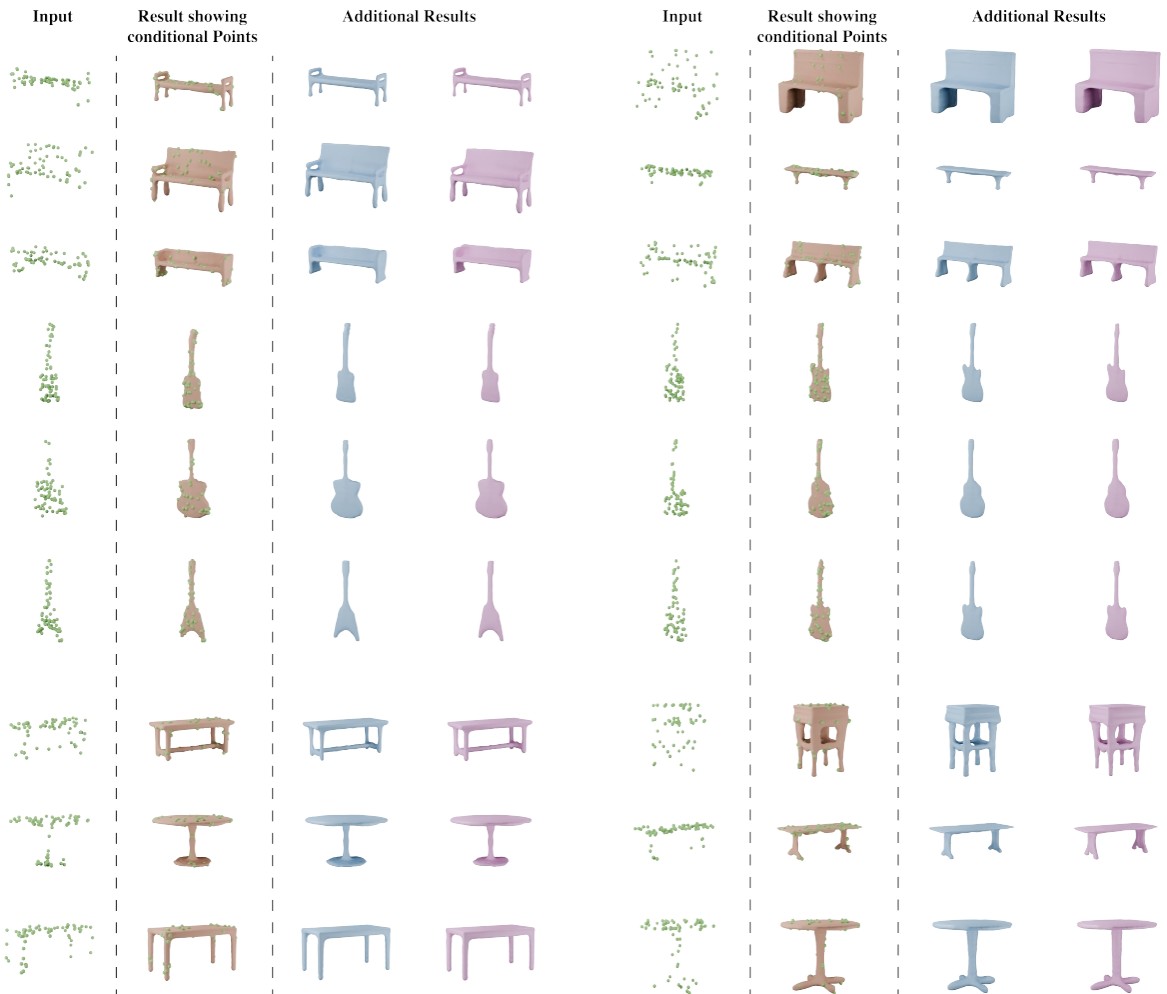

*Figure 14.* We show 1-step functional SDF generation results. The leftmost column visualizes the conditioning points, overlaid on the first generated mesh shown in the second column. Additional generated samples are presented in the remaining columns.

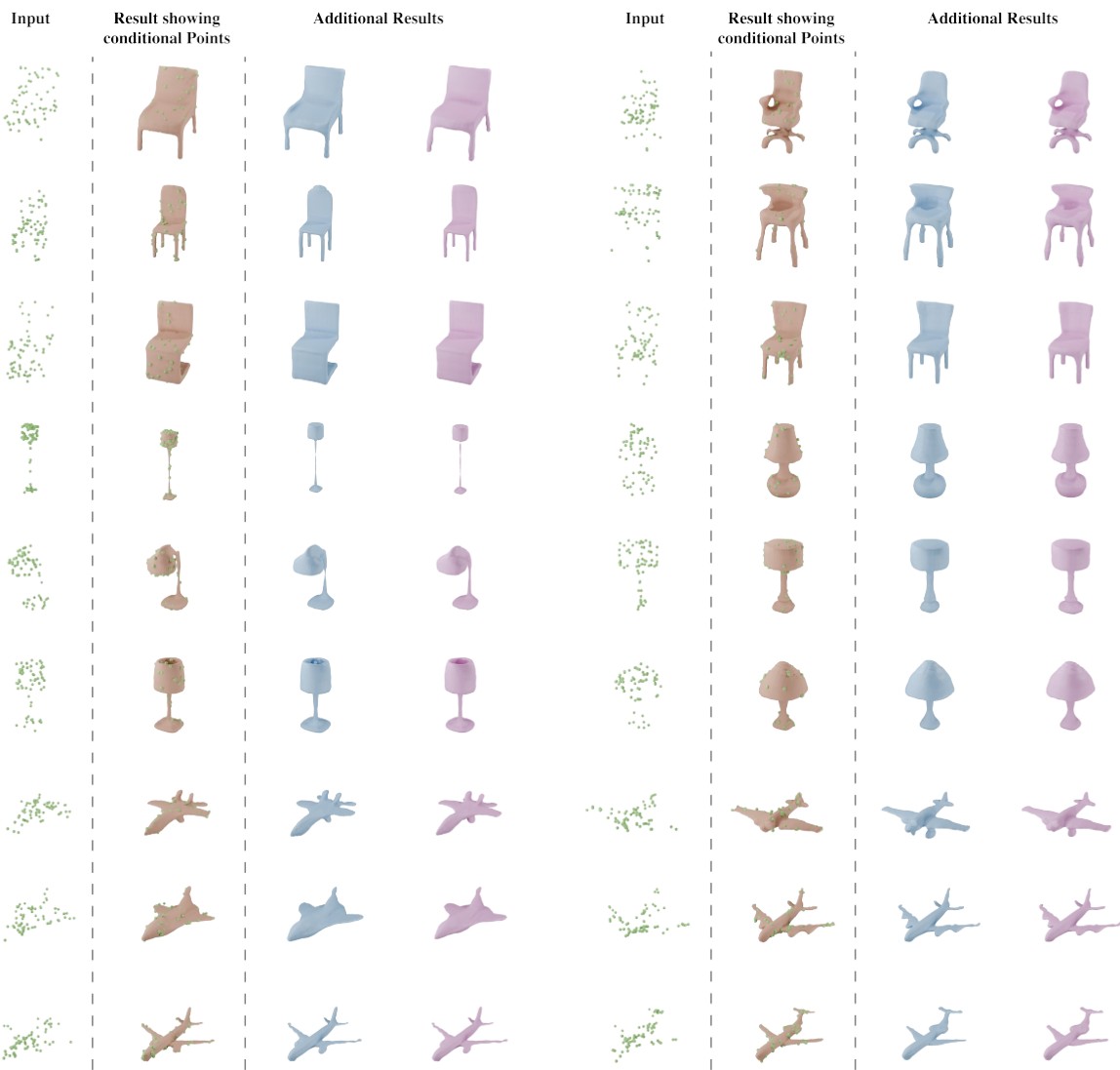

*Figure 15.* We show 1-step functional SDF generation results. The leftmost column visualizes the conditioning points, overlaid on the first generated mesh shown in the second column. Additional generated samples are presented in the remaining columns.

**Dataset**   We conduct experiments on the ShapeNet dataset (Chang et al., 2015) using the preprocessing and data splits provided by PointFlow (Yang et al., 2019). We focus our evaluation on two object categories: airplanes and chairs. Each shape in the processed dataset contains 15,000 points, from which 2,048 points are randomly sampled at every training iteration. The training set includes 2,832 airplane shapes and 4,612 chair shapes. For evaluation, we report sample quality metrics against the corresponding reference sets, which comprise 405 for airplanes and 662 for chairs. Following PointFlow, all shapes are normalized using a global normalization scheme, where the mean is computed per axis over the entire training set and a single standard deviation is applied across all axes.

**Metrics**   To assess the performance of point cloud generative models at the distribution level, we compare a generated set $S_g$ with a reference set $S_r$ with Coverage (COV) and 1-Nearest Neighbor Accuracy (1-NNA), both of which rely on a pairwise distance defined between point clouds.

Coverage (COV) measures the extent to which the generated samples span the variability of the reference distribution. Specifically, each reference shape is associated with its closest counterpart in the generated set, and COV is defined as the fraction of generated shapes that are selected as nearest neighbors by at least one reference shape. As a result, COV primarily reflects sample diversity and sensitivity to mode collapse, while being largely agnostic to the fidelity of individual generated point clouds.

$$\text{COV}(S_g, S_r) = \frac{|\{\arg\min_{Y \in S_r} D(X, Y) \mid X \in S_g\}|}{|S_r|} \tag{56}$$

1-Nearest Neighbor Accuracy (1-NNA) evaluates how well the generated and reference distributions are aligned. This metric treats the union of $S_g$ and $S_r$ as a labeled dataset and computes the leave-one-out accuracy of a 1-NN classifier, where each sample is assigned the label of its nearest neighbor.

$$\text{1-NNA}(S_g, S_r) = \frac{\sum_{X \in S_g} \mathbf{1}[N_X \in S_g] + \sum_{Y \in S_r} \mathbf{1}[N_Y \in S_r]}{|S_g| + |S_r|}, \tag{57}$$

where $N_X$ (resp., $N_Y$) denotes the nearest neighbor of $X$ (resp., $Y$) in $(S_g \cup S_r) \setminus \{X\}$.

For both COV and 1-NNA, nearest neighbors are determined using either the Chamfer Distance (CD) or the Earth Mover's Distance (EMD). CD evaluates mutual proximity by aggregating point-to-set nearest-neighbor distances in both directions, while EMD computes the minimal transport cost between two point clouds by enforcing a one-to-one correspondence. CD and EMD are defined as:

$$\text{CD}(X, Y) = \sum_{x \in X} \min_{y \in Y} \|x - y\|_2^2 + \sum_{y \in Y} \min_{x \in X} \|x - y\|_2^2, \tag{58}$$

$$\text{EMD}(X, Y) = \min_{\gamma: X \to Y} \sum_{x \in X} \|x - \gamma(x)\|_2, \tag{59}$$

where $X$ and $Y$ denote two point clouds with the same cardinality, $\|\cdot\|_2$ is the Euclidean norm, and $\gamma$ is a bijection between points in $X$ and $Y$.

**Result**   For point cloud generation, we present 1-step unconditional samples for two ShapeNet categories in Figure 17. All models are trained on ShapeNet using the LION architecture. Quantitative results are reported in Table 12, where our method achieves the best generation quality compared to prior approaches.

## D. Additional Results&Experiments

### D.1. Ablation Study: Rationale for the Second Local Linear Approximation

In Equation 10, during the derivation, we apply local linear approximation to the term $u_{t \to t + \Delta t}$ at two different places. The first approximation appears in an independent summation term in Equation 10, where $u_{t \to t + \Delta t}$ is approximated by $u_{t \to t}$. The motivation of this approximation is straightforward: by reducing it to the instantaneous velocity $u_{t \to t}(x)$, we can further replace it with the conditional instantaneous velocity $u_t(x|x_1)$, thereby incorporating explicit supervision from the dataset.

The second approximation is applied to the update $x_{t + \Delta t} = \Delta t u_{t \to t + \Delta t}(x_t) + x_t$, where $u_{t \to t + \Delta t}$ is again approximated

*Table 12.* Unconditional generation results on the Airplane and Chair categories at a resolution of 2048 points. We report one-nearest neighbor accuracy (1-NNA) and coverage (COV) under Chamfer Distance (CD) and Earth Mover's Distance (EMD). For 1-NNA, lower is better (↓), while for COV, higher is better (↑). Bold and underlined numbers indicate the best and second-best performance for each metric under the one-step and multi-step settings, respectively. Global normalization is applied to both training and test sets following LION (Vahdat et al., 2022).

| Method | Steps | Airplane | | | | Chair | | | |
|---|---|---|---|---|---|---|---|---|---|
| | | 1-NNA↓ | | COV↑ | | 1-NNA↓ | | COV↑ | |
| | | CD | EMD | CD | EMD | CD | EMD | CD | EMD |
| MFM-point (Molodyk et al., 2025) | 1400 | 65.36 | **57.21** | – | – | 54.92 | 53.25 | – | – |
| LION (Vahdat et al., 2022) | 1000 | 67.41 | 61.23 | 47.16 | 49.63 | 53.70 | 52.34 | 48.94 | 52.11 |
| FrePoLat (Zhou et al., 2024) | 1000 | 65.25 | 62.10 | 45.16 | 47.80 | 52.35 | 53.23 | 50.28 | 50.93 |
| NSOT (Hui et al., 2025) | 1000 | 68.64 | 61.85 | – | – | 55.51 | 57.63 | – | – |
| DiT-3D (Mo et al., 2023) | 1000 | **62.35** | 58.67 | 53.16 | 54.39 | **49.11** | **50.73** | 50.00 | 56.38 |
| PVD (Zhou et al., 2021) | 1000 | 73.82 | 64.81 | 48.88 | 52.09 | 56.26 | 53.32 | 49.84 | 50.60 |
| PVD-DDIM (Zhou et al., 2021) | 100 | 76.21 | 69.84 | 44.23 | 49.75 | 61.54 | 57.73 | 46.32 | 48.19 |
| DPM (Luo & Hu, 2021) | 100 | 76.42 | 86.91 | 48.64 | 33.83 | 60.05 | 74.77 | 44.86 | 35.50 |
| ShapeGF (Cai et al., 2020) | 10 | 80.00 | 76.17 | 45.19 | 40.25 | 68.96 | 65.48 | 48.34 | 44.26 |
| PSF (Wu et al., 2023) | 1 | **71.11** | **61.09** | 46.17 | 52.59 | 58.92 | 54.45 | 46.71 | 49.84 |
| r-GAN (Achlioptas et al., 2018) | 1 | 98.40 | 96.79 | 30.12 | 14.32 | 83.69 | 99.70 | 24.27 | 15.13 |
| 1-GAN (CD) (Achlioptas et al., 2018) | 1 | 87.30 | 93.95 | 38.52 | 21.23 | 68.58 | 83.84 | 41.99 | 29.31 |
| 1-GAN (EMD) (Achlioptas et al., 2018) | 1 | 89.49 | 76.91 | 38.27 | 38.52 | 71.90 | 64.65 | 38.07 | 44.86 |
| PointFlow (Yang et al., 2019) | 1 | 75.68 | 70.74 | 47.90 | 46.41 | 62.84 | 60.57 | 42.90 | 50.00 |
| DPF-Net (Klokov et al., 2020) | 1 | 75.18 | 65.55 | 46.17 | 48.89 | 62.00 | 58.53 | 44.71 | 48.79 |
| SoftFlow (Kim et al., 2020) | 1 | 76.05 | 65.80 | 46.91 | 47.90 | 59.21 | 60.05 | 41.39 | 47.43 |
| SetVAE (Kim et al., 2021) | 1 | 75.31 | 77.65 | 43.70 | 48.40 | 58.76 | 61.48 | 46.83 | 44.26 |
| EMF (ours) | 1 | 72.84 | 62.72 | **50.37** | **55.56** | **56.42** | **54.08** | **47.89** | **52.87** |

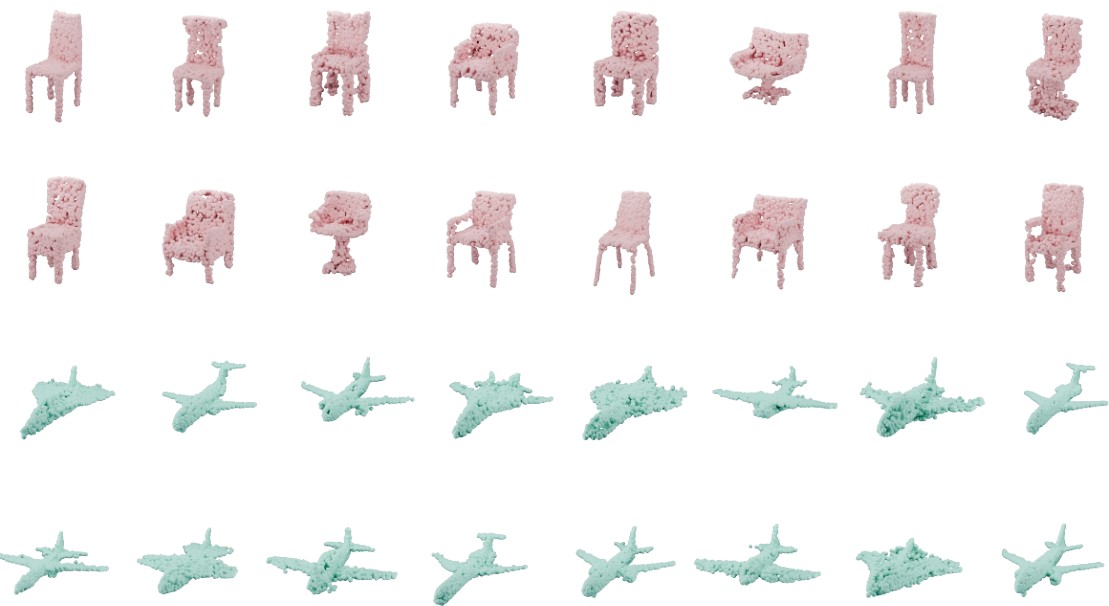

*Figure 17.* 1-step point cloud generation result trained on ShapeNet(Chang et al., 2015) dataset.

*Table 13.* Comparison of memory and computational cost between our method and MeanFlow for unconditional (CelebA-HQ) and conditional (ImageNet-1000) generation using the DiT-B/2 model. "Peak" denotes the maximum GPU memory usage during training, "Fixed" refers to the constant memory overhead, and aux-EMF indicates our method with a 4-block auxiliary head. All experiments are conducted on a single H200 GPU with batch sizes of 64 for CelebA-HQ and 128 for ImageNet-1000, using EMA and AdamW optimization with mixed-precision (FP16) training in PyTorch.

| Method | Dataset | Peak Memory | Fixed Memory | Speed / Iter | FID |
|---|---|---|---|---|---|
| MeanFlow | CelebA-HQ | 32.1GB | 2.3GB | 151.4ms | 12.4 |
| EMF (compute $u_{t\to t+\Delta t}$) | CelebA-HQ | 23.3GB | 2.3GB | 91.74ms | 11.2 |
| EMF | CelebA-HQ | 23.3GB | 2.3GB | 91.2 ms | **10.9** |
| aux-EMF | CelebA-HQ | **17.6GB** | 2.8 GB | **84.2 ms** | 11.7 |
| MeanFlow | ImageNet | 101.9GB | 2.4GB | 400.9ms | 11.1 |
| EMF (compute $u_{t\to t+\Delta t}$) | ImageNet | 71.7GB | 2.4GB | 232.6ms | - |
| EMF | ImageNet | **57.9GB** | 2.4GB | **198.8ms** | 7.2 |

by $u_{t\to t}$. This approximation is primarily introduced for memory efficiency. This design choice is particularly important for conditional generation. During training, conditional MeanFlow employs CFG, which replaces $u(x|x_1)$ with $wu(x|x_1) + (1-w-k)u_{t\to t}^\theta(x_t, C_0) + ku_{t\to t}^\theta(x_t, C)$, where $C$ denotes the label of $x_1$ and $C_0$ is the null label. In this formulation, the term $u_{t\to t}^\theta(x_t, C)$ can be directly reused in the computation of $x_{t+\Delta t} \approx \Delta t u_{t\to t}(x_t) + x_t$, which helps reduce memory consumption. For unconditional generation, although the computation of $x_{t+\Delta t}$ requires two stop-gradient forward passes and one trainable forward pass regardless of whether $u_{t\to t+\Delta t}$ is approximated, we empirically observe that using the exact $u_{t\to t+\Delta t}$ does not improve generation quality, and moreover it prevents the use of the multi-head technique described in Figure 3, leading to increased memory usage and computational cost. Quantitative results are reported in Table 13.

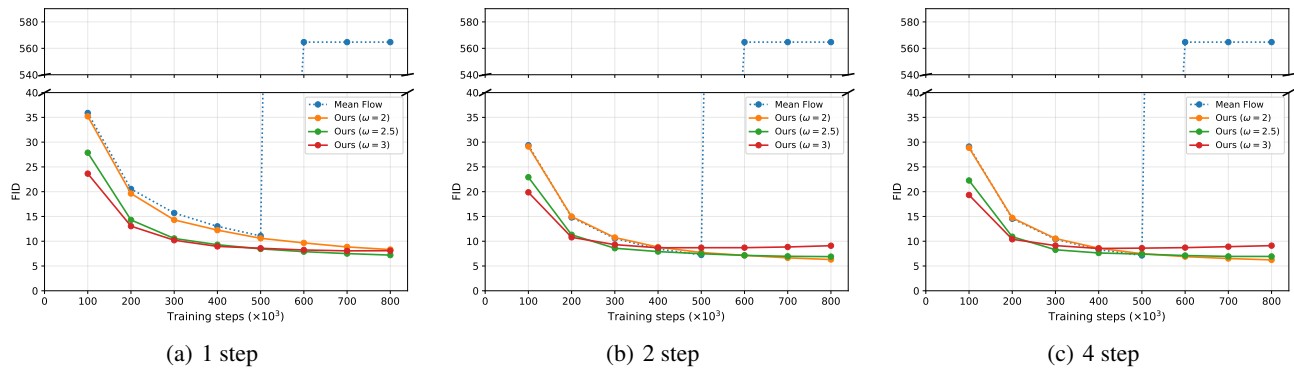

*Figure 18.* Training dynamics of our method and MeanFlow on ImageNet under 1-step, 2-step, and 4-step generation. For our method, we compare different classifier-free guidance (CFG) scales, while MeanFlow uses a fixed CFG scale of 2.0, which is the best-performing setting reported in (Geng et al., 2025a).

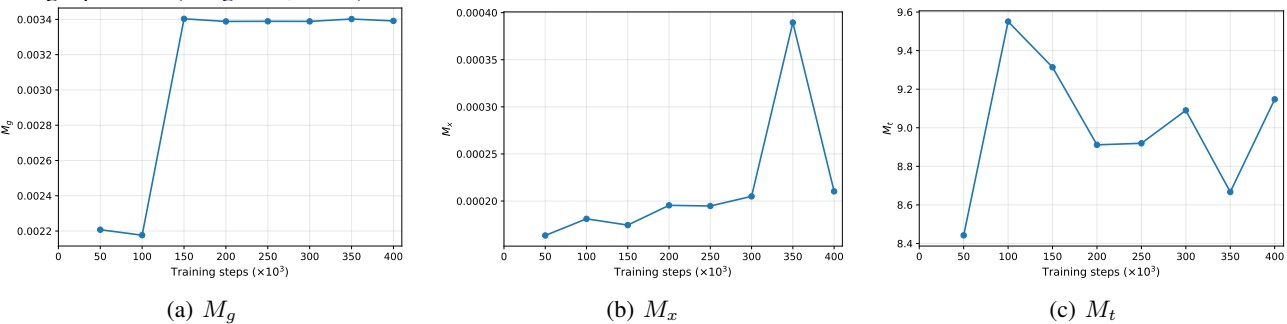

*Figure 19.* Evolution of the parameters $M_g$, $M_t$, and $M_x$ in Assumption 1 over the course of training.

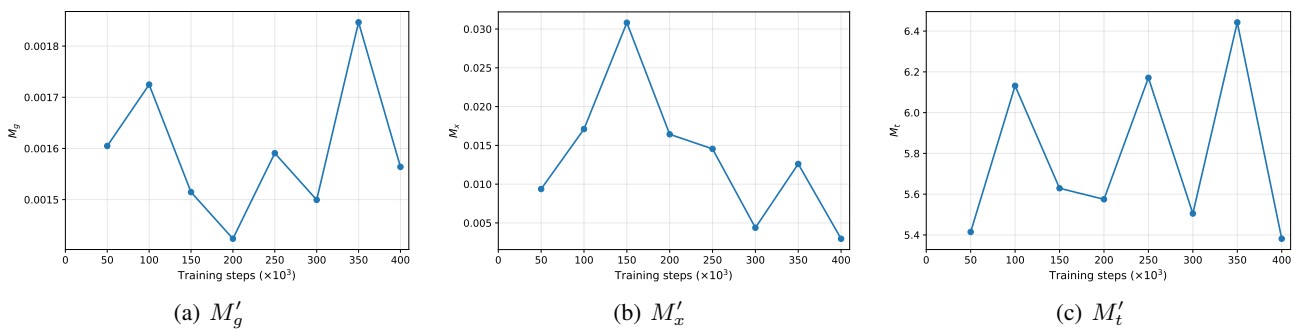

(a) $M_g'$           (b) $M_x'$           (c) $M_t'$

*Figure 20.* Evolution of the parameters $M_g'$, $M_t'$, and $M_x'$ in Assumption 2 over the course of training.

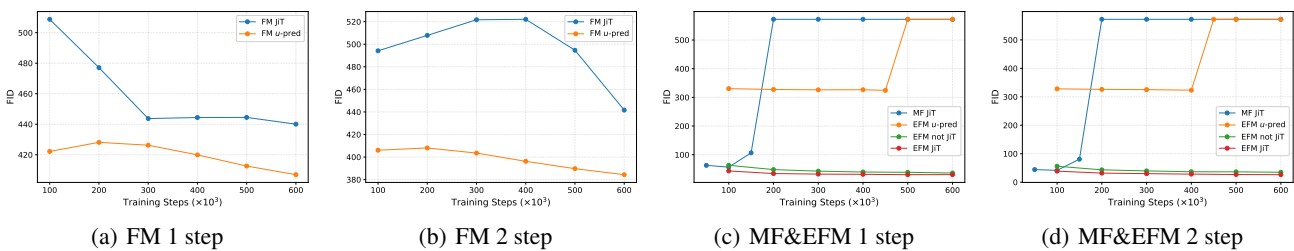

(a) FM 1 step      (b) FM 2 step      (c) MF&EFM 1 step      (d) MF&EFM 2 step

*Figure 21.* Training dynamics of FID for different JiT-based generative methods. We compare JiT-style flow matching (i.e., $x_1$-prediction FM with $u$-loss), $u$-prediction flow matching, JiT-style MeanFlow, $u$-prediction EMF, JiT-style EMF, and $x_1$-prediction EMF. Even with the JiT formulation, MeanFlow remains unstable, highlighting its inherent optimization difficulty. Similarly, $u$-prediction EMF still exhibits unstable behavior, indicating that the $x_1$-prediction variant is essential for stable and reliable training.

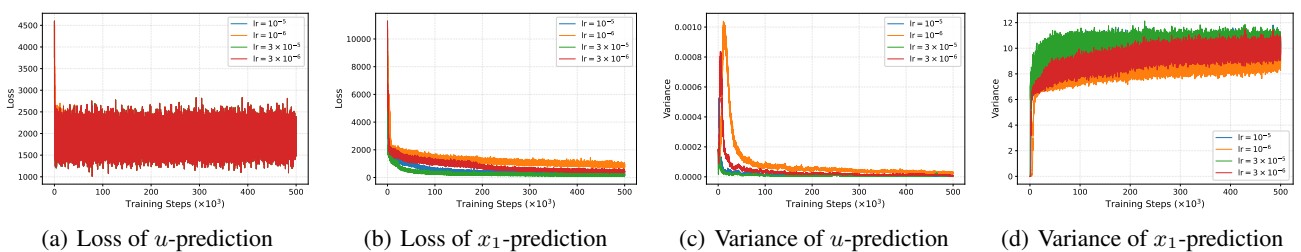

(a) Loss of $u$-prediction    (b) Loss of $x_1$-prediction    (c) Variance of $u$-prediction    (d) Variance of $x_1$-prediction

*Figure 22.* Training behavior of $u$-prediction and $x_1$-prediction Flow Matching variants across different learning rates. We report the variance of network outputs, where zero variance indicates a constant SDF field and corresponds to variance collapse. The $u$-prediction variant suffers from spatial variance collapse and unstable optimization, whereas the $x_1$-prediction model exhibits stable variance and smooth training dynamics.

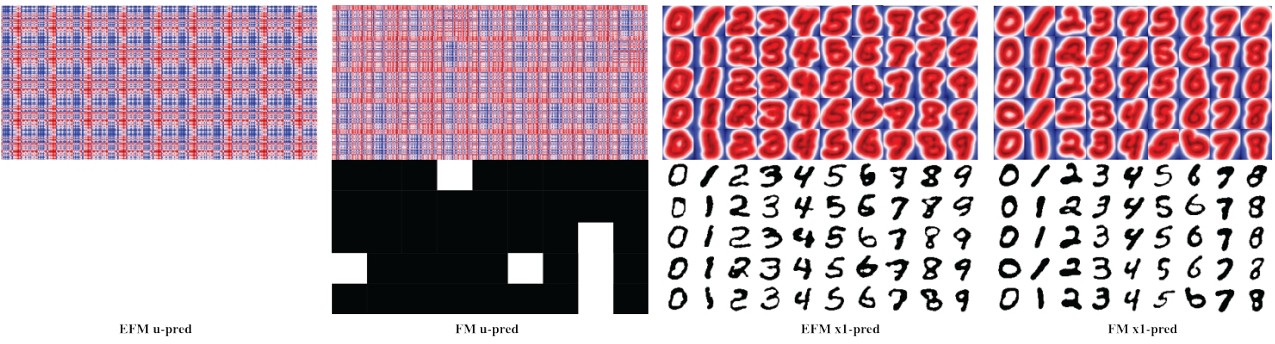

*Figure 23.* Comparison of SDF generation using different prediction variants. Due to spatial variance collapse in $u$-prediction flow matching, the model fails to generate meaningful digit SDFs. Since $u$-prediction EMF relies on accurate estimation of instantaneous velocities, it also fails in this setting. In contrast, $x_1$-prediction flow matching learns reliable dynamics, enabling $x_1$-prediction EMF to successfully generate high-quality SDFs.

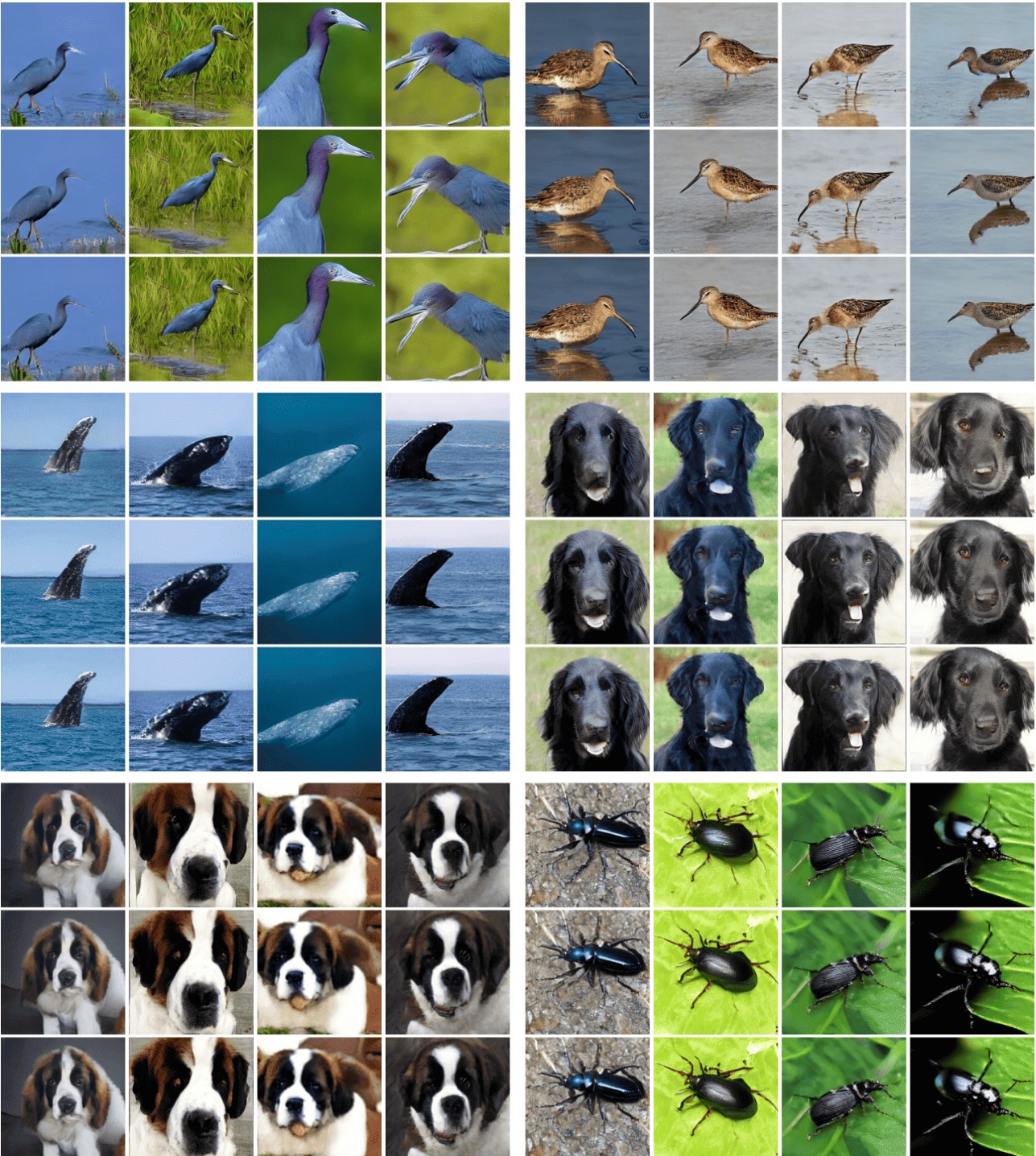

*Figure 24.* Part 1 of ImageNet (Deng et al., 2009) generation result. First, second and third row shows 1-step, 2-steps and 4-steps generation respectively using the same condition.

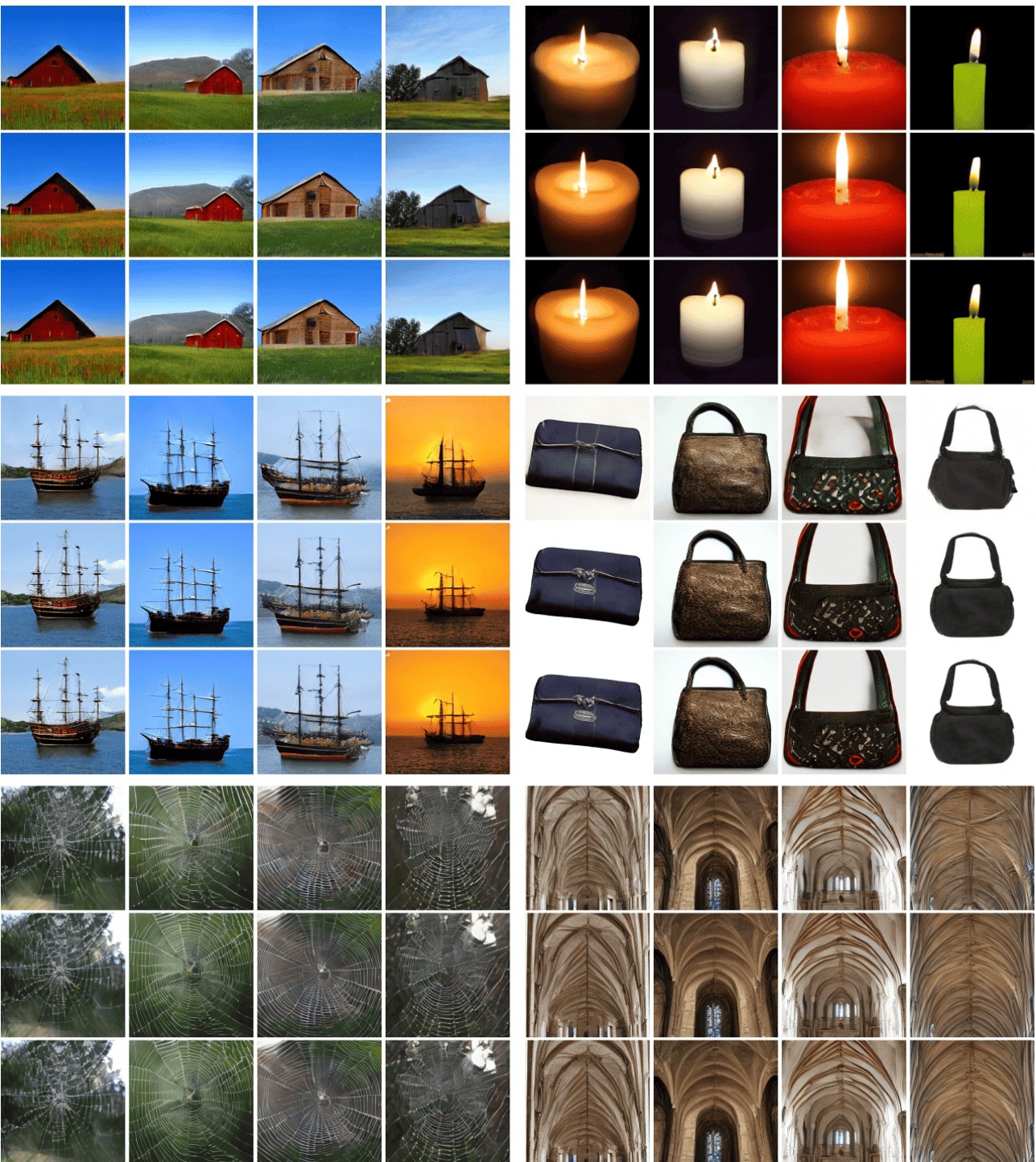

*Figure 25.* Part 2 of ImageNet (Deng et al., 2009) generation result. First, second and third row shows 1-step, 2-steps and 4-steps generation respectively using the same condition.

