# OpenReview forum: "Trajectory Consistency for One-Step Generation on Euler Mean Flows"
_ICML.cc/2026/Conference — ICML 2026 regular_

### Official Review · Reviewer_u3Pu · 2026-03-11

**Soundness:** 3
**Presentation:** 3
**Significance:** 3
**Originality:** 3
**Overall Recommendation:** 4
**Confidence:** 3

**Summary:**

This paper proposes Euler Mean Flows (EMF), a flow-based generative framework designed to achieve high-quality one-step or few-step generation with extremely low sampling costs. The core challenge in one-step or few-step generation lies in learning "trajectory consistency." Traditional flow models lack a reference flow map directly from the data distribution to supervise this long-range trajectory consistency. Leveraging the semigroup property of flow maps, the authors propose a local linear approximation to replace the hard-to-optimize long-range consistency constraints. This linear surrogate enables direct data supervision for long-horizon flow maps without requiring explicit Jacobian-vector product computations (JVP-free). Furthermore, this framework unifies both $u$-prediction and $x_1$-prediction variants. The proposed method demonstrates superior optimization stability and generation quality across image synthesis, particle-based geometry generation, and functional generation tasks. Compared to existing one-step generation methods (e.g., MeanFlow), EMF reduces training time and memory consumption by approximately 50%.

**Compliance With Llm Reviewing Policy:**

Affirmed.

**Final Justification:**

While the authors' responses have clarified most of my points, the presentation of the manuscript and figures is still below the publication standard. Since significant polishing is still needed, I am keeping my rating at Weak Accept.

**Key Questions For Authors:**

None

**Limitations:**

yes

**Strengths And Weaknesses:**

Strengths

- Through a series of theorems (Theorems 4.1-4.4), the paper not only points out the theoretical limitations of traditional conditional flow maps but also rigorously proves the mathematical rationality and validity of the proposed local linear approximation and surrogate loss functions, ensuring logical self-consistency.
- The method is extensively tested across various data modalities. This fully demonstrates the universality, stability, and efficiency of the JVP-free framework across different generative tasks.


Weaknesses

- The figure is overly abstract and fails to clearly illustrate the core mechanism of how EMF avoids JVP computations through "local linearization." It is recommended to add computational graphs or mathematical symbols to intuitively highlight the fundamental differences from prior works.
- Compared to prior works like MeanFlow, the core breakthroughs of this paper—such as eliminating JVP, reducing memory overhead, and supporting $x_1$-prediction—are scattered across different sections. It is recommended to include a centralized comparative summary in the Introduction (e.g., specific metrics on memory and computational complexity) so that readers can quickly grasp the paper's core contributions.

---

> ### Author Rebuttal · Authors · 2026-03-31
>
> We thank the reviewer for this helpful suggestion and agree that two aspects can be made clearer in the current presentation: (1) the figure and related formula-based discussion should more explicitly illustrate how EMF avoids JVPs through local linearization, and (2) the paper would benefit from a centralized comparison that summarizes the core practical advances of EMF over prior work such as MeanFlow. Since the ICML rebuttal does not allow us to upload revised figures or a new PDF, we cannot directly replace these materials here. In the revision, we will address both points by redrawing the figure with clearer symbolic annotations and a more direct comparison to prior JVP-based formulations, and by adding a concise comparative summary in the Introduction with concrete quantitative comparisons.
>
> ## 1. Clarifying the figure and the formula-based discussion
> More concretely, we will revise the figure to explicitly show:
> (1) the trajectory-consistency decomposition $\psi_{t\to r}=\psi_{s\to r}\circ\psi_{t\to s}$ with $s=t+\Delta t$;
> (2) the local linearization $\psi_{t\to s}(x_t)\approx x_t+\Delta t\,u_t(x_t\mid x_1)$; and
> (3) how the conditional velocity $u_t(x_t\mid x_1)$, computed from data samples $x_1\in D$, replaces $u_t(x_t)$ in the local linearization and thereby provides direct data supervision to the long flow map.
>
> We will also revise the related discussion in the main text (currently L169--L184) to make the motivation more explicit through formulas. Specifically, $L^{TC}$ by itself does not provide direct data supervision, since all terms come from the model. It enforces consistency, but does not ensure that the learned long-range flow map matches the target distribution (cf. L149--159, right). We will then more clearly summarize that prior methods fall into two classes with mathematical formulas, extending the discussion in L169--184 (left).
>
> **Progressive Extension** methods (e.g., ShortCut, PSD, SplitMeanFlow) extend instantaneous $u_t$ to long flow maps through consistency. For example, ShortCut first learns $|u^\theta_{t\to t}(x)-u_1(x\mid x_1)|^2$, where $u_1(x\mid x_1)=\frac{x_1-x}{1-t}$ and $x_1\in D$, and then extends to longer intervals via $|u^\theta_{t\to t+2d}(x_t)-\frac{1}{2}(u^\theta_{t\to t+d}(x_t)+u^\theta_{t+d\to t+2d}(x_{t+d}))|^2$. Thus, $u^\theta_{t\to t+2d}(x)$ does not receive direct supervision from $D$; the long flow map is learned from model outputs rather than directly from data supervision.
>
> **Continuous methods** (e.g., MeanFlow, ESD, LSD) directly supervise long flow maps using data-based conditional velocities together with a continuous PDE-based formulation. For example, MeanFlow uses $|u^\theta_{t\to r}(x)-sg[u_t(x\mid x_1)+(r-t)(u_t(x\mid x_1)+\partial_x u^\theta_{t\to r}(x)+\partial_t u^\theta_{t\to r}(x))]|^2$, with $u_t(x\mid x_1)=\frac{x_1-x}{1-t}$ and $x_1\in D$. This gives direct supervision, but requires JVPs, which hurt memory, speed, stability, and practical usability.
>
> By contrast, our method uses $|u^\theta_{t\to r}(x_t)-sg(u_t(x_t\mid x_1)+(r-t)\frac{u^\theta_{t+\Delta t\to r}(x_t+\Delta t\,u^\theta_{t\to t}(x_t))-u^\theta_{t\to r}(x_t)}{\Delta t})|^2$, with the same direct supervision term $u_t(x_t\mid x_1)=\frac{x_1-x_t}{1-t}$, $x_1\in D$, for $u^\theta_{t\to r}(x_t)$, while avoiding JVPs.
>
> We believe these revisions will make both the figure and the conceptual distinction from prior work much clearer at a glance.
>
> ## 2. Centralized comparison with prior work
>
> In the revision, we will add a centralized comparison in the Introduction to clearly summarize the core advances of EMF over prior methods such as MeanFlow. Specifically, we will highlight the practical benefits of removing JVPs, including reduced memory usage, faster training, improved optimization stability, support for $x_1$-prediction, and broader applicability of one-step training. We will also consolidate the main empirical findings currently scattered across different sections, including stronger FID in latent-space one-step generation and, to the best of our knowledge, being the first to support pixel-space one-step generation. More specifically, we will clarify that the improved stability is reflected in more stable one-step training on SDF generation, whereas the broader applicability is reflected in support for point-cloud generation, where sparse C++/CUDA operators commonly used in practice often do not readily support JVPs, making traditional JVP-based methods such as MeanFlow difficult to apply. These points will be summarized together with concrete quantitative comparisons, including specific metrics on memory and computational complexity, so that readers can more quickly grasp the paper’s core contributions.

---

> > ### Author Rebuttal · Reviewer_u3Pu · 2026-04-02
> >
> > I would like to thank the authors for their response, which has addressed most of my concerns. The comparison between this work and prior methods provided in the rebuttal is very clear. I now better appreciate that the core advantage of EMF lies in utilizing finite difference to bypass the unstable JVP. Although this may appear to be a relatively minor theoretical refinement, the authors have effectively demonstrated its significance through both theoretical analysis and empirical results. However, as the current manuscript is still somewhat unpolished and requires substantial revisions, I have decided to maintain my score as a Weak Accept.

---

> > > ### Author Response · Authors · 2026-04-03
> > >
> > > Thank you again for your thoughtful follow-up and for recognizing that our rebuttal addressed the main concerns. We also sincerely appreciate your candid feedback that the current manuscript would benefit from further polishing and revision.
> > >
> > > We would like to note that we are already working to revise the paper accordingly. In particular, the planned improvements to the experimental presentation have been described in more detail in our response to reviewer XUbN. We are also revising the manuscript to strengthen the intuition and visual explanation of the method, and to provide a clearer summary of the core practical and conceptual advances of EMF in the Introduction. Our goal is to address the remaining presentation concerns as carefully as possible and to make the final manuscript clearer and more polished.
> > >
> > > If there are any particular aspects that you feel would especially benefit from further clarification or revision, we would be sincerely grateful for any additional guidance.

---

### Official Review · Reviewer_XUbN · 2026-03-11

**Soundness:** 3
**Presentation:** 1
**Significance:** 2
**Originality:** 2
**Overall Recommendation:** 3
**Confidence:** 4

**Summary:**

This paper introduces Euler Mean Flows (EMF), a novel framework designed to overcome the computational inefficiencies of standard Mean Flows by eliminating the reliance on Jacobian-vector product (JVP) operations. By establishing a JVP-free formulation based on a theoretically and empirically validated locality assumption, the proposed EMF framework enables highly efficient single-step generative modeling. Furthermore, the authors demonstrate that EMF significantly enhances training speed and computational efficiency, ultimately achieving superior single-step image generation performance on challenging benchmarks such as ImageNet and CelebA-HQ-256 compared to existing baselines, including the Shortcut Model.

**Compliance With Llm Reviewing Policy:**

Affirmed.

**Final Justification:**

While the concerns regarding the novelty and technical value of this work have been largely resolved, and the claims and content of the paper are acceptable, I have a remaining concern about the overall organization of the manuscript. Specifically, many of the key claims and the experimental results that are essential to supporting those claims are placed in the appendix rather than in the main paper. While making good use of the appendix is not inherently problematic, I believe that at minimum, the core results and discussions necessary to substantiate the paper's central arguments should be present in the main text.

I am aware that this concern may be somewhat subjective, and I would like to discuss with the AC and other reviewers whether the current use of the appendix is appropriate before finalizing my assessment. If the AC and other reviewers do not share this concern, or if the authors provide a clear and concrete revision plan, I will raise my assessment accordingly.

**Key Questions For Authors:**

Please refer to the detailed points raised in the Weaknesses section above for full context. Specifically, the authors are strongly encouraged to address the following critical questions during the rebuttal:

1. **Methodological Novelty:** How does the proposed Euler Mean Flows fundamentally differ from the Shortcut Model, given that the final objective relies heavily on Equation 6 (sharing the trajectory consistency objective with the Shortcut Model) rather than the standard Mean Flows formulation in Equation 5?
2. **Hyperparameter Design ($\Delta t$):** What is the exact value or sampling strategy used for $\delta t$? Could an ablation study or further empirical analysis be provided to justify this specific design choice?
3. **Experimental Setups:** What is the rationale for conducting the computational cost comparison on ImageNet with smaller batch sizes? Additionally, why was aux-EMF utilized solely for the cost comparison and excluded from the main performance evaluation?
4. **Missing Results:** Can the missing 128-batch, 4-step FID scores for both EMF and MF on CelebA-HQ-256 (Table 6), as well as the baseline score for JiT with its default configuration (Table 7), be provided?

**Limitations:**

Yes.

**Strengths And Weaknesses:**

### Strengths

* The paper proposes an efficient framework, Euler Mean Flows (EMF), which is JVP-free while demonstrating successful single-step generation performance across various tasks.
* The validity of the locality assumption in EMF is supported by both theoretical and empirical results.
* Compared to the Shortcut Model, which also supports JVP-free single-step generative modeling, the proposed EMF achieves superior performance on benchmarks such as ImageNet and CelebA-HQ-256.
* It is empirically demonstrated that EMF enhances training speed and efficiency compared to standard Mean Flows by effectively eliminating the JVP operation.

### Weaknesses

**1. Methodological Novelty and Relation to the Shortcut Model:** Although EMF is introduced as a JVP-free formulation of Mean Flows, it fundamentally appears to be a specific instantiation of the Shortcut Model. Specifically, Equation 4 represents a trajectory consistency objective that was previously proposed and incorporated as a core objective in the Shortcut Model. Furthermore, as stated in the first paragraph of Section 4.2, Equation 5 (the standard Mean Flows formulation) is  bypassed in favor of Equation 6 to derive the final objective. However, this strong reliance on the consistency loss term from the Shortcut Model is not discussed in the manuscript, raising concerns regarding the novelty of EMF.

**2. Missing Details on Key Hyperparameters ($\Delta t$):** Following the observation above, the primary distinction between EMF and the Shortcut Model seems to lie in the configuration of the time step difference, $\Delta t$, between $t$ and $r$. While the Shortcut Model sets $\Delta t = (t - r) / 2$, EMF sets $\Delta t$ to a minimum value. However, the exact value used, the sampling strategy for $\Delta t$, and an ablation study justifying this critical design choice are conspicuously missing from the manuscript.

**3. Manuscript Completeness and Organization:** The overall presentation falls short of the standards expected for ICML. Despite the main body remaining under the 8-page limit, crucial experimental results—including reported scores and setup details necessary to assess the validity of the claims—are relegated to the appendix. Consequently, the main text fails to independently substantiate its core arguments. Reorganizing the content to ensure the main manuscript is self-contained is strongly recommended.
Additionally, while demonstrating effectiveness across various setups is commendable, including an excess of supplementary experiments within the main text dilutes the core message. The central contribution should focus on the inefficient training cost of Mean Flows (due to JVP) and the introduction of EMF as an alternative. Therefore, it is suggested that Sections 6.2.3–6.2.5 be moved to the appendix, allowing the main text to focus deeply on ImageNet and the paper's key findings.

**4. Unjustified Experimental Setups and Missing Data:** Several experimental configurations lack proper justification, making it difficult to fully trust the empirical results:
* The rationale for conducting the cost comparison on ImageNet with smaller batch sizes remains unclear.
* It is unexplained why aux-EMF is utilized exclusively for computational cost comparisons but excluded from the main performance benchmarks.
* In Table 6, the 128-batch, 4-step FID scores for EMF and MF on CelebA-HQ-256 are missing. Assuming the trained models are available, these scores must be reported for a comprehensive evaluation.
* In Table 7, the score for JiT with its default configuration (x-pred, u-loss) is omitted and should be provided.

---

> ### Author Rebuttal · Authors · 2026-03-31
>
> ## 1.Novelty and relation to Shortcut
> **The main question we address is:can long flow maps get direct data supervision without JVPs?** So our work is not just a better MeanFlow.It asks a basic question in long flow-map learning.See also our reply to **Reviewer Da3y** for the motivation.For **Shortcut**,our point is as follows.
>
> We agree that both EMF and Shortcut involve trajectory consistency(Eq.4);this is also true for MeanFlow,AlphaFlow,and related methods.However,this does not imply EMF is a specific instantiation of the Shortcut Model.The key difference is how supervision is given to long-range transport,and this makes EMF and Shortcut different methods.
>
> The trajectory consistency loss
> $L^{TC}=E[|\psi^\theta_{t\to r}(x_t)-\psi^\theta_{s\to r}(\psi^\theta_{t\to s}(x_t))|^2]$
> does not itself give direct data supervision,since all terms come from the model.It enforces consistency,but does not ensure that long flow maps match the target distribution.Shortcut adds data supervision only at the FM boundary through $|s_\theta(x_t,t,0)-(x_1-x_0)|^2$,
> and then extends it to longer ranges by $E|s_\theta(x_t,t,2d)-(s_\theta(x_t,t,d)+s_\theta(x_{t+d},t,d))/2|^2$
> with $d=1/128,...,1/2$.**So for long-range maps,Shortcut’s loss uses only model outputs,without direct supervision from dataset samples**;long maps are learned from shorter learned maps,rather than through direct data supervision.
>
> **Our method differs here.** We inject direct data supervision into the objective for long-range flow maps: $|u^\theta_{t\to r}(x_t)-(u_t(x_t\mid x_1)+sg((r-t-\Delta t)\frac{u^\theta_{t+\Delta t\to r}(x_t')-u^\theta_{t\to r}(x_t)}{\Delta t}))|^2$ with $u_t(x|x_1)=\frac{x_1-x}{1-t},x_1\in D$. **This answers our main question:long flow maps can get direct data supervision without JVPs.** Thus,unlike Shortcut,our long-range loss is not only model consistency:it still has an explicit term from $D$,while avoiding JVPs.Theorems 4.3 and 4.4 justify it.
>
> **Shortcut extends boundary supervision step by step,while our method keeps direct data supervision in the long-range loss itself,without JVPs.** We will make this clearer in the revision.
>
> Unlike Shortcut,our method uses a fixed $\Delta t=0.001$ rather than a schedule,which is simpler in practice.
>
> ## 2.Choice of $\Delta t$
>
> In all experiments,we use a fixed $\Delta t=10^{-3}$.Our method uses a forward Euler step over the short interval $t\to s=t+\Delta t$,so $\Delta t$ should stay small.Thus,we do not use a $\Delta t$ schedule.We agree this should have been stated more clearly.We also show a study below.The results show stability over a wide range of small $\Delta t$(CelebA-HQ).
>
> |$\Delta t$|1e-4|3e-4|6e-4|1e-3|2e-3|3e-3|1e-2|3e-2|1e-1|
> |-|-|-|-|-|-|-|-|-|-|
> |FID|25.8|12.7|10.9|10.9|10.7|10.8|10.7|12.2|25.2|
>
> ## 3.Paper organization
>
> In the revision,we will move Assumptions 1–2 and Lemma 1 to the appendix,and move more key results to the main text.
>
> We briefly clarify why the SDF and point-cloud results matter(6.2.3--6.2.5).Our claim is not only better FID and lower cost,but also **better stability** and **wider applicability** without JVPs.In particular,point-cloud uses PVCNN,whose sparse C++/CUDA ops do not support JVPs (Sec.5.3),so MeanFlow cannot be applied there.
>
> ## 4. Setups and missing data
> We thank the reviewer for noticing these issues.
> - **ImageNet batch size:** the table text has a typo.All ImageNet runs use batch size 256,as already shown in Table 3.We will fix this.
> - **aux-EMF:** We mainly introduce aux-EMF for unconditional generation,where EMF’s cost gain over MeanFlow is smaller than in the conditional case(see L365–L384,left).So we test aux-EMF only there.For conditional generation,EMF already gives a clear cost gain,so aux-EMF is less needed and may hurt accuracy.
> - **Missing CelebA-HQ-256 scores:** We sincerely apologize that the 128-step and 4-step FID entries in Table 6 were accidentally left blank due to the rush before the deadline.
>
> |Method|128-step|4-step|1-step|
> |-|-|-|-|
> |MF|7.8|8.9|12.4|
> |EMF|7.6|8.4|10.8|
> - **Missing JiT default result/label errors:** we agree that the **JiT default baseline** and some missing baseline variants should be included.These were missed by the deadline.We also found label errors in the original table: the two EMF rows for(x1-pred,x1-loss)and(x1-pred,u-loss) were swapped, and the JiT row for(x1-pred,u-loss)was mislabeled as(x1-pred,x1-loss).The corrected table is below.We also add results for the x1-pred version of MeanFlow,which was introduced in the concurrent work Pixel Mean Flow.
>
> |Method|Variant|128-Step|2-Step|1-Step|
> |-|-|-|-|-|
> |JiT|$u$-pred,$u$-loss|339.7|384.4|407.0|
> |JiT|$x_1$-pred,$x_1$-loss|42.3|429.3|399.4|
> |JiT|$x_1$-pred,$u$-loss|27.9|441.6|440.1|
> |MeanFlow|$u$-pred,$u$-loss|321.3|323.9|327.0|
> |MeanFlow|$x_1$-pred,$x_1$-loss|42.2|41.5|56.8|
> |MeanFlow|$x_1$-pred,$u$-loss|24.0|26.6|39.2|
> |EMF|$u$-pred,$u$-loss|329.4|323.3|324.6|
> |EMF|$x_1$-pred,$x_1$-loss|35.8|34.8|36.3|
> |EMF|$x_1$-pred,$u$-loss|21.4|26.4|30.6|

---

> > ### Author Rebuttal · Reviewer_XUbN · 2026-04-02
> >
> > While the concerns about novelty of this paper and lack of details are resolved, I have following remaining questions.
> >
> > > 2. Choice of $\Delta_t$
> >
> > Could the authors clarify why the performance degrades with smaller $\Delta_t$​ (e.g., 1e-4)? Intuitively, a smaller $\Delta_t
> > $ should lead to a more accurate approximation in EMF, and thus one would expect performance to improve rather than degrade. An explanation of this seemingly counterintuitive behavior would be appreciated.
> >
> > > 3. Paper organization
> >
> > While the concerns regarding the novelty and technical value of this work have been largely resolved, and the claims and content of the paper are acceptable, I have a remaining concern about the overall organization of the manuscript. Specifically, many of the key claims and the experimental results that are essential to supporting those claims are placed in the appendix rather than in the main paper. While making good use of the appendix is not inherently problematic, I believe that at minimum, the core results and discussions necessary to substantiate the paper's central arguments should be present in the main text.
> > I acknowledge that the authors have indicated their intention to move experimental content from the appendix to the main paper in the revision. However, given the current structure, it is unclear whether sufficient space exists in the main paper to accommodate the minimum necessary results and discussions for all presented experiments. Therefore, I would appreciate it if the authors could provide a concrete revision plan—specifically, what content in the main paper will be reorganized, and which experiments or tables from the appendix (with specific references) will be moved to the main text to ensure the completeness of the paper.
> >
> > I am aware that this concern may be somewhat subjective, and I would like to discuss with the AC and other reviewers whether the current use of the appendix is appropriate before finalizing my assessment. If the AC and other reviewers do not share this concern, or if the authors provide a clear and concrete revision plan, I will raise my assessment accordingly.

---

> > > ### Author Response · Authors · 2026-04-02
> > >
> > > Thank you for your encouraging feedback.We also appreciate the insightful question.
> > >
> > > ## 2. Choice of $\Delta t$
> > > For finite-difference computation, the total **numerical error** has two parts: (1) **truncation error**, and (2) **roundoff error from finite precision** ([1], p. 181).
> > > [1] Richard L. Burden and J. Douglas Faires. *Numerical Analysis*, 9th Edition.
> > >
> > > - **Truncation error.** This is the error of the formula itself **in infinite precision**.For our forward-difference scheme,it is $O(\Delta t)$.**At this level,as the reviewer noted,a smaller $\Delta t$ gives a better approximation.**
> > >
> > > - **Roundoff error.** In practice,we use **finite precision** (e.g.,fp32 or fp16).This error does **not** always decrease as $\Delta t$ gets smaller.When $\Delta t$ is small,it can become large.**Then making $\Delta t$ even smaller can make the result worse.**
> > >
> > > This explains our trend.When $\Delta t$ is large,truncation error dominates,so a smaller $\Delta t$ helps.When $\Delta t$ is very small,roundoff error dominates,so a smaller $\Delta t$ hurts.Thus,our $\Delta t$ study looks for a range that balances these two errors and minimizes the total numerical error ([1],p.181).In our study,this range is fairly wide,which supports the robustness of our method.
> > >
> > > We explain this effect in two ways.
> > >
> > > - **(1)Roundoff error.** Suppose we use FP16,whose effective resolution is about $6\times 10^{-5}$.Let $f(t)=1+t$,so $f'(0)=1$.If $\Delta t=10^{-6}$,then $1+\Delta t$ may still be stored as $1$.Then $\frac{f(0+\Delta t)-f(0)}{\Delta t}=\frac{1-1}{10^{-6}}=0$,which is wrong.But if $\Delta t=10^{-1}$,then $\frac{f(0+\Delta t)-f(0)}{\Delta t}=\frac{1.1-1}{10^{-1}}=1$,which is correct.So,in finite precision,a smaller $\Delta t$ does not always give a better finite-difference result.
> > >
> > > - **(2) Toy numerical example.** We also build a simple test:$f(t)=(I+\alpha tA)^Kx$,and compare the finite-difference approximation against the closed-form derivative $f'(0)=K\alpha Ax$,where $\alpha=10^{-2}$,$K=32$,$A\in\mathbb{R}^{256\times256}$,and $x\in\mathbb{R}^{256}$.We set $A=V\Lambda V^{-1}$, where $\Lambda=\mathrm{diag}(\lambda_1,\ldots,\lambda_n)$ has log-spaced entries in $[1,100]$,and $V$ is a matrix with condition number $100$.
> > >
> > > |$\Delta t$|3e-5|1e-4|3e-4|1e-3|3e-3|6e-3|1e-2|2e-2|3e-2|1e-1|
> > > |-|-|-|-|-|-|-|-|-|-|-|
> > > |error|0.98|0.79|0.53|0.25|0.05|0.02|0.02|0.08|0.22|10.1|
> > >
> > > The error curve is U-shaped: it first goes down as $\Delta t$ gets smaller,since truncation error drops;after about $6\times10^{-3}$,it goes up again,since roundoff error dominates.This is the same trend predicted by theory,and it matches our experiments.
> > >
> > > So,the worse result at very small $\Delta t$ does not contradict finite-difference theory.It is the usual trade-off between **truncation error** and **roundoff error**.The key is to find a range that balances them.
> > > ## 3.Paper organization
> > > We agree that the key results for the main claims should be in the main paper,not mostly in the appendix.We have a clear revision plan,and it fits within the page limit.
> > >
> > > - **Free space from theory.** In the current draft,Assumption 1,Lemma 1,Assumption 2,and their validation experiments(Sec.6.1)take substantial space.These are not the direct proof of our main claims.The main theory results are Thm.4.3 and Thm.4.4.The assumptions,lemma,and validation mainly support these theorems,so we will move them to the appendix and keep only short references in the main text,e.g.,“Based on Assumption A (validated in App.B)and Lemma C,we have...(Thm.4.3).”This frees more than 3/4 page.We will also move Function-Based Image Generation to the appendix,since it is more supplementary than the other four applications.Together,these changes free about one page.
> > >
> > > - **Move key tables to the main paper.** We will move five key tables and discussions into the main text:(1)Latent Image Generation(Tab.6);(2)Pixel-Space Image Generation(Tab.7);(3)the memory/speed/quality table(Tab.5);(4)SDF(Tab.9);(5)the $\Delta t$ study table.We will also add the key rows that are now only in the rebuttal,so the main paper contains the core comparisons for our claims.
> > >
> > > - **Layout is feasible.** Tables(2),(3),(4),(5) can be long single-column tables,and Table (1)can be double-column.For Table(3),we can remove the Dataset column and split ImageNet and CelebA with row headers.For Table(2),we can shorten u-pred/u-loss to u/u and explain it in the caption.Table(1)will stay double-column.After these changes,all key tables fit in the freed space,and we still have about 14 single-column lines left.
> > >
> > > - **Use the remaining space for clarity.** We will use the remaining lines to expand Sec.4.1 by adding the key formulas now given in our rebuttal.The current intuition part is mostly verbal,and these formulas will make it easier to follow.
> > >
> > > So,within the 8-page limit,we can move the key experimental tables into the main paper and also strengthen the intuition part.We believe this will clearly improve the paper organization and completeness.

---

### Official Review · Reviewer_1QyJ · 2026-03-12

**Soundness:** 3
**Presentation:** 2
**Significance:** 3
**Originality:** 2
**Overall Recommendation:** 4
**Confidence:** 4

**Summary:**

This paper studies one-step and few-step generative modeling from the perspective of **trajectory consistency** in flow-based models. The authors argue that existing approaches to learning long-range flow maps either construct them indirectly from short-range transitions, which may accumulate approximation error, or rely on gradient-based consistency objectives such as JVP-related formulations, which can be computationally expensive and unstable in practice.

To address this, the paper proposes **Euler Mean Flows (EMF)**, a new framework that starts from the semigroup property of flow maps and derives a tractable surrogate objective through **local linearization**. This formulation provides more direct supervision for long-horizon flow-map prediction while avoiding explicit Jacobian-related computation. Based on this idea, the paper further develops a **JVP-free training framework** that supports both **u-prediction** and **x1-prediction** within a unified view.

The paper also includes a theoretical analysis suggesting that, under mild assumptions, the proposed surrogate objective approximates the original trajectory consistency objective. Experimentally, EMF is evaluated on a diverse set of tasks, including latent-space image generation, pixel-space image generation, signed distance function generation, point cloud generation, and function-based image generation. The reported results indicate that EMF improves training stability and generation quality under fixed step budgets, while also reducing memory and runtime costs compared with prior methods such as MeanFlow.

**Compliance With Llm Reviewing Policy:**

Affirmed.

**Final Justification:**

The authors’ detailed and thoughtful rebuttal has satisfactorily addressed all my questions, with no further concerns from my side. I will maintain my existing review recommendation, which was intentionally framed on the positive side to reflect my view that the post-rebuttal manuscript falls into the borderline accept range.

**Key Questions For Authors:**

1. **On the validity of the local linearization surrogate:**
   The method is motivated by approximating the original trajectory-consistency objective through a local Euler-style linearization. Could the authors clarify under what practical regimes this approximation is expected to be sufficiently tight? In particular, do you have empirical evidence on how sensitive performance is to the local interval size, time-pair sampling strategy, or the fraction of samples where the approximation is applied over very short vs. longer spans?
   *Why this matters:* A convincing response would strengthen my confidence that the proposed surrogate is not only theoretically motivated but also robust in realistic training settings. If the method is highly sensitive to these choices, that would weaken my assessment of its practical soundness.

2. **On the gap between the theory and the full practical method:**
   The theoretical justification appears to rely on assumptions such as reasonably accurate local field learning, yet the empirical discussion suggests that in some domains the \(u\)-prediction formulation can fail and that \(x_1\)-prediction is needed instead. Could the authors explain more clearly how the theory should be interpreted in the regimes where \(u\)-prediction is inadequate? Is there a principled reason why the \(x_1\)-prediction variant should remain well behaved beyond the current empirical evidence?
   *Why this matters:* If the authors can better connect the practical success of the full method to the theoretical analysis, my confidence in the paper’s soundness would increase. Otherwise, I would view the theory as somewhat narrower than the empirical claims.

3. **On ablations isolating the source of improvement:**
   Could the authors provide more detailed ablations that separate the benefits of (i) the Euler/local-linearization surrogate itself, (ii) removing JVP computation, and (iii) switching between \(u\)-prediction and \(x_1\)-prediction? Right now, it is somewhat difficult to tell which component contributes most to the final gains in stability, quality, and efficiency.
   *Why this matters:* A strong ablation would make the contribution much clearer and help establish whether the paper’s main advance is conceptual, algorithmic, or primarily an implementation-level efficiency gain. This could positively affect my assessment of both soundness and originality.

4. **On comparison to the closest related work:**
   Since the paper is closest in spirit to MeanFlow and related long-range consistency methods, could the authors sharpen the comparison by explicitly stating what new capability is enabled by EMF beyond improved efficiency? For example, are there settings where prior JVP-based methods are unusable or fundamentally unstable, but EMF remains trainable?
   *Why this matters:* The answer would directly affect my view of the paper’s originality and significance. If EMF clearly opens regimes that prior methods cannot handle well, I would view the contribution as more substantial.

5. **On scalability and generality:**
   The experiments cover several domains, which is a strength, but could the authors comment on how EMF scales with model size and dataset complexity relative to prior methods? In particular, does the efficiency advantage remain similar at larger scales, or is it mainly visible in the current benchmark regime?
   *Why this matters:* A stronger response here would increase my confidence in the broader significance of the work. If the gains persist at larger scales, I would see the method as more likely to influence future practice.

**Limitations:**

no

The paper does not adequately discuss either its methodological limitations or its potential societal impact. The conclusion briefly mentions future work, such as extending the method to broader tasks, larger models, and more general theoretical settings, but this is not a substantive discussion of limitations. In addition, the Impact Statement is extremely brief and effectively states that there may be societal consequences but that none need to be specifically highlighted, which is not sufficient.

I would encourage the authors to strengthen this part by explicitly discussing: (1) when the local linearization surrogate may fail or become inaccurate, (2) how sensitive the method is to time discretization, prediction parameterization, and domain choice, (3) whether the gains are expected to persist at larger scales, and (4) the standard misuse risks associated with faster and cheaper generative modeling systems. A clearer statement of the scope and boundaries of the current empirical/theoretical claims would also improve transparency.

**Strengths And Weaknesses:**

**Strengths and Weaknesses**

This submission tackles an important problem in modern generative modeling: how to obtain strong one-step or few-step generation while preserving trajectory consistency in flow-based models. Overall, I find the paper technically interesting, practically motivated, and stronger than a purely incremental engineering paper, although I also think some aspects of the theory, empirical validation, and positioning could be sharpened.

**Soundness.**
A major strength of the paper is that the method is not presented as a purely heuristic modification. The authors start from the semigroup perspective on flow maps, identify the difficulty of directly supervising long-range trajectory consistency, and derive a surrogate objective through a local linearization argument. This gives the method a principled foundation. The paper further provides theoretical results connecting the proposed surrogate loss to the original consistency objective under explicit regularity assumptions, and this is stronger than simply proposing a new loss without analysis. On the empirical side, the paper evaluates the method across multiple settings, including latent-space image generation, pixel-space image generation, SDF reconstruction/generation, point cloud generation, and function-based image generation. This breadth supports the claim that the proposed JVP-free formulation is broadly usable rather than narrowly tuned to one benchmark. The practical efficiency gains relative to MeanFlow, especially in memory and training speed, are also meaningful and appear well aligned with the stated motivation.

That said, I do have some soundness-related reservations. The theoretical guarantees rely on assumptions that may be reasonable locally, but they are still fairly strong, and the key condition that the instantaneous field is learned accurately is central to the argument. In practice, the paper itself shows that this condition can fail in some settings, which is why the x1-prediction variant becomes necessary. This does not invalidate the method, but it does mean that the cleanest theory does not fully cover the full practical story. More broadly, the theoretical analysis mainly justifies the surrogate under local approximation regimes; it does not fully explain when the approximation will be tight enough in realistic large-scale training, or how sensitive performance is to choices such as the local step size, time sampling, or the proportion of samples with r=t. I also would have liked stronger ablations isolating which parts of the method matter most in practice, especially the local-linearization design choices, the time-weighting in x1-prediction, and the training heuristics used for stability.

**Presentation.**
The paper is generally well structured, and the high-level narrative is clear: trajectory consistency is important, existing direct methods are expensive or unstable because of JVP-based objectives, and EMF replaces them with a local Euler-style surrogate that is easier to optimize. The problem formulation, method, and experiments follow a sensible progression. I also appreciate that the paper attempts to unify u-prediction and x1-prediction within one framework, and that it includes concrete algorithmic descriptions rather than leaving the method at an abstract level.

However, the presentation could still be improved. The notation is quite dense, and several parts of the derivation are difficult to parse on a first read, especially the transition from the semigroup relation to the practical surrogate loss. The discussion around why the conditional flow-map view breaks down, why the u-prediction formulation is insufficient in some domains, and how exactly the x1-prediction version remedies this is somewhat scattered and would benefit from a clearer conceptual explanation. I also think the paper could do a better job of positioning itself relative to closely related methods such as MeanFlow, SplitMF, and Shortcut in a more explicit “what is inherited, what is changed, and why this matters” comparison. Right now, the main novelty can be understood, but the paper makes the reader work harder than necessary to isolate it.

**Significance.**
I view the problem as significant. One-step and few-step generation remain important because inference efficiency is a real bottleneck, and methods that reduce training overhead while preserving generation quality are practically valuable. The JVP-free aspect is especially meaningful because it improves compatibility with efficient implementations and with sparse computation settings that are otherwise difficult for gradient-based consistency objectives. The paper’s experiments on point clouds, functional representations, and SDFs make this practical point stronger: the contribution is not only “a slightly better image-generation loss,” but also a training formulation that may unlock settings where JVP-heavy methods are inconvenient or unsupported. If the method proves robust across more architectures and scales, I can see it being adopted as a useful recipe for efficient flow-based generation.

At the same time, I would characterize the significance as moderate rather than transformative. The work is best understood as an improved training formulation within an already active line of research, not as a fundamentally new paradigm for generative modeling. The strongest practical evidence is against closely related flow-consistency baselines, especially MeanFlow, and while those comparisons are relevant, the broader impact would be easier to judge with more extensive large-scale experiments and stronger demonstrations of downstream adoption advantages. So I do think the paper addresses an important problem and makes a useful advance, but I am somewhat less convinced that it substantially changes the broader trajectory of the field.

**Originality.**
The paper has a real originality component, though in my view it is moderate rather than very high. The key idea—replacing a difficult long-range consistency objective with a locally linearized Euler-style surrogate derived from the semigroup property—is conceptually meaningful and not merely a small hyperparameter tweak. The JVP-free training formulation is also a practically relevant innovation, and the extension to both u-prediction and x1-prediction under one umbrella adds value. In that sense, the work offers a new perspective on how to train one-step flow-based generators more efficiently.

The main limitation on originality is that the work sits very close to recent methods on trajectory consistency and long-range flow-map learning. The overall framing, goals, and even some of the mathematical objects are strongly tied to MeanFlow and related papers. As a result, the paper may feel to some readers like a refined and more practical variant of that line rather than a genuinely distinct conceptual departure. I do think the paper has enough novelty to stand on its own, especially because the JVP-free surrogate seems nontrivial and practically meaningful, but the authors should be careful not to oversell the originality relative to the most closely related prior work.

**Overall assessment.**
Overall, the paper’s main strengths are its principled motivation, its practically meaningful JVP-free formulation, its broad empirical scope, and its convincing efficiency/stability improvements over closely related baselines. Its main weaknesses are that the theory depends on assumptions whose practical validity is only partially established, the empirical analysis could include stronger ablations and sensitivity studies, and the originality is meaningful but still relatively close to existing trajectory-consistency methods. I see this as a technically solid and relevant contribution with clear practical value, even if the novelty and broader impact are somewhat more moderate than the strongest papers in the area.

---

> ### Author Rebuttal · Authors · 2026-03-31
>
> ## 1.The local-linearization surrogate
> EMF uses three times $t$,$s=t+\Delta t$,and $r$.Here $\Delta t$ is very small,while $r-t$ can be long.The local linearization is used only on $[t,s]$,not on $[t,r]$.EMF also does not need special sampling to be effective.We use the same default $t,r$ sampling as MeanFlow:75% of samples use $r=t$,and otherwise $t,r$ are sampled at random uniformly.
>
> We use fixed $\Delta t=1e-3$ in the paper,and here we also add a $\Delta t$ study:
>
> |$\Delta t$|1e-4|3e-4|6e-4|1e-3|2e-3|3e-3|1e-2|3e-2|1e-1|
> |-|-|-|-|-|-|-|-|-|-|
> |FID|25.8|12.7|10.9|10.9|10.7|10.8|10.7|12.2|25.2|
>
> If $\Delta t$ is too small,the diff term is noisy.If it is too large,the local fit is poor.EMF is stable from 6e-4 to 1e-2,showing robustness.
>
> ## 2.Theory vs.Practice
> In fact,theory and practice match well.Our theory does not say that u-pred should always work;rather,it says that u-pred EMF requires an accurate learned local target.This is exactly what we observe:u-pred EMF fails in the same regimes where u-pred FM already fails,namely where the local velocity field is hard to learn.Theoretically,Theorem 4.3 shows that the gap between the original Trajectory Consistency Loss and u-pred EMF is controlled by $\mathbb{E}\|u_{t\to t}^\theta(x)-u_t(x)\|^2$.Thus,when the local velocity field is hard to learn,u-pred EMF should degrade.Empirically,for pixel-space image generation,Table 7 shows poor u-pred FM(first row,consistent with JiT),and u-EMF is also poor;for SDF generation,Fig.22 shows the same trend:u-pred FM is poor,then u-EMF is poor.
>
> We use x1-pred there following JiT,mainly for high-dimensional settings.EMF does not rely on one prediction form,but on whether the local target can be learned well.As discussed in JiT,in high-dimensional tasks,the x1-pred target stays closer to the data manifold,while velocity/noise targets are more spread out and harder to learn.Therefore,in tasks such as image generation and SDF generation,x1-pred EMF is needed because its local target can be learned more accurately.
>
> In sum,theory and experiments tell the same story:when local velocity learning works well,u-pred EMF works well;when it is hard,u-pred EMF degrades for exactly the reason described by Theorem 4.3.This usually happens in high-dimensional settings,where x1-pred EMF is needed.
>
> ## 3.Ablations
> We do not claim that $x_1$-pred is always better than u-pred.In EMF,u-pred is the default,while x1-pred is used only in high-dimensional tasks such as pixel-space image generation and SDF generation.So its gain is task-based(Fig.22 and the last three rows of Tab.7).For latent-space and point-cloud generation,we still use u-pred,as in prior work.So those results already show the gain of EMF apart from the change in prediction form.
>
> We separate the roles of removing JVP and the local-linearization surrogate below.Removing JVP alone mainly improves stability(loss curve in fig.4)and efficiency,with about 50\% gains in memory and speed in our setting,but without the surrogate the long flow map loses direct data supervision and gives the worst final performance.The local-linearization surrogate restores direct data supervision in the JVP-free setting.Combining the two gives EMF,which achieves both better stability and stronger final performance.
>
> |Method|EMF|w/o removing JVP|w/o Local Linearization|
> |-|-|-|-|
> |FID|10.9|12.4|13.0|
>
> In sum,x1-pred mainly helps high-dimensional tasks; removing JVP mainly improves stability,efficiency,and applicability; and the local-linearization surrogate preserves direct data supervision for long flow maps without JVP.
> ## 4.The closest related work
> The core question is:can long flow maps get direct data supervision without JVPs?Prior methods cannot.Progressive Extension methods(ShortCut,SplitMeanFlow,PSD)avoid JVPs,but their long flow maps do not get direct data supervision.Continuous-based methods(MeanFlow,AlphaFlow,LSD,ESD)do give direct supervision,but need JVPs.Our method fills this gap by giving direct data supervision to long flow maps without JVPs.Due to space limits,more detail is in our reply to **Reviewer Da3y,Q1**
>
> Beyond speed,EMF also boosts final quality and works in 2 cases that MeanFlow does not.Point-cloud generation often uses PVCNN,whose sparse C++ libs often do not support JVP,so MeanFlow cannot be used while EMF can.Also,under mixed-precision ImageNet training,Fig.18 shows that MeanFlow breaks late in training,while EMF stays stable.
>
> ## 5.Scaling and generality
> We agree that tests on larger image models and datasets would be useful,but could not add them due to limited computing resources.To partly make up for this,we tested more 3D tasks to check whether the JVP-free form stays stable and useful.We will add the scaling test as future work.
>
> ## 6.Limitation
> We agree and will clarify the failure cases of the local surrogate, the sensitivity to discretization/parameterization/domain, the expected scope at larger scales, and the standard misuse risks of faster generative systems.

---

> > ### Author Rebuttal · Reviewer_1QyJ · 2026-04-02
> >
> > Thank you for the detailed rebuttal. My concerns are largely resolved. The authors provide a much clearer explanation of the practical regime in which the local-linearization surrogate is expected to work, including a useful sensitivity study on the local step size, and they also clarify the relationship between the theory and the practical use of $u$-prediction versus $x_1$-prediction. I also appreciate the added ablation separating the effects of removing JVP from those of the local-linearization surrogate, as this makes the source of the gains much clearer. In addition, the rebuttal sharpens the comparison to the closest related work by explaining more explicitly what EMF enables beyond efficiency alone, including settings where prior JVP-based methods are difficult or unstable to use. While broader large-scale scaling evidence would still be valuable in the final paper, I think the rebuttal addresses the main technical and conceptual concerns sufficiently and improves my confidence in the contribution.

---

> > > ### Author Response · Authors · 2026-04-02
> > >
> > > Thank you again for your thoughtful and constructive feedback. We are very grateful for the opportunity to clarify our work, and we are glad that our rebuttal helped address the main concerns regarding the practical regime of the surrogate, the connection between theory and practice, the source of the observed gains, and the comparison with prior JVP-based methods. If there are any remaining concerns that we have not yet addressed sufficiently, we would sincerely appreciate the opportunity to clarify them further. We would also be grateful if the clarifications provided in the rebuttal could be taken into account in your final assessment.

---

### Official Review · Reviewer_Da3y · 2026-03-12

**Soundness:** 2
**Presentation:** 2
**Significance:** 3
**Originality:** 2
**Overall Recommendation:** 4
**Confidence:** 4

**Summary:**

The paper, Euler MeanFlows (EMFs), presents a new family of MeanFlow-like loss formulations for few-step generation which circumvent the need for memory extensive vector-Jacobian products. The approach poses both a velocity and $x_1$ prediction formulation, each with its own set of benefits. There is significant theory to justify the loss formulation, with additional experiments proposed to benchmark performance relative to vanilla MeanFlows.

**Compliance With Llm Reviewing Policy:**

Affirmed.

**Final Justification:**

Many of my concerns around the benchmarking were resolved with dedicated experiments and clearer justification. The results obtained are also promising.

**Key Questions For Authors:**

- How well does the approach perform relative to the best MeanFlow-variants in terms of FID performance and memory use (https://arxiv.org/abs/2510.20771, https://arxiv.org/abs/2505.18825)? This includes methods that do require JVP calls.

- Comparing with SplitMeanFlow - a method that does not require JVP calls - how does FID performance and memory use compare?

- How does the training stability compare between methods that do not require JVP calls (SplitMeanFlow vs. EMFs)?

- How do the data prediction variants of the loss formulation compare to the ESD, LSD, and PSD loss formulations by Boffi et al. (https://arxiv.org/pdf/2505.18825)?

- How does the approach perform in distillation settings compared to other MeanFlow variants?

**Limitations:**

Yes

**Strengths And Weaknesses:**

**Strengths**

- The additional memory requirements of the JVP calls in conventional MeanFlow training are not insignificant. JVPs can significantly increase memory usage and implementation complexity, especially in sparse or irregular domains. The approach demonstrates that removing these calls can roughly improve memory use by 2$\times$ in the reported experiments.

- There is a moderate but consistent improvement over vanilla MeanFlows across several tasks (latent images and SDF generation).

- In addition to a velocity prediction variant, a $x_1$-prediction loss has also been derived and included, which clearly has value in cases where velocity prediction fails---a clear example for this is shown in the paper.

**Weaknesses**

- The velocity prediction loss formulation looks like a finite difference approximation to the gradient which serves to replace the JVP call - it remains unclear whether this scales well or how the impact of $\Delta t$ impacts the stability of the loss. I don't clearly see any performed ablations on this local approximation.

- Other variants of MeanFlows that have built on top of it are excluded from benchmarking, e.g., AlphaFlow (https://arxiv.org/abs/2510.20771). This approach is cited and compared to in the relevant work but not contrasted against in terms of FID performance and memory usage.

- Experimental results are buried in the Appendix without clear results included in the main text. Experimental results are limited and a fair comparison against baselines is missing.

---

> ### Author Rebuttal · Authors · 2026-03-31
>
> ## 1. Clarification of Motivation and Contributions
> We clarify that our work centers on one question: **Can long flow maps receive direct data supervision without using JVPs?**, rather than merely improving MeanFlow.
>
> 1.**Motivation**:Trajectory Consistency Loss $L^{TC}=\mathbb{E}[|\psi^\theta_{t\to r}(x_t)-\psi^\theta_{s\to r}(\psi^\theta_{t\to s}(x_t))|^2]$ does not give direct data supervision,since all terms come from the model.It enforces consistency,but does not ensure that long flow maps match the target distribution.Prior works fall into two classes:
> -  **Progressive Extension** (ShortCut,PSD,SplitMeanFlow):these methods extend instantaneous $u_t$ to long flow maps by consistency.For example,Shortcut first learns $|u^\theta_{t\to t}(x)-u_1(x|x_1)|^2,u_1(x|x_1)=\frac{x_1-x}{1-t},x_1\in D$
>       and then extends to longer intervals via $|u^\theta_{t\to t+2d}(x_t)-\frac{1}{2}[u^\theta_{t\to t+d}(x_t)+u^\theta_{t+d\to t+2d}(x_{t+d})]|^2$.Thus,$u^\theta_{t\to t+2d}(x)$ gets no direct supervision from $D$.**Long maps are learned from model outputs,not direct data supervision.**
> - **Continuous methods** (MeanFlow,ESD,LSD):these methods directly supervise long flow maps using data-based conditional velocities.For example,MeanFlow uses $|u_{t\to r}^\theta(x)-sg[u_t(x|x_1)+(r-t)(u_t(x|x_1)\partial_x u_{t\to r}^\theta(x)+\partial_tu_{t\to r}^\theta(x))] )|^2$ with $u_t(x|x_1)=\frac{x_1-x}{1-t},x_1\in D$.This gives direct supervision,but requires JVPs,hurting memory,speed,stability,and use.
>
> **AlphaFlow** combines both and still needs JVPs.
>
> 2.**Contributions**:
> - We ask and answer whether long flow maps can receive **direct dataset supervision without JVPs**.
> - We derive **Euler MeanFlow** $|u_{t\to r}^\theta(x_t)-sg(u_t(x_t|x_1)+(r-t)\frac{u_{t+\Delta t\to r}^\theta(x_t + \Delta t u^\theta_{t\to t}(x_t))-u_{t\to r}^\theta(x_t)}{\Delta t})|^2$ with direct supervision $u_t(x|x_1)=\frac{x_1-x}{1-t},x_1\in D$ for $u^\theta_{t\to r}(x_t)$, no JVPs, and simple fixed $\Delta t$.
> - We further derive an **$x_1$-prediction formulation**,the **first such formulation enabling one-step pixel-space generation** in this framework.
> - Compared with MeanFlow,our method avoids JVPs and improves **stability,memory,speed,and applicability**.
> - We validate this on both **2D and 3D tasks**,including images,point clouds,and SDFs.
>
> ## 2.$x_1$-prediction
>
> **We distinguish the flow map $\psi_{t\to r}(x)$ from its parameterization**.The former is used at inference,while $u_{t\to r}$ and our proposed $x_{t\to r}$ are two training parameterizations of the same flow map.The importance of parameterization is supported by the **JiT** paper and our **Fig.22** and **Tab.7**.In Boffi et al.,ESD/LSD/PSD are defined at the flow-map level and usually instantiated with $u_{t\to r}$;under this choice,ESD recovers MeanFlow,PSD(Uniform) recovers SplitMeanFlow,and PSD(Midpoint) recovers ShortCut.By contrast,we do not redefine the flow map,but introduce an $x_1$-prediction-style parameterization $x_{t\to r}$.Importantly,$x_{t\to r}$ is **not** the endpoint $x_r$,but the intersection of the line through $x_t$ and $x_r$ with the hyperplane $t=1$,analogous to $x_1$-prediction in flow matching (L289--L292,left).The results below show that **$x_1$-pred EMF is better than ESD/LSD/PSD in pixel space**.(CelebA-HQ;JiT)
>
> ||LSD|PSD(U)/SplitMeanFlow|PSD(M)/ShortCut|ESD/MeanFlow|x1-pred EFM|
> |-|-|-|-|-|-|
> |FID|371.7|343.8|406.2|327.0|30.6|
> ## 3.Baseline
> We did not include them originally for the following reason,but now provide comparisons below.**AlphaFlow** still requires JVPs and hybrid training,so it is not a clean baseline for our goal.**SplitMeanFlow** was not evaluated on mainstream datasets and has no official implementation.**Flow Map Matching** is better viewed as a framework than a practical baseline for our setting,as discussed above. All on CelebA-HQ and unified setting.
>
> |Method|PeakMem(TotalMem)(GB)|FixedMem(GB)|Speed/Iter(ms)|FID|
> |-|-|-|-|-|
> |MeanFlow/ESD|32.1|2.3|151.4|12.4|
> |EMF(Ours)|23.3|2.3|91.2|10.9|
> |aux-EMF(Ours)|17.6|2.8|84.2|11.7|
> |AlphaFlow|32.6|2.3|127.7|11.6|
> |SplitMeanFlow/PSD(U)|23.8|2.3|94.6|15.7|
> |ShortCut/PSD(M)|23.8|2.3|94.6|20.5|
> |LSD|52.2|2.3|426.7|12.3|
>
> For SplitMeanFlow,we assess stability by training loss curves(Fig.4)and collapse.Empirically,SplitMeanFlow,like EMF,shows decreasing loss and stable training,but its final FID is much worse than EMF.
> ## 4.$\Delta t$
> We use a fixed $\Delta t=10^{-3}$ in all experiments.The study below shows stability over a wide range of $\Delta t$,supporting its robustness.
> |$\Delta t$|1e-4|3e-4|6e-4|1e-3|2e-3|3e-3|1e-2|3e-2|1e-1|
> |-|-|-|-|-|-|-|-|-|-|
> |FID|25.8|12.7|10.9|10.9|10.7|10.8|10.7|12.2|25.2|
>
> ## 5.Experimental Results and Distillation
>
> In revision,we will move more experiments to the main paper. Our evaluation covers both 2D and 3D tasks.The two 3D tasks,SDF and Point Cloud,are important 3D vision tasks.
>
> Like MeanFlow and AlphaFlow,we focus on training from scratch,not distillation.

---

> > ### Author Rebuttal · Reviewer_Da3y · 2026-04-04
> >
> > I would like to thank the authors for addressing my concerns. I'm happy to raise my score based on the additional experiments performed and the quality of the results obtained.

---

> > > ### Author Response · Authors · 2026-04-04
> > >
> > > Thank you very much for your follow-up and for your positive reassessment of our work. We sincerely appreciate your recognition of our rebuttal, and we are glad that it was able to address your concerns. We are also very grateful for your willingness to raise your score.

---

### Decision · Program_Chairs · 2026-04-30

**Decision:**

Accept (regular)

**Comment:**

This paper proposes Euler Mean Flows (EMF), a JVP-free framework for one-step and few-step generation based on a local linear surrogate for long-range trajectory consistency. Overall, I recommend acceptance. Across the reviews, there is broad recognition that the paper makes a technically meaningful contribution: the method is well motivated, the theoretical development is coherent, and the rebuttal successfully addressed most of the substantive technical concerns, including questions about novelty relative to related methods, the role of the local-linearization surrogate, the choice of the local step size, and comparisons to additional baselines. Multiple reviewers explicitly stated that their main technical concerns were resolved after rebuttal, and the paper’s empirical results support the claim that removing JVP can improve training stability, speed, and memory efficiency while maintaining or improving generation quality across several domains.

My main reservation is about presentation and paper organization, especially the fact that many of the core experimental results are currently placed in the appendix rather than the main paper. I carefully considered Reviewer XUbN’s concern on this point, and after reading the manuscript myself, I agree that this substantially hurts readability and weakens the communication of the paper’s main message. The main text should be more self-contained, and the authors should seriously reconsider the balance between method/theory and experiments so that the central empirical evidence appears in the main paper rather than primarily in the appendix.

That said, I do not believe this writing and organization issue alone should justify rejecting a paper whose technical contribution is otherwise recognized by the reviewers. Since the camera-ready version will allow additional space, I view this as a revision issue rather than a reason for rejection. I therefore recommend acceptance, while strongly encouraging the authors to substantially reorganize the paper for the final version and ensure that the key experimental results and discussions are moved into the main text.